# Synonymous Variational Inference for Perceptual Image Compression

Zijian Liang[1]   Kai Niu[1][2]   Changshuo Wang[1]   Jin Xu[1]   Ping Zhang[3]

## Abstract

Recent contributions of semantic information theory reveal the set-element relationship between semantic and syntactic information, represented as synonymous relationships. In this paper, we propose a synonymous variational inference (SVI) method based on this synonymity viewpoint to reanalyze the perceptual image compression problem. It takes perceptual similarity as a typical synonymous criterion to build an ideal synonymous set (Synset), and approximate the posterior of its latent synonymous representation with a parametric density by minimizing a partial semantic KL divergence. This analysis theoretically proves that the optimization direction of perception image compression follows a triple tradeoff that can cover the existing rate-distortion-perception schemes. Additionally, we introduce synonymous image compression (SIC), a new image compression scheme that corresponds to the analytical process of SVI, and implement a progressive SIC codec to fully leverage the model's capabilities. Experimental results demonstrate comparable rate-distortion-perception performance using a single progressive SIC codec, thus verifying the effectiveness of our proposed analysis method.

## 1. Introduction

Image compression is a typical topic for lossy source coding, aiming to achieve the optimal tradeoff between reconstructed image quality and coding rate. Following the rate-distortion optimization instructed by Shannon's classic information theory (1948; 1959), traditional image compression like JPEG (Wallace, 1991) and BPG (Bellard, 2015)

have pursued this goal with the peak signal-to-noise ratio (PSNR) and multi-scale structural similarity index (MS-SSIM) quality metrics through handcrafted algorithms including transform coding, quantization, and entropy coding. With the growing research in artificial intelligence, recent advancements in learned image compression (LIC) (Ballé et al., 2018; Minnen et al., 2018; Cheng et al., 2020; He et al., 2021; 2022a; Li et al., 2024) combine the optimization principles of traditional image coding with the capabilities of deep learning. Its basic optimization idea can be theoretically analyzed using a variational inference method similar to that in the variational auto-encoder (Kingma & Welling, 2013), leading to a loss function in the form of a rate-distortion tradeoff (Ballé et al., 2017; 2018). These methods have demonstrated significant rate-distortion performance compared with conventional methods.

As the inconsistency exists between "low distortion" and "high perceptual quality", Blau and Michaeli explored the tradeoff between distortion and perception (2018). They further incorporated the perceptual constraint into compression limit analysis and explored the rate-distortion-perception tradeoff (Blau & Michaeli, 2019). Further theoretical analysis (Theis & Agustsson, 2021; Theis & Wagner, 2021; Yan et al., 2021; Qian et al., 2022; Theis, 2024; Hamdi et al., 2025) and empirical results (Agustsson et al., 2019; Mentzer et al., 2020; He et al., 2022b; Theis et al., 2022; Agustsson et al., 2023; Muckley et al., 2023; Hoogeboom et al., 2023; Xu et al., 2023; Careil et al., 2024) demonstrate the effectiveness of this new optimization direction and suggest that perceptual image compression (PIC) with high perceptual quality at low bitrates can be achieved with generative compression (Santurkar et al., 2018) empowered by generative adversarial networks (GAN) (Goodfellow et al., 2014) and diffusion models (Ho et al., 2020).

The great success of the rate-distortion-perception trade-off has effectively shifted the focus from symbol-level accuracy in traditional image compression and tends more towards semantic information accuracy, which aligns with Shannon and Weaver's discussion on communication problem levels (Weaver, 1953). However, these works adopt diverse empirical optimization approaches for perceptual optimization, such as Kullback-Leibler (KL) divergence, discriminator-based adversarial loss like Wasserstein divergence (Blau & Michaeli, 2019), or mixed with "perceptual" measure like

[1]Key Laboratory of Universal Wireless Communications, Ministry of Education, Beijing University of Posts and Telecommunications, Beijing, China [2]Peng Cheng Laboratory, Shenzhen, China [3]State Key Laboratory of Networking and Switching Technology, Beijing University of Posts and Telecommunications, Beijing, China. Correspondence to: Kai Niu <niukai@bupt.edu.cn>.

*Proceedings of the 42nd International Conference on Machine Learning*, Vancouver, Canada. PMLR 267, 2025. Copyright 2025 by the author(s).

LPIPS (Zhang et al., 2018; Mentzer et al., 2020; Muckley et al., 2023) and DISTS (Ding et al., 2020). The diversity and inconsistency of these methods motivate us to establish a unified perspective for perceptual image compression with a well-established mathematical theoretical framework.

In this paper, we model the perceptual image compression problem mathematically based on a semantic information theory viewpoint. Recent advancements in semantic information theory (Niu & Zhang, 2024) highlight a set-element relationship between semantic information (i.e., the meaning) and syntactic information (i.e., data samples), where one meaning can be expressed in diverse syntactic forms. Building on this synonymity perspective, manipulating a set of samples with the same meaning (referred as to a **synonymous set**, abbreviated as "**Synset**") should be considered as the principle of semantic information processing. This viewpoint has the potential to surpass the theoretical limits of classical information theory while relaxing symbol-level accuracy (i.e., distortion) requirements. We emphasize that although the concept of synonymity originates from text data, it is universal to various types of natural data. For example, in image data, perceptual similarity between different images can be seen as a typical synonymous relationship.

On this basis, we re-analyze the optimization goal of perceptual image compression and introduce a novel variational inference method to analyze its optimization direction, aiming to guide the design of an image compression scheme. The contributions of our paper are as follows:

1. We propose *Synonymous Variational Inference* (SVI), a novel variational inference method to analyze the optimization direction of perceptual image compression. By building an ideal synset based on a typical criterion of perceptual similarity, it approximates the posterior of the corresponding latent synonymous representation with a parametric density by minimizing a partial semantic KL divergence. This method theoretically proves that the optimization direction of perceptual image compression is an expected rate-distortion-perception tradeoff form that covers the existing rate-distortion-perception schemes. To the best knowledge of the authors, our method is **the first work that can theoretically explain the fundamental reason for the divergence measure's existence in existing perceptual image compression schemes**.

2. We establish *Synonymous Image Compression* (SIC), a new image compression scheme that corresponds to the analytical process of SVI. By solely encoding the latent synonymous representation partially, SIC interprets this information as an equivalent quantized latent synset. It reconstructs multiple images satisfying the synonymous relationship with the original image by

multiple sampling the detailed representations independently from this latent synset.

3. We implement a progressive SIC codec to validate the theoretical analysis result, fully leveraging the model's capabilities. Experimental results demonstrate comparable rate-distortion-perception performance using a single neural progressive SIC image codec, thus verifying our method's effectiveness.

## 2. Background

### 2.1. Rate-Distortion Theory and Variational Inference

As one of the fundamental theorems in Shannon's classical information theory, rate-distortion theory (Shannon, 1948; Shannon et al., 1959) aims to address the lossy compression problem. It provides a theoretical lower bound of the compression rate $R(D)$ with a given distortion $D$, which can be characterized as a rate-distortion function (Thomas & Joy, 2006)

$$
\begin{aligned}
R(D) = \min_{p(\hat{x}|x)} \ & I\left(X; \hat{X}\right) \\
\text{s.t.} \quad & \mathbb{E}_{x,\hat{x} \sim p(x,\hat{x})}\left[d(x,\hat{x})\right] \leq D,
\end{aligned}
\tag{1}
$$

in which $I\left(X; \hat{X}\right)$ represents mutual information between the source $X$ and the reconstructed $\hat{X}$, numerically equal to the average coding rate for compressing $X$ with a given lossy codec; $D$ can be any reference distortion measure satisfying the condition that $d(x,\hat{x}) = 0$ if and only if $x = \hat{x}$, typified by the mean squared error (MSE).

To achieve this, learned image compression achieves optimal rate-distortion performance through end-to-end optimization training. While the ultimate optimization target remains the rate-distortion tradeoff, aligning with the continuous changes in neural network model training, the optimization process is achieved through variational inference for the generative model, specifically the variational auto-encoders (Kingma & Welling, 2013; Ballé et al., 2017; 2018). The core idea of variational inference is to build a parametric latent density $q(\tilde{\boldsymbol{y}}|\boldsymbol{x})$ and minimize the KL divergence, a standard measure in classical information theory, to approximate the true posterior $p_{\tilde{\boldsymbol{y}}|\boldsymbol{x}}(\tilde{\boldsymbol{y}}|\boldsymbol{x})$, i.e.,

$$
\mathbb{E}_{\boldsymbol{x}\sim p(\boldsymbol{x})} D_{\mathrm{KL}}\left[q||p_{\tilde{\boldsymbol{y}}|\boldsymbol{x}}\right] = \mathbb{E}_{\boldsymbol{x}\sim p(\boldsymbol{x})}\mathbb{E}_{\tilde{\boldsymbol{y}}\sim q}
$$
$$
\left[\underbrace{\overbrace{\log q(\tilde{\boldsymbol{y}}|\boldsymbol{x})}^{0} - \log p_{\boldsymbol{x}|\tilde{\boldsymbol{y}}}(\boldsymbol{x}|\tilde{\boldsymbol{y}})}_{\text{weighted distortion}} \underbrace{- \log p_{\tilde{\boldsymbol{y}}}(\tilde{\boldsymbol{y}})}_{\text{rate}}\right]
\tag{2}
$$
$$
+ \text{ const.}
$$

As the first term equals 0 under the assumption of a uniform density on the unit interval centered on $\boldsymbol{y}$, and the last term is a constant, the optimization simplifies to the sum of a weighted distortion and a coding rate, thereby achieving the optimal rate-distortion tradeoff.

## 2.2. The Rate-Distortion-Perception tradeoff

Since Blau and Michaeli demonstrated the apparent trade-off between perceptual quality and distortion measure that widely exists in various distortion measures (2018), they extended the classic rate-distortion tradeoff to a triple tradeoff version (2019). Specifically, they define the perceptual quality index $d_p\left(p_{\boldsymbol{x}}, p_{\hat{\boldsymbol{x}}}\right)$ based on some divergence between distributions of the source and reconstructed images, and build a new lower bound of compression rate $R\left(D, P\right)$ with considerations of the perception index, i.e.,

$$
\begin{aligned}
R\left(D, P\right) = \min_{p(\hat{\boldsymbol{x}}|\boldsymbol{x})} \ & I\left(\boldsymbol{X}; \hat{\boldsymbol{X}}\right) \\
\text{s.t.} \quad & \mathbb{E}_{\boldsymbol{x}, \hat{\boldsymbol{x}} \sim p(\boldsymbol{x}, \hat{\boldsymbol{x}})}\left[d\left(\boldsymbol{x}, \hat{\boldsymbol{x}}\right)\right] \leq D, \\
& d_p\left(p_{\boldsymbol{x}}, p_{\hat{\boldsymbol{x}}}\right) \leq P.
\end{aligned} \tag{3}
$$

Building on this triple tradeoff relationship, the perceptual image compression methods (Agustsson et al., 2019; Mentzer et al., 2020; He et al., 2022b; Agustsson et al., 2023; Muckley et al., 2023) typically optimize the model using the following loss function form:

$$
\begin{aligned}
\mathcal{L}_{RDP} = & \lambda_r \cdot I\left(\boldsymbol{X}; \hat{\boldsymbol{X}}\right) + \lambda_d \cdot \mathbb{E}_{\boldsymbol{x}, \hat{\boldsymbol{x}} \sim p(\boldsymbol{x}, \hat{\boldsymbol{x}})}\left[d\left(\boldsymbol{x}, \hat{\boldsymbol{x}}\right)\right] \\
& + \lambda_p \cdot d_p\left(p_{\boldsymbol{x}}, p_{\hat{\boldsymbol{x}}}\right).
\end{aligned} \tag{4}
$$

## 2.3. Semantic Information Theory

As the optimization towards perceptual quality is more inclined to the accuracy of conveying meaning (the semantic problem) instead of the symbol-level accuracy that classical information theory focuses on (the technical problem) (Weaver, 1953), this paper considers analyzing the problem of perceptual image compression based on semantic information theory.

Research on semantic information theory has been ongoing since the 1950s, with various viewpoints such as logical probability (Carnap & Bar-Hillel, 1952; Bar-Hillel & Carnap, 1953; Barwise & Perry, 1981; Floridi, 2004; Bao et al., 2011) and fuzzy probability (De Luca & Termini, 1972; 1974; Al-Sharhan et al., 2001) employed to discuss the essence and measures of semantic information, but no consensus has been reached over time. Furthermore, these viewpoints provide limited theoretical guidance for the practical coding of natural information sources.

However, a recent contribution to semantic information theory (Niu & Zhang, 2024) presents a potential turning point in the field, which suggests understanding the semantic information from a synonymity perspective. In this theory, semantic information is processed based on a fundamental principle: considering a set of syntactic samples with the same meaning (referred to as a synset). Corresponding semantic information measures are also provided. As an im-

portant foundation, **a semantic variable $\mathring{U}$ [1] corresponds to various possible synsets $\mathcal{U}_{i_s} = \{u_i \mid i \in \mathcal{N}_{i_s}\}$,** where each sample $u_i$ is a possible value of the syntactic variable $U$ and shares the same semantic meaning with all the possible values $\{u_j \mid j \in \mathcal{N}_{i_s}\}$ indexed in $\mathcal{N}_{i_s}$. On this basis, the semantic entropy of $\mathring{U}$ is defined by

$$
H_s\left(\mathring{U}\right) = -\sum_{i_s} \sum_{i \in \mathcal{N}_{i_s}} p\left(u_i\right) \log \left(\sum_{i \in \mathcal{N}_{i_s}} p\left(u_i\right)\right), \tag{5}
$$

in which the probability of the synset $p\left(\mathcal{U}_{i_s}\right)$ is defined as the sum of the probabilities of all the samples $p\left(u_i\right)$ within it. This directly leads to the inequality between the semantic entropy and the classical Shannon entropy, i.e., $H_s\left(\mathring{U}\right) \leq H\left(U\right)$, being apparently valid, since the uncertainty of syntactic samples is no longer the focus.

As the foundation of the synonymous variational inference proposed in this paper, a new form of KL divergence needs to be introduced from (Niu & Zhang, 2024), referred to as *partial semantic KL divergence* $D_{\mathrm{KL},s}\left[q || p_s\right]$, which is defined as

$$
D_{\mathrm{KL},s}\left[q || p_s\right] = \sum_{i_s} \sum_{i \in \mathcal{N}_{i_s}} q\left(u_i\right) \log \frac{q\left(u_i\right)}{p\left(\mathcal{U}_{i_s}\right)}, \tag{6}
$$

which represents a divergence between a syntactic distribution $q$ and a semantic distribution $p_s$. [2] Clearly, these two distributions emphasize different levels of information, i.e., the syntactic level and the semantic level. However, examining the distance between these distributions holds significant physical meaning in perceptual image compression, which will be detailed in Section 3.2.

# 3. Synonymous Variational Inference: A Semantic Information Viewpoint

## 3.1. Overview

Consider an image codec, in which the encoder captures the semantic information of the image, while the decoder reconstructs an image with the same semantics as the original instead of directly restoring the original image's pixels. Obviously, in natural image data, there are typically multiple images that share the same semantic information as the original image. For example, all images that exhibit certain perceptual similarities to the original image can be considered to convey the same meaning "to some extent"

---

[1]Refer to as $\tilde{U}$ in Niu and Zhang's paper (2024). The ring hat symbol "°" is appplied to distinguish it from the tilde hat symbol "~" commonly used in variational inference.

[2]The relationship between the partial semantic KL divergence and the standard KL divergence satisfies $D_{\mathrm{KL},s}\left[q || p_s\right] \leq D_{\mathrm{KL}}\left[q || p\right]$, which can be referred to as *partial semantic relative entropy* in Niu and Zhang's paper (2024).

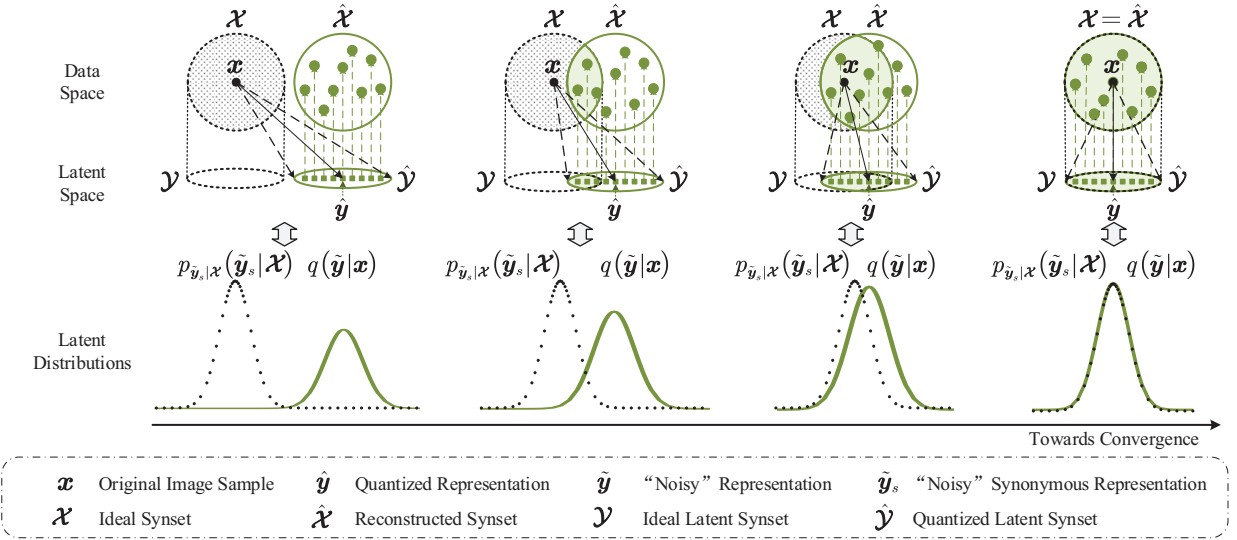

*Figure 1.* An illustration of the optimization directions of synonymous image compression. By continuously minimizing the partial semantic KL divergence $D_{\mathrm{KL},s}\left[q||p_{\tilde{\boldsymbol{y}}_s|\boldsymbol{\mathcal{X}}}\right]$ in latent space, the reconstructed synset $\hat{\boldsymbol{\mathcal{X}}}$ gradually approaches the ideal synset $\boldsymbol{\mathcal{X}}$ until complete overlap occurs. At that point, every sample $\hat{\boldsymbol{x}}_j \in \hat{\boldsymbol{\mathcal{X}}}$ is a "synonym" of the original image sample $\boldsymbol{x}$.

while retaining distinct detailed information.[3] Therefore, when placing these images in a synset, there must be a latent representation and the corresponding coding sequence capable of capturing the shared characteristics among all samples in this set, which should be learned by the semantic encoder. Using this representation, the semantic decoder can sample any image from the synset of the original image, which satisfies certain perceptual similarity with the source image if the synonymous judge criterion is the perceptual similarity criterion.

When considering using a deep neural network to design the above codec, its continuous optimization problem can be analyzed using a similar idea to the variational autoencoder, based on variational inference (Kingma & Welling, 2013). Figure 1 illustrates the optimization direction of this problem and provides a schematic representation of the achievable effects within the data space. Ideally, the source image $\boldsymbol{x}$ can be considered as a sample in the ideal synset $\boldsymbol{\mathcal{X}}$, shown as the dashed circles in Figure 1. Correspondingly, there must be an ideal posterior $p_{\tilde{\boldsymbol{y}}_s|\boldsymbol{\mathcal{X}}}(\tilde{\boldsymbol{y}}_s|\boldsymbol{\mathcal{X}})$ in the latent space to represent the latent synonymous representations $\boldsymbol{y}_s$ that capture the shared characteristics of all the samples $\{\boldsymbol{x}_i|\boldsymbol{x}_i \in \boldsymbol{\mathcal{X}}\}$. However, in practice, the ideal synset $\boldsymbol{\mathcal{X}}$ is unavailable, so we rely on the original image $\boldsymbol{x}$ to construct a parametric latent density $q(\tilde{\boldsymbol{y}}|\boldsymbol{x})$ to approximate this pos-

terior by minimizing the partial semantic KL divergence between these two distributions, i.e.,

$$\min \mathbb{E}_{\boldsymbol{x} \sim p(\boldsymbol{x})} D_{\mathrm{KL},s}\left[q||p_{\tilde{\boldsymbol{y}}_s|\boldsymbol{\mathcal{X}}}\right]. \tag{7}$$

Once this distribution divergence is minimized, a generative model $p_{\boldsymbol{x}|\tilde{\boldsymbol{y}}_s,\hat{\boldsymbol{y}}_\epsilon}(\boldsymbol{x}|\tilde{\boldsymbol{y}}_s,\hat{\boldsymbol{y}}_\epsilon)$ (i.e., the semantic decoder, in which $\hat{\boldsymbol{y}}_\epsilon$ is a sampling for details) can be finally optimized. On this basis, the reconstructed synset $\hat{\boldsymbol{x}}$ produced by the semantic decoder can be considered as a sample of the ideal synset $\boldsymbol{\mathcal{X}}$, which ensures exhibiting certain perceptual similarities to the original image $\boldsymbol{x}$.

### 3.2. Synonymous Variational Inference

Unlike the usual variational inference, the two distributions of (7) work at different levels: The parametric density $q(\tilde{\boldsymbol{y}}|\boldsymbol{x})$ works at the syntactic level, while the true posterior $p_{\tilde{\boldsymbol{y}}_s|\boldsymbol{\mathcal{X}}}(\tilde{\boldsymbol{y}}_s|\boldsymbol{\mathcal{X}})$ operates at the semantic level, represented in the form of synsets. However, since $\tilde{\boldsymbol{y}}$ can be decomposed into a combination or concatenation of a synonymous representation $\tilde{\boldsymbol{y}}_s$ and a detailed representation $\tilde{\boldsymbol{y}}_\epsilon$, it is possible to effectively process the minimization of the partial semantic KL divergence. To distinguish from the existing variational inference methods, we give the following definition.

**Definition 3.1.** *Synonymous Variational Inference* (SVI) is a generic variational inference method that approximates the true posterior of the synonymous representations with a parametric density at the syntactic level by minimizing the partial semantic KL divergence (7).

By applying the proposed synonymous variational inference,

---

[3]In practice, people may have varying judgments about whether two images have the same meaning, due to their varying judge criteria. Thus, samples in a synset based on a given synonymous criterion must share some specific semantic information but do not necessarily have completely identical meanings.

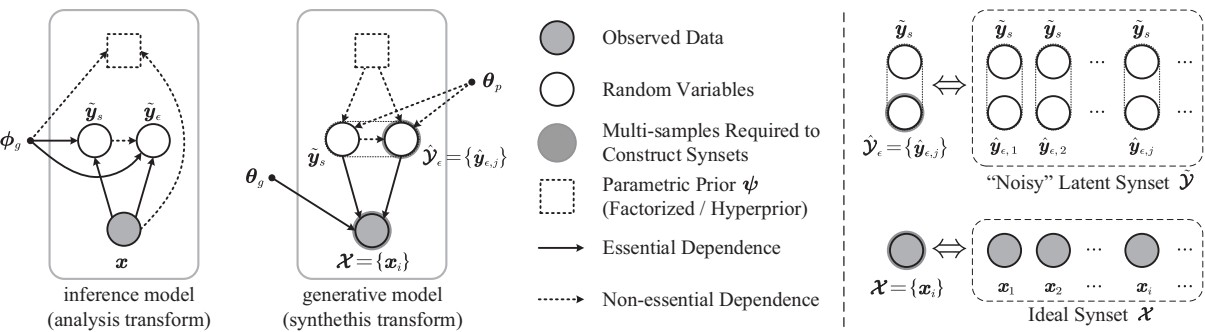

*Figure 2.* Left: Representation of the proposed encoder as a synonymous variational inference model, and corresponding decoder as a generative Bayesian model. The latent representation $\tilde{\boldsymbol{y}}$ is a merge of the synonymous representation $\tilde{\boldsymbol{y}}_s$ and the detailed representation $\tilde{\boldsymbol{y}}_\epsilon$, achieved through some form of merging or splicing. A fully factorized (Ballé et al., 2017) or a hyperprior-like (Ballé et al., 2018; Minnen et al., 2018) entropy model can be employed in the "Parametric Prior" item. An autoregressive (Minnen et al., 2018) or a parallel (He et al., 2021) context model can also be utilized in $\boldsymbol{\theta}_p$. These two types of methods can be used for accurate probability estimations of $\tilde{\boldsymbol{y}}_s$ or predictions for $\hat{\boldsymbol{y}}_\epsilon$. Right: Illustrations for the equivalent relationship of the "noisy" latent synset $\tilde{\mathcal{Y}}$ and the ideal synset $\boldsymbol{\mathcal{X}}$.

the optimization direction of perceptual image compression can be determined. To facilitate the subsequent derivation, we first state the following lemma (the detailed proof can be found in Appendix A.1):

**Lemma 3.2.** *When the source considers the existence of an ideal synset $\boldsymbol{\mathcal{X}}$ and the decoder places the reconstructed sample in a reconstructed synset $\tilde{\hat{\boldsymbol{\mathcal{X}}}}$, the minimization of the expected negative log synonymous likelihood term*

$$\min \mathbb{E}_{\boldsymbol{x} \sim p(\boldsymbol{x})} \mathbb{E}_{\tilde{\boldsymbol{y}} \sim q} \left[ -\log p_{\boldsymbol{\mathcal{X}} | \tilde{\boldsymbol{y}}_s} \left( \boldsymbol{\mathcal{X}} | \tilde{\boldsymbol{y}}_s \right) \right]$$
$$\Longleftrightarrow \min \lambda_d \cdot \mathbb{E}_{\boldsymbol{x} \sim p(\boldsymbol{x})} \mathbb{E}_{\tilde{\boldsymbol{y}} \sim q} \mathbb{E}_{\tilde{\boldsymbol{x}}_i \in \tilde{\boldsymbol{\mathcal{X}}} | \tilde{\boldsymbol{y}}_s} \left[ d \left( \boldsymbol{x}, \tilde{\boldsymbol{x}}_i \right) \right] \quad (8)$$
$$+ \lambda_p \cdot \mathbb{E}_{\tilde{\boldsymbol{y}} \sim q} \mathbb{E}_{\tilde{\boldsymbol{x}}_i \in \tilde{\boldsymbol{\mathcal{X}}} | \tilde{\boldsymbol{y}}_s} D_{KL} \left[ p_{\boldsymbol{x}} || p_{\tilde{\boldsymbol{x}}_i} \right],$$

*in which $\lambda_d$ and $\lambda_p$ are the tradeoff factors for the expected distortion (typically expected means-squared error, i.e., E-MSE) term and the expected KL divergence (E-KLD) term, respectively.*

Based on this, we propose the following theorem:

**Theorem 3.3.** *For an image source $\boldsymbol{x} \sim p(\boldsymbol{x})$ together with its bounded expected distortion $\mathbb{E}_{\boldsymbol{x} \sim p(\boldsymbol{x})} \mathbb{E}_{\hat{\boldsymbol{x}}_i \in \hat{\boldsymbol{\mathcal{X}}} | \hat{\boldsymbol{y}}_s} \left[ d \left( \boldsymbol{x}, \hat{\boldsymbol{x}}_i \right) \right]$ and expected KL divergence $\mathbb{E}_{\hat{\boldsymbol{x}}_i \in \hat{\boldsymbol{\mathcal{X}}} | \hat{\boldsymbol{y}}_s} D_{KL} \left[ p_{\boldsymbol{x}} || p_{\hat{\boldsymbol{x}}_i} \right]$, the minimum achievable rate of perceptual image compression is*

$$R(\boldsymbol{\mathcal{X}}) = \min_{p(\hat{\boldsymbol{\mathcal{X}}} | \boldsymbol{x})} I \left( \boldsymbol{X}; \tilde{\hat{\boldsymbol{X}}} \right)$$
$$\text{s.t.} \quad \mathbb{E}_{\boldsymbol{x} \sim p(\boldsymbol{x})} \mathbb{E}_{\hat{\boldsymbol{x}}_i \in \hat{\boldsymbol{\mathcal{X}}} | \hat{\boldsymbol{y}}_s} \left[ d \left( \boldsymbol{x}, \hat{\boldsymbol{x}}_i \right) \right] \leq D, \quad (9)$$
$$\mathbb{E}_{\hat{\boldsymbol{x}}_i \in \hat{\boldsymbol{\mathcal{X}}} | \hat{\boldsymbol{y}}_s} D_{KL} \left[ p_{\boldsymbol{x}} || p_{\hat{\boldsymbol{x}}_i} \right] \leq P,$$

*where $I \left( \boldsymbol{X}; \tilde{\hat{\boldsymbol{X}}} \right) = H_s \left( \tilde{\hat{\boldsymbol{X}}} \right) - H_s \left( \tilde{\hat{\boldsymbol{X}}} | \boldsymbol{X} \right)$ with semantic variable $\tilde{\hat{\boldsymbol{X}}}$ corresponds to the reconstructed synset $\hat{\boldsymbol{\mathcal{X}}}$.*

*Proof.* As stated in Figure 2, the model of synonymous image compression can be considered as a generalized variational auto-encoder. By using the proposed SVI, i.e., minimizing the partial semantic KL divergence given in (7),

$$\mathbb{E}_{\boldsymbol{x} \sim p(\boldsymbol{x})} D_{\text{KL},s} \left[ q || p_{\tilde{\boldsymbol{y}}_s | \boldsymbol{\mathcal{X}}} \right] = \mathbb{E}_{\boldsymbol{x} \sim p(\boldsymbol{x})} \mathbb{E}_{\tilde{\boldsymbol{y}} \sim q}$$
$$\left[ \underbrace{\log q \left( \tilde{\boldsymbol{y}} | \boldsymbol{x} \right)}_{0} - \log p_{\boldsymbol{\mathcal{X}} | \tilde{\boldsymbol{y}}_s} \left( \boldsymbol{\mathcal{X}} | \tilde{\boldsymbol{y}}_s \right) - \log p_{\tilde{\boldsymbol{y}}_s} \left( \tilde{\boldsymbol{y}}_s \right) \right] \quad (10)$$
$$+ \text{const.}$$

The first term equals 0 under the assumption of a uniform density on the unit interval centered on $\boldsymbol{y}$, and the last term is a constant for a determined $\boldsymbol{x}$ and corresponding ideal synset $\boldsymbol{\mathcal{X}}$. For the third term, with a determined inference and generative model, the coding rate of the synonymous representation $\mathbb{E}_{\boldsymbol{x} \sim p(\boldsymbol{x})} \mathbb{E}_{\tilde{\boldsymbol{y}} \sim q} \left[ -\log p_{\tilde{\boldsymbol{y}}_s} \left( \tilde{\boldsymbol{y}}_s \right) \right]$ is equal to $I \left( \boldsymbol{X}; \tilde{\hat{\boldsymbol{X}}} \right)$, as stated in Appendix A.2.

By Lemma 3.2, the minimization of the second term is equivalent to minimizing a weighted expected distortion $\mathbb{E}_{\boldsymbol{x} \sim p(\boldsymbol{x})} \mathbb{E}_{\tilde{\boldsymbol{y}} \sim q} \mathbb{E}_{\tilde{\boldsymbol{x}}_i \in \tilde{\boldsymbol{\mathcal{X}}} | \tilde{\boldsymbol{y}}_s} \left[ d \left( \boldsymbol{x}, \tilde{\boldsymbol{x}}_i \right) \right]$ plus a weighted E-KLD term $\mathbb{E}_{\tilde{\boldsymbol{y}} \sim q} \mathbb{E}_{\tilde{\boldsymbol{x}}_i \in \tilde{\boldsymbol{\mathcal{X}}} | \tilde{\boldsymbol{y}}_s} D_{\text{KL}} \left[ p_{\boldsymbol{x}} || p_{\tilde{\boldsymbol{x}}_i} \right]$. These weights can be considered as Lagrange multipliers to the rate term, which makes the optimization goal equivalent to minimizing $I \left( \boldsymbol{X}; \tilde{\hat{\boldsymbol{X}}} \right)$ with quantized bounded expected distortion and E-KLD constraints to obtain the optimal $p(\hat{\boldsymbol{\mathcal{X}}} | \boldsymbol{x})$, shown as (9). This target corresponds to a ***Synonymous Rate-Distortion-Perception Tradeoff***, which can be shown as

$$\mathcal{L}_{\boldsymbol{\mathcal{X}}} = \lambda_r \cdot \mathbb{E}_{\boldsymbol{x} \sim p(\boldsymbol{x})} \left[ -\log p_{\hat{\boldsymbol{y}}_s} \left( \hat{\boldsymbol{y}}_s \right) \right]$$
$$+ \lambda_d \cdot \mathbb{E}_{\boldsymbol{x} \sim p(\boldsymbol{x})} \mathbb{E}_{\hat{\boldsymbol{x}}_i \in \hat{\boldsymbol{\mathcal{X}}} | \hat{\boldsymbol{y}}_s} \left[ d \left( \boldsymbol{x}, \hat{\boldsymbol{x}}_i \right) \right] \quad (11)$$
$$+ \lambda_p \cdot \mathbb{E}_{\hat{\boldsymbol{x}}_i \in \hat{\boldsymbol{\mathcal{X}}} | \hat{\boldsymbol{y}}_s} D_{\text{KL}} \left[ p_{\boldsymbol{x}} || p_{\hat{\boldsymbol{x}}_i} \right],$$

thus we finish the proof of the theorem. $\square$

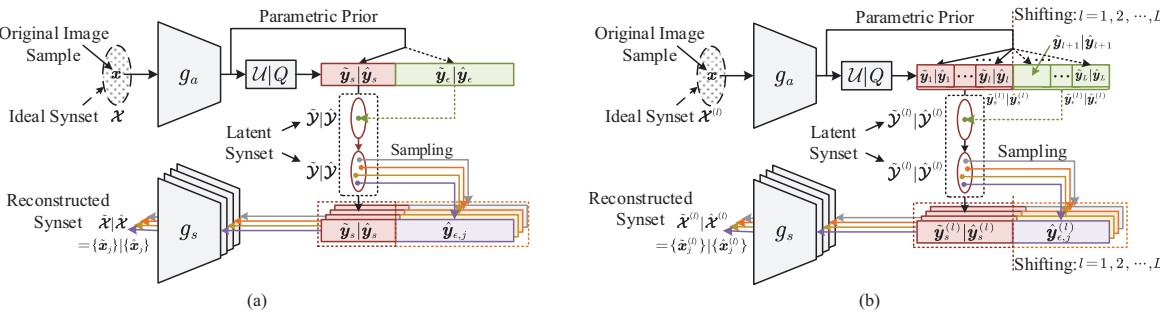

*Figure 3.* Processing frameworks of SIC. (a): The general framework. (b): The progressive framework.

Due to space limitations, only a brief outline of the proof is provided above. Please refer to Appendix A.2 for detailed proof. Additionally, we state that the existing rate-distortion-perception tradeoff (4) is a special case of (11) when there is only one sample $\hat{x}$ in the reconstructed synset $\hat{\mathcal{X}}$. More comprehensive discussions are provided in Appendix A.3.

## 4. Synonymous Image Compression

According to Figure 2, the main difference between our proposed processing scheme and the general LIC method is that the generator only needs part of the accurate latent features, while the other part can be obtained through random sampling instead of accurate coding. Thus, the generator can obtain all the information required to reconstruct the image. To this end, we name this new image compression scheme *Synonymous Image Compression* (SIC).

A general framework of SIC is given in Figure 3(a). The SIC codec requires an analysis transform $g_a\left(\boldsymbol{x}; \boldsymbol{\phi}_g\right)$ and a synthesis transform $g_s\left(\hat{\boldsymbol{y}}_s, \hat{\boldsymbol{y}}_{\epsilon,j}; \boldsymbol{\theta}_g\right)$ to achieve the bidirectional nonlinear mapping between the data space and the latent space. Only the synonymous representation $\hat{\boldsymbol{y}}_s$ is required to be coded, and this representation is equivalent to a quantized latent synset $\hat{\mathcal{Y}}$ contains all the samples with different detailed representations. With obtaining $\hat{\boldsymbol{y}}_s$, the generative model can generate multiple images $\hat{\mathcal{X}} = \{\hat{\boldsymbol{x}}_j\}$ by sampling diverse detailed representations $\left\{\hat{\boldsymbol{y}}_{\epsilon,j}\right\}$ in $\hat{\mathcal{Y}}$.

A uniform noise with unit interval should be used for $\boldsymbol{y}_s$ to achieve efficient continuous optimization, and an entropy model assisted by an arbitrary form of parametric prior should be employed to estimate the coding rate for $\hat{\boldsymbol{y}}_s$, whereas both these two are not essential for $\boldsymbol{y}_\epsilon$ and $\hat{\boldsymbol{y}}_\epsilon$. Although this aligns with the analytical process of synonymous variational inference, it results in the SIC model's capabilities being underutilized, as the details still have the potential to provide additional information.

The progressive framework of SIC is proposed to solve this problem, as shown in Figure 3(b). It partition the latent fea-

ture $\hat{\boldsymbol{y}}$ into $L$ synonymous levels, treating the first $l$ levels as the synonymous representation $\hat{\boldsymbol{y}}_s^{(l)}$, while the subsequent levels are considered as the detailed representation $\hat{\boldsymbol{y}}_\epsilon^{(l)}$. By varying $l$ from 1 to $L$ and optimizing through the loss function of the corresponding level in the training process, the SIC model can be optimized for approaching different ideal synsets $\mathcal{X}^{(l)}$. After training, synonymous representations at each level can be encoded progressively and fed to the decoder, making SIC a progressive image codec that can produce images at diverse synonymous levels (corresponding to varying coding rates) using a single generator. In our upcoming experiments, we implement a progressive SIC model to fully leverage the model's capabilities, where the $L$ levels are equally slicing the channels $C$ into $L$ groups, shown as Figure 7 in Appendix C.1.

The designing of the training loss for the progressive SIC model should take several practical issues into account. Firstly, while the loss function requires traversing samples in the reconstructed synset $\hat{\mathcal{X}}$, this is not practical during the actual training process. Therefore, the real training loss function utilizes a small number of reconstructed samples and computes the arithmetic mean as a practical approximation. Secondly, the training should also take into account the synonymous rate-distortion-perception trade-offs at each level, meaning that the trade-offs between levels are also required to be balanced. To this end, we design a group of loss functions for the progressive SIC model that alternatively trains for the level $l = 1, 2, \cdots, L$ step by step, i.e.,

$$\mathcal{L}^{(l)} = \alpha\mathcal{L}_{\mathcal{X}}^{(l)} + (1 - \alpha)\mathcal{L}_{\mathcal{X}}^{(L)} + \mathcal{L}_c^{(l)}, l = 1, 2, \cdots, L, \quad (12)$$

in which $\mathcal{L}_{\mathcal{X}}^{(l)}$ is represented by

$$\mathcal{L}_{\mathcal{X}}^{(l)} = \mathbb{E}_{\boldsymbol{x}\sim p(\boldsymbol{x})}\Big[-\lambda_r^{(l)} \cdot \log p_{\hat{\boldsymbol{y}}_s^{(l)}}\left(\hat{\boldsymbol{y}}_s^{(l)}\right) +$$
$$\frac{1}{M}\sum_{i=1}^{M}\left(\lambda_d^{(l)} \cdot \text{MSE}\left(\boldsymbol{x}, \hat{\boldsymbol{x}}_i^{(l)}\right) + \lambda_p^{(l)} \cdot \text{LPIPS}\left(\boldsymbol{x}, \hat{\boldsymbol{x}}_i^{(l)}\right)\right)\Big], \quad (13)$$

where the MSE is utilized as the distortion item, and the LPIPS (Zhang et al., 2018) directly replace by the KL diver-

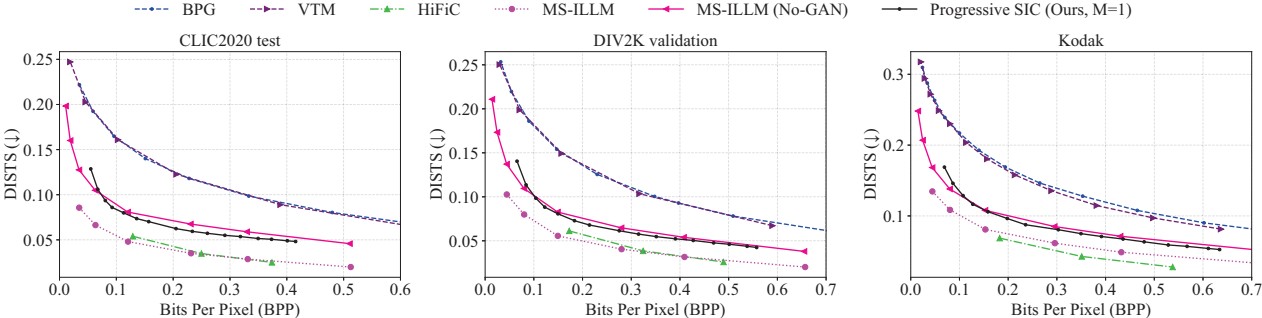

Figure 4. Comparisons of methods using DISTS on different datasets. Each point on the HiFiC and MS-ILLM performance curves is from a single model, while our entire performance curves are achieved by a single progressive SIC model.

gence term since accurately calculating the KL divergence for image datasets is challenging; $\alpha$ is set between the $l$ level and the $L$ level (equivalent to the conventional rate-distortion-perception tradeoff) to indirectly achieve multi-level tradeoffs; $L_c^{(l)}$ is additional constraints in training, which detailed elaborated in Appendix C.1. Moreover, $M$ denotes the number of reconstructed samples in training. Subsequent experimental results will show that with $M = 1$ the proposed method can achieve comparable rate-distortion-perception performance, while bigger $M$ will perform advantages in certain synonymous levels.

## 5. Experimental Illustration

In this section, we examine the effectiveness of our proposed analytical theory through experimental results.

### 5.1. Implementation Setup and Comparison Schemes

**Model architecture:** The analysis transform $g_a$ and the synthesis transform $g_s$ are implemented using the Swin Transformer (Liu et al., 2021), while the coding rate is estimated using a joint autoregressive and hierarchical prior architecture (Minnen et al., 2018) based on deep convolutional neural networks, with structural adjustments made for our level partitioning mechanism. We set the number of latent representation channels to $C = 512$ and the number of the equally partitioned synonymous levels to $L = 16$, giving each synonymous level a channel dimension of 32. This allows a single progressive SIC codec to support 16 coding rates and their corresponding image quality levels.

**Model Training:** We randomly select 100,000 images from the OpenImages V6 dataset (Kuznetsova et al., 2020) as the training data, resizing them to a uniform resolution of 256×256 using random crop and resized crop for the training process. We train our model for $1 \times 10^6$ iterations with a batch size of 16, a learning rate of $1 \times 10^{-4}$, and the AdamW optimizer with a weight decay of $5 \times 10^{-5}$. We evaluate

our models with the test set of CLIC2020 (Toderici et al., 2020), the validation set of DIV2K (Agustsson & Timofte, 2017), and the Kodak dataset [4]. Refer to Appendix C.1 for the hyperparameter settings and implementation details.

**Comparison schemes:** The comparison schemes for traditional image compression use BPG (Bellard, 2015) and the state-of-the-art VTM [5] to serve as the benchmarks for distortion measures. The PIC schemes for comparison include HiFiC (Mentzer et al., 2020) and MS-ILLM (Muckley et al., 2023), including the No-GAN fine-tuning version of MS-ILLM. These two schemes use adversarial loss to optimize perceptual quality, serving as benchmarks for perceptual measures. As the pre-training model of MS-ILLM, the No-GAN scheme directly uses LPIPS, the same perception measure as our choice, which is the focus of comparison.

**Evaluation Metrics:** PSNR is utilized for distortion measuring, while LPIPS and DISTS (Ding et al., 2020) are for perception measuring. It should be noted that the DISTS measure, due to its resampling tolerance, aligns more closely with the human understanding of perceptual similarity-typified synonymous relationships than LPIPS, thus as our focus in our following analysis.

### 5.2. Performance Analysis

We first examine our progressive SIC model's capabilities in the adaptation of full-rate perception qualities. Figure 4 compares the performance of our method and the comparison schemes under DISTS, in which our model is trained with sampling $\hat{\boldsymbol{y}}_{\epsilon,j}$ only once (i.e., $M = 1$), as this follows the common usage in existing generative model training. As shown in this figure, our scheme can achieve perceptual quality adaptability across various rates using a single model, with the perceptual quality of the reconstructed im-

---

[4]Kodak PhotoCD dataset, URL http://r0k.us/graphics/kodak/.

[5]VVCSoftware_VTM, URL https://vcgit.hhi.fraunhofer.de/jvet/VVCSoftware_VTM.git, Version 23.4.

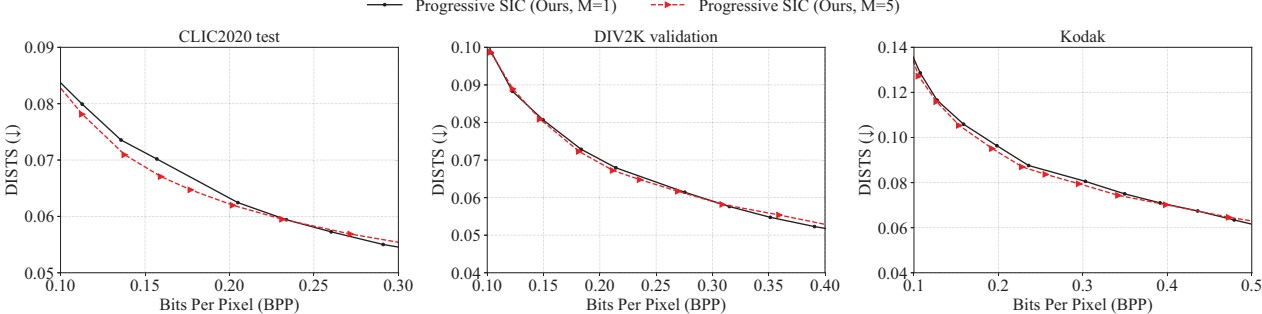

*Figure 5.* Comparisons of our progressive SIC schemes with different sampling numbers in reconstructed $\hat{\mathcal{X}}$ on different datasets.

age improving as the coding rate increases.

For the concerned DISTS measure, our method surpasses the No-GAN MS-ILLM solution (also trained with LPIPS) in a large coding rate range. This performance is demonstrated under conditions where the PSNR quality continuously approaches and even exceeds the comparison No-GAN schemes, and the LPIPS quality remains very similar, thus verifying a comparable rate-distortion-perception performance, shown as Figure 11 in Appendix C.2. This surpassing reflects the advantage of incorporating the concept of synset in our proposed SVI. However, this advantage is modest compared to HiFiC and MS-ILLM guided by the GAN's adversarial loss, since our loss directly uses LPIPS to replace the KL divergence. Therefore, utilizing discriminative mechanisms in SIC models with the synonymous viewpoint (i.e., replacing the E-KLD term) may further enhance perceptual quality in future considerations.

In addition, Figure 5 compares the DISTS qualities for reconstruction sampling numbers $M = 1$ and $M = 5$ under identical hyperparameter settings. With $M = 5$, the expected distortion and perception loss are estimated via arithmetic mean, aligning more closely with theoretical optimization directions. Results indicate that expanding the reconstructed synset $\hat{\mathcal{X}}$ by increasing $\hat{\boldsymbol{y}}_{\epsilon,j}$ samples during training offers slight performance advantages across various datasets in general. These advantages are especially evident in the low and medium rate range, while the intersections at relatively high rates are due to insufficient fine-tuning of the hyperparameter settings, which needs more precise exploration in future works.

For further additional analysis and visualization results, please refer to Appendix C.2.

## 6. Limitations

According to the above experimental results and analysis, the limitation of the implemented progressive SIC model mainly lies in **using the LPIPS measure to replace the**

**divergence term in the loss function**. Based on related PIC work, the **GAN-based adversarial loss** is suggested as a replacement for the KL divergence term to further improve model performance.

To verify the impact of adversarial loss on performance, we build a discriminator based on HiFiC's structure and fine-tune the synthesis transform $g_s$ of our $M = 1$ and $M = 5$ models with non-saturating loss (Goodfellow et al., 2014) for $2 \times 10^5$ steps as supplementary experiments. Specifically, we use a single conditional discriminator (refer to Appendix D.1) to obtain the discriminative loss for reconstructed images at all synonymous levels, with the synonymous representation $\hat{\boldsymbol{y}}_s^{(l)}$ of each synonymous level $l$ introduced as a separate condition. The non-saturating loss can be expressed as

$$\mathcal{L}_G^{(l)} = \mathbb{E}_{\boldsymbol{x},\hat{\boldsymbol{x}}_i^{(l)}} \left[ -\log \left( D \left( \hat{\boldsymbol{x}}_i^{(l)}, \hat{\boldsymbol{y}}_s^{(l)} \right) \right) \right], \quad (14)$$

$$\begin{aligned} \mathcal{L}_D^{(l)} = &\mathbb{E}_{\boldsymbol{x},\hat{\boldsymbol{x}}_i^{(l)}} \left[ -\log \left( 1 - D \left( \hat{\boldsymbol{x}}_i^{(l)}, \hat{\boldsymbol{y}}_s^{(l)} \right) \right) \right] \\ &+ \mathbb{E}_{\boldsymbol{x} \sim p(\boldsymbol{x})} \left[ -\log \left( D \left( \boldsymbol{x}, \hat{\boldsymbol{y}}_s^{(l)} \right) \right) \right], \end{aligned} \quad (15)$$

and thus the perception term in Equation (13) is required to be adjusted from solely the LPIPS term to the sum of the LPIPS term and the generative loss in Equation (14), i.e.,

$$\text{LPIPS}\left(\boldsymbol{x}, \hat{\boldsymbol{x}}_i^{(l)}\right) \Rightarrow \text{LPIPS}\left(\boldsymbol{x}, \hat{\boldsymbol{x}}_i^{(l)}\right) + \mathcal{L}_G^{(l)}. \quad (16)$$

Figure 6 shows the fine-tuned performance using DISTS as the evaluation measure, while other measures, including PSNR, LPIPS, and FID (Heusel et al., 2017), and visualization results are given in Appendix D.2. These results show that:

- **The perceptual quality (DISTS, FID) has improved**. While LPIPS performance remains nearly unchanged, DISTS and FID show greater gains at higher bitrates, with DISTS gradually approaching that of MS-ILLM

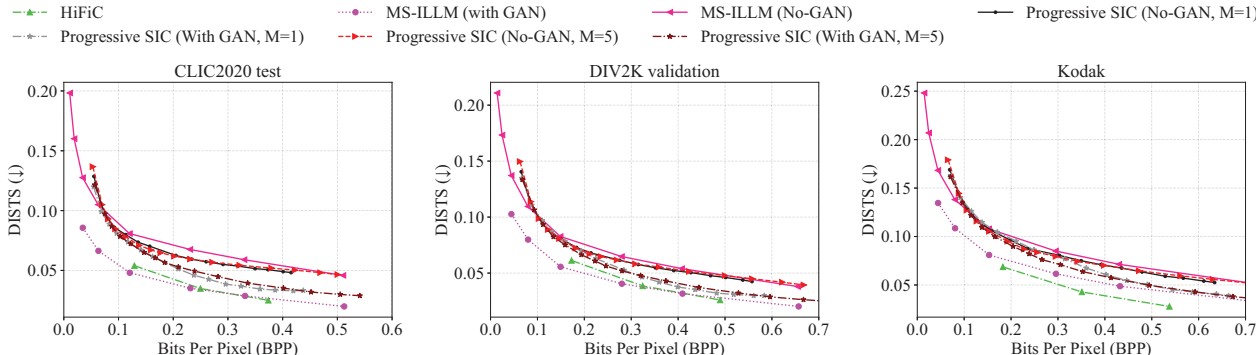

*Figure 6.* Comparisons of methods using DISTS on different datasets (supplemented fine-tuned model performance). Each point on the HiFiC and MS-ILLM performance curves is from a single model, while our entire performance curves are achieved by a single progressive SIC model.

(with GAN). This means that the fine-tuning loss with the adversarial loss is closer to the ideal optimization direction, which can make better perceptual qualities of the reconstructed images. Besides, while DISTS and FID values change little at low rates (bpp $< 0.10$), the visual qualities improve significantly.

- **The improvement of DISTS is relatively obvious, while the enhancement in FID remains limited**. This is a noteworthy phenomenon since **it is absent in the non-sampling schemes**, as confirmed by the experimental results in the MS-ILLM paper (Muckley et al., 2023). Unlike DISTS' resampling tolerance, FID focuses on the consistency of the distribution between the original and reconstructed image groups. This suggests that our implementation is still insufficient in optimizing the distribution of reconstructed images, especially in our detailed sampling mechanism.

- **The gap compared to HiFiC and MS-ILLM is still obvious, especially on FID**. In addition to the reason of the detailed sampling mechanism, this issue may also be due to insufficient fine-tuning with multiple synonymous layers battling against each other, or inefficient perceptual loss is chosen. Besides, to address the limited improvement at low bit rates (bpp $< 0.10$), another solution is to find a better mechanism than equal channel slicing for synonymous level partitioning.

- **The distortion (PSNR) has degraded**, which can verify the distortion-perception tradeoff as mentioned in (Blau & Michaeli, 2018).

Based on the above analyses, the current SIC framework still has limitations but shows potential for further exploration and offers belief directions for future research on perceptual image compression.

## 7. Relevant Thoughts on Semantic Information Theory

As an extension to synonymity-based semantic information theory, our work provide guidance for practical image coding designs. In Appendix B, we further discuss the relationships with existing conclusions in semantic information theory, including the semantic entropy and the down semantic mutual information proposed in (Niu & Zhang, 2024).

## 8. Conclusions

In this paper, we consider the perceptual image compression problem from the perspective of synonymity-based semantic information theory. Specifically, we propose a synonymous variational inference (SVI) method to re-analyze the optimization direction of perceptual image compression. Based on this analysis method, we theoretically prove that the optimization direction of perceptual image compression is a triple tradeoff, i.e., synonymous rate-distortion-perception tradeoff, which is compatible with the existing rate-distortion-perception tradeoff empirically presented by previous works. Additionally, we propose a new perceptual image compression scheme, namely synonymous image compression, corresponding to the SVI analytical process, and implement a rough progressive SIC model to fully leverage the model's capabilities. Experimental results demonstrate full-rate rate-distortion perception performance and notable advantages on DISTS, thereby verifying the effectiveness of our proposed analysis method.

## Software and Data

We will upload code for reproducing our results to the repository at https://github.com/ZJLiang6412/SynonymousImageCompression.

## Acknowledgements

This work was supported by the National Natural Science Foundation of China (No. 62293481, No. 62471054).

## Impact Statement

Our work makes a significant contribution to semantic information processing. A key challenge in semantic information theory is the lack of a unified viewpoint, as many perspectives fail to effectively guide coding design. Guided by the synonymity of semantic information, this paper demonstrates the fundamental theoretical reason why the distribution distance, used to measure perceptual quality in perceptual image compression empirically, appears in the optimization objective. Moreover, our theoretical analysis provides consistent and universal guidance for designing image compression approaches, regardless of whether compression is considered for perceptual quality. To summarize, our contribution links semantic information theory to practical image coding problems, which paves the way for a more unified viewpoint of semantic information theory.

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

# A. The Proof of the Key Theoretical Results

In the main text, we present Lemma 3.2 and provide a brief proof of the optimization direction of SIC in Theorem 3.3, based on this lemma. In this appendix section, we first provide detailed proofs of Lemma 3.2 and Theorem 3.3, and briefly discuss their relationships with the existing image compression theories.

## A.1. The Proof of Lemma 3.2

**Lemma 3.2.** *When the source considers the existence of an ideal synset $\mathcal{X}$ and the decoder places the reconstructed sample in a reconstructed synset $\tilde{\mathcal{X}}$, the minimization of the expected negative log synonymous likelihood term*

$$\min \mathbb{E}_{\boldsymbol{x} \sim p(\boldsymbol{x})} \mathbb{E}_{\tilde{\boldsymbol{y}} \sim q} \left[ -\log p_{\mathcal{X}|\tilde{\boldsymbol{y}}_s} \left( \mathcal{X}|\tilde{\boldsymbol{y}}_s \right) \right] \iff \min \mathbb{E}_{\tilde{\boldsymbol{y}} \sim q} \mathbb{E}_{\tilde{\boldsymbol{x}}_i \in \tilde{\mathcal{X}}|\tilde{\boldsymbol{y}}_s} \left\{ \lambda_d \cdot \mathbb{E}_{\boldsymbol{x} \sim p(\boldsymbol{x})} \left[ d\left( \boldsymbol{x}, \tilde{\boldsymbol{x}}_i \right) \right] + \lambda_p \cdot D_{KL} \left[ p_{\boldsymbol{x}} || p_{\tilde{\boldsymbol{x}}_i} \right] \right\}, \tag{17}$$

*in which $\lambda_d$ and $\lambda_p$ are the tradeoff factors for the expected distortion term and the expected KL divergence term, respectively.*

*Proof.* According to the generative model presented in Figure 2, any sample $\boldsymbol{x}_i$ in the ideal synset $\mathcal{X}$ is expected to be capable of being generated using a synonymous representation $\tilde{\boldsymbol{y}}_s$ and a detailed sample $\hat{\boldsymbol{y}}_{\epsilon,j}$, in which the detailed sample $\hat{\boldsymbol{y}}_{\epsilon,j}$ is sampled based on a specific prior $p_{\hat{\boldsymbol{y}}_\epsilon|\tilde{\boldsymbol{y}}_s} \left( \hat{\boldsymbol{y}}_\epsilon|\tilde{\boldsymbol{y}}_s; \boldsymbol{\psi}, \boldsymbol{\theta}_p \right)$. It should be noted that $\hat{\boldsymbol{y}}_{\epsilon,j} \neq \tilde{\boldsymbol{y}}_\epsilon$ since the detailed representation $\tilde{\boldsymbol{y}}_\epsilon$ is not required to be coded and transmitted to the decoder end.

Additionally, according to the semantic information theory based on synonymity (Niu & Zhang, 2024), the probability of a synset is equal to the sum of the probability or the integral of the density of each sample within the set. Herein, we consider the integral form because image samples within an ideal synset can typically be transformed into one another through continuous changes.

Based on the above factors, we can expand the expression on the left side of (17) as follows:

$$\mathbb{E}_{\boldsymbol{x} \sim p(\boldsymbol{x})} \mathbb{E}_{\tilde{\boldsymbol{y}} \sim q} \left[ -\log p_{\mathcal{X}|\tilde{\boldsymbol{y}}_s} \left( \mathcal{X}|\tilde{\boldsymbol{y}}_s; \boldsymbol{\theta}_g \right) \right]$$

$$\stackrel{(a)}{=} \mathbb{E}_{\boldsymbol{x} \sim p(\boldsymbol{x})} \mathbb{E}_{\tilde{\boldsymbol{y}} \sim q} \left[ -\log \int_{\hat{\boldsymbol{y}}_{\epsilon,j}} p_{\mathcal{X}|\tilde{\boldsymbol{y}}_s, \hat{\boldsymbol{y}}_{\epsilon,j}} \left( \mathcal{X}|\tilde{\boldsymbol{y}}_s, \hat{\boldsymbol{y}}_{\epsilon,j}; \boldsymbol{\theta}_g \right) \cdot p_{\hat{\boldsymbol{y}}_{\epsilon,j}|\tilde{\boldsymbol{y}}_s} \left( \hat{\boldsymbol{y}}_{\epsilon,j}|\tilde{\boldsymbol{y}}_s; \boldsymbol{\psi}, \boldsymbol{\theta}_p \right) d\hat{\boldsymbol{y}}_{\epsilon,j} \right]$$

$$\stackrel{(b)}{=} \mathbb{E}_{\boldsymbol{x} \sim p(\boldsymbol{x})} \mathbb{E}_{\tilde{\boldsymbol{y}} \sim q} \left\{ -\log \mathbb{E}_{\hat{\boldsymbol{y}}_{\epsilon,j}|\tilde{\boldsymbol{y}}_s \sim p_{\hat{\boldsymbol{y}}_{\epsilon,j}|\tilde{\boldsymbol{y}}_s}} \left[ p_{\mathcal{X}|\tilde{\boldsymbol{y}}_s, \hat{\boldsymbol{y}}_{\epsilon,j}} \left( \mathcal{X}|\tilde{\boldsymbol{y}}_s, \hat{\boldsymbol{y}}_{\epsilon,j}; \boldsymbol{\theta}_g \right) \right] \right\} \tag{18}$$

$$\stackrel{(c)}{=} \mathbb{E}_{\boldsymbol{x} \sim p(\boldsymbol{x})} \mathbb{E}_{\tilde{\boldsymbol{y}} \sim q} \left\{ -\log \mathbb{E}_{\hat{\boldsymbol{y}}_{\epsilon,j}|\tilde{\boldsymbol{y}}_s \sim p_{\hat{\boldsymbol{y}}_{\epsilon,j}|\tilde{\boldsymbol{y}}_s}} \left[ \int_{\boldsymbol{x}_i \in \mathcal{X}} p_{\boldsymbol{x}_i|\tilde{\boldsymbol{y}}_s, \hat{\boldsymbol{y}}_{\epsilon,j}} \left( \boldsymbol{x}_i|\tilde{\boldsymbol{y}}_s, \hat{\boldsymbol{y}}_{\epsilon,j}; \boldsymbol{\theta}_g \right) d\boldsymbol{x}_i \right] \right\}$$

$$\stackrel{(d)}{=} \mathbb{E}_{\boldsymbol{x} \sim p(\boldsymbol{x})} \mathbb{E}_{\tilde{\boldsymbol{y}} \sim q} \left\{ -\log \mathbb{E}_{\tilde{\boldsymbol{x}}_j \in \tilde{\mathcal{X}}|\tilde{\boldsymbol{y}}_s} \left[ \int_{\boldsymbol{x}_i \in \mathcal{X}} p_{\boldsymbol{x}_i|\tilde{\boldsymbol{x}}_j} \left( \boldsymbol{x}_i|\tilde{\boldsymbol{x}}_j \right) d\boldsymbol{x}_i \right] \right\},$$

where (a) is achieved by introducing $\hat{\boldsymbol{y}}_{\epsilon,j}$ into the conditional probability with a corresponding integral; (b) is according to the definition of mathematical expectation; (c) is according to the integral relationship between the probability of the ideal synset and its samples as stated before; (d) is based on a determined generator $\boldsymbol{\theta}_g$ which can definitely map the input $\tilde{\boldsymbol{y}}_s$ and $\hat{\boldsymbol{y}}_{\epsilon,j}$ to the output $\tilde{\boldsymbol{x}}_j$, i.e., $\tilde{\boldsymbol{x}}_j = g_s \left( \tilde{\boldsymbol{y}}_s, \hat{\boldsymbol{y}}_{\epsilon,j}; \boldsymbol{\theta}_g \right)$. It should be noted that in the equation (d), the reconstructed sample $\tilde{\boldsymbol{x}}_j$ is not required to be completely aligned with the source synonymous sample $\boldsymbol{x}_i$ but for the overall ideal synset $\mathcal{X}$, which is why two subscripts, i.e., $i$ and $j$, are used here. By minimizing this term, we can use the synonym representation $\hat{\boldsymbol{y}}_s$ to obtain the reconstructed synset $\hat{\mathcal{X}}$, bringing it closer to the ideal synset $\mathcal{X}$, thus achieving the optimization objective shown in Figure 1.

Since the above results only involve the sample $\boldsymbol{x}_i$ from the ideal synset centered by the original image $\boldsymbol{x}$ and the reconstructed sample $\tilde{\boldsymbol{x}}_j$ obtained by the SIC codec, the influence of the original image sample $\boldsymbol{x}$ has not directly involved. To this end, we consider to rewrite (18) as

$$\mathbb{E}_{\boldsymbol{x} \sim p(\boldsymbol{x})} \mathbb{E}_{\tilde{\boldsymbol{y}} \sim q} \left\{ -\log \mathbb{E}_{\tilde{\boldsymbol{x}}_j \in \tilde{\mathcal{X}}|\tilde{\boldsymbol{y}}_s} \left[ \int_{\boldsymbol{x}_i \in \mathcal{X}} p_{\boldsymbol{x}_i|\tilde{\boldsymbol{x}}_j} \left( \boldsymbol{x}_i|\tilde{\boldsymbol{x}}_j \right) d\boldsymbol{x}_i \right] \right\}$$

$$\stackrel{(a)}{=} \mathbb{E}_{\boldsymbol{x} \sim p(\boldsymbol{x})} \mathbb{E}_{\tilde{\boldsymbol{y}} \sim q} \left\{ -\log \mathbb{E}_{\tilde{\boldsymbol{x}}_j \in \tilde{\mathcal{X}}|\tilde{\boldsymbol{y}}_s} \left[ \int_{\boldsymbol{x}_i \in \mathcal{X}} p_{\tilde{\boldsymbol{x}}_j|\boldsymbol{x}_i} \left( \tilde{\boldsymbol{x}}_j|\boldsymbol{x}_i \right) \cdot \frac{p_{\boldsymbol{x}_i} \left( \boldsymbol{x}_i \right)}{p_{\tilde{\boldsymbol{x}}_j} \left( \tilde{\boldsymbol{x}}_j \right)} d\boldsymbol{x}_i \right] \right\} \tag{19}$$

$$\stackrel{(b)}{=} \mathbb{E}_{\boldsymbol{x} \sim p(\boldsymbol{x})} \mathbb{E}_{\tilde{\boldsymbol{y}} \sim q} \left\{ -\log \mathbb{E}_{\tilde{\boldsymbol{x}}_j \in \tilde{\mathcal{X}}|\tilde{\boldsymbol{y}}_s} \left[ \int_{\boldsymbol{x}_i \in \mathcal{X}} p_{\tilde{\boldsymbol{x}}_j|\boldsymbol{x}_i} \left( \tilde{\boldsymbol{x}}_j|\boldsymbol{x}_i \right) \cdot \frac{p_{\boldsymbol{x}} \left( \boldsymbol{x} \right)}{p_{\tilde{\boldsymbol{x}}_j} \left( \tilde{\boldsymbol{x}}_j \right)} \cdot \frac{p_{\tilde{\boldsymbol{x}}_j} \left( \tilde{\boldsymbol{x}}_j \right)}{p_{\boldsymbol{x}} \left( \boldsymbol{x} \right)} \cdot \frac{p_{\boldsymbol{x}_i} \left( \boldsymbol{x}_i \right)}{p_{\tilde{\boldsymbol{x}}_j} \left( \tilde{\boldsymbol{x}}_j \right)} d\boldsymbol{x}_i \right] \right\},$$

in which (a) is by the Bayes' theorem, and (b) is achieved by introducing the reciprocal terms $\frac{p_{\boldsymbol{x}}(\boldsymbol{x})}{p_{\tilde{\boldsymbol{x}}_j}(\tilde{\boldsymbol{x}}_j)}$ and $\frac{p_{\tilde{\boldsymbol{x}}_j}(\tilde{\boldsymbol{x}}_j)}{p_{\boldsymbol{x}}(\boldsymbol{x})}$.

Furthermore, since any sample $\boldsymbol{x}_i$ in the ideal synset (including the original image sample $\boldsymbol{x}$) shares the same synonymous representation $\tilde{\boldsymbol{y}}_s$ by the ideal parametric SIC encoder with $\phi_{\boldsymbol{g}}^*$, the posterior term in the (19) satisfy the following equations:

$$p_{\tilde{\boldsymbol{x}}_j|\boldsymbol{x}_i}(\tilde{\boldsymbol{x}}_j|\boldsymbol{x}_i) = p_{\tilde{\boldsymbol{x}}_j|\tilde{\boldsymbol{y}}_s}(\tilde{\boldsymbol{x}}_j|\tilde{\boldsymbol{y}}_s) = p_{\tilde{\boldsymbol{x}}_j|\boldsymbol{x}}(\tilde{\boldsymbol{x}}_j|\boldsymbol{x}). \tag{20}$$

Therefore, the result of (19) can be further derived as

$$
\begin{aligned}
&\mathbb{E}_{\boldsymbol{x}\sim p(\boldsymbol{x})}\mathbb{E}_{\tilde{\boldsymbol{y}}\sim q}\left\{-\log\mathbb{E}_{\tilde{\boldsymbol{x}}_j\in\tilde{\boldsymbol{\mathcal{X}}}|\tilde{\boldsymbol{y}}_s}\left[\int_{\boldsymbol{x}_i\in\boldsymbol{\mathcal{X}}}p_{\tilde{\boldsymbol{x}}_j|\boldsymbol{x}_i}(\tilde{\boldsymbol{x}}_j|\boldsymbol{x}_i)\cdot\frac{p_{\boldsymbol{x}}(\boldsymbol{x})}{p_{\tilde{\boldsymbol{x}}_j}(\tilde{\boldsymbol{x}}_j)}\cdot\frac{p_{\tilde{\boldsymbol{x}}_j}(\tilde{\boldsymbol{x}}_j)}{p_{\boldsymbol{x}}(\boldsymbol{x})}\cdot\frac{p_{\boldsymbol{x}_i}(\boldsymbol{x}_i)}{p_{\tilde{\boldsymbol{x}}_j}(\tilde{\boldsymbol{x}}_j)}d\boldsymbol{x}_i\right]\right\}\\
&\overset{(a)}{=}\mathbb{E}_{\boldsymbol{x}\sim p(\boldsymbol{x})}\mathbb{E}_{\tilde{\boldsymbol{y}}\sim q}\left\{-\log\mathbb{E}_{\tilde{\boldsymbol{x}}_j\in\tilde{\boldsymbol{\mathcal{X}}}|\tilde{\boldsymbol{y}}_s}\left[\int_{\boldsymbol{x}_i\in\boldsymbol{\mathcal{X}}}\left(p_{\tilde{\boldsymbol{x}}_j|\boldsymbol{x}}(\tilde{\boldsymbol{x}}_j|\boldsymbol{x})\cdot\frac{p_{\boldsymbol{x}}(\boldsymbol{x})}{p_{\tilde{\boldsymbol{x}}_j}(\tilde{\boldsymbol{x}}_j)}\right)\cdot\frac{p_{\tilde{\boldsymbol{x}}_j}(\tilde{\boldsymbol{x}}_j)}{p_{\boldsymbol{x}}(\boldsymbol{x})}\cdot\frac{p_{\boldsymbol{x}_i}(\boldsymbol{x}_i)}{p_{\tilde{\boldsymbol{x}}_j}(\tilde{\boldsymbol{x}}_j)}d\boldsymbol{x}_i\right]\right\}\\
&\overset{(b)}{=}\mathbb{E}_{\boldsymbol{x}\sim p(\boldsymbol{x})}\mathbb{E}_{\tilde{\boldsymbol{y}}\sim q}\left\{-\log\mathbb{E}_{\tilde{\boldsymbol{x}}_j\in\tilde{\boldsymbol{\mathcal{X}}}|\tilde{\boldsymbol{y}}_s}\left[\int_{\boldsymbol{x}_i\in\boldsymbol{\mathcal{X}}}p_{\boldsymbol{x}|\tilde{\boldsymbol{x}}_j}(\boldsymbol{x}|\tilde{\boldsymbol{x}}_j)\cdot\frac{p_{\tilde{\boldsymbol{x}}_j}(\tilde{\boldsymbol{x}}_j)}{p_{\boldsymbol{x}}(\boldsymbol{x})}\cdot\frac{p_{\boldsymbol{x}_i}(\boldsymbol{x}_i)}{p_{\tilde{\boldsymbol{x}}_j}(\tilde{\boldsymbol{x}}_j)}d\boldsymbol{x}_i\right]\right\}\\
&=\mathbb{E}_{\boldsymbol{x}\sim p(\boldsymbol{x})}\mathbb{E}_{\tilde{\boldsymbol{y}}\sim q}\left\{-\log\mathbb{E}_{\tilde{\boldsymbol{x}}_j\in\tilde{\boldsymbol{\mathcal{X}}}|\tilde{\boldsymbol{y}}_s}\left[p_{\boldsymbol{x}|\tilde{\boldsymbol{x}}_j}(\boldsymbol{x}|\tilde{\boldsymbol{x}}_j)\cdot\frac{p_{\tilde{\boldsymbol{x}}_j}(\tilde{\boldsymbol{x}}_j)}{p_{\boldsymbol{x}}(\boldsymbol{x})}\cdot\int_{\boldsymbol{x}_i\in\boldsymbol{\mathcal{X}}}\frac{p_{\boldsymbol{x}_i}(\boldsymbol{x}_i)}{p_{\tilde{\boldsymbol{x}}_j}(\tilde{\boldsymbol{x}}_j)}d\boldsymbol{x}_i\right]\right\}\\
&\overset{(c)}{=}\mathbb{E}_{\boldsymbol{x}\sim p(\boldsymbol{x})}\mathbb{E}_{\tilde{\boldsymbol{y}}\sim q}\left\{-\log\mathbb{E}_{\tilde{\boldsymbol{x}}_j\in\tilde{\boldsymbol{\mathcal{X}}}|\tilde{\boldsymbol{y}}_s}\left[p_{\boldsymbol{x}|\tilde{\boldsymbol{x}}_j}(\boldsymbol{x}|\tilde{\boldsymbol{x}}_j)\cdot\frac{p_{\tilde{\boldsymbol{x}}_j}(\tilde{\boldsymbol{x}}_j)}{p_{\boldsymbol{x}}(\boldsymbol{x})}\cdot\left(\frac{1}{|\boldsymbol{\mathcal{X}}|}\int_{\boldsymbol{x}_i\in\boldsymbol{\mathcal{X}}}\frac{p_{\boldsymbol{x}_i}(\boldsymbol{x}_i)}{p_{\tilde{\boldsymbol{x}}_j}(\tilde{\boldsymbol{x}}_j)}d\boldsymbol{x}_i\right)\cdot|\boldsymbol{\mathcal{X}}|\right]\right\}\\
&\overset{(d)}{\leq}\mathbb{E}_{\boldsymbol{x}\sim p(\boldsymbol{x})}\mathbb{E}_{\tilde{\boldsymbol{y}}\sim q}\mathbb{E}_{\tilde{\boldsymbol{x}}_j\in\tilde{\boldsymbol{\mathcal{X}}}|\tilde{\boldsymbol{y}}_s}\left[-\log p_{\boldsymbol{x}|\tilde{\boldsymbol{x}}_j}(\boldsymbol{x}|\tilde{\boldsymbol{x}}_j)+\log\frac{p_{\boldsymbol{x}}(\boldsymbol{x})}{p_{\tilde{\boldsymbol{x}}_j}(\tilde{\boldsymbol{x}}_j)}-\log\frac{1}{|\boldsymbol{\mathcal{X}}|}\int_{\boldsymbol{x}_i\in\boldsymbol{\mathcal{X}}}\frac{p_{\boldsymbol{x}_i}(\boldsymbol{x}_i)}{p_{\tilde{\boldsymbol{x}}_j}(\tilde{\boldsymbol{x}}_j)}d\boldsymbol{x}_i-\log|\boldsymbol{\mathcal{X}}|\right]\\
&\overset{(e)}{\leq}\mathbb{E}_{\boldsymbol{x}\sim p(\boldsymbol{x})}\mathbb{E}_{\tilde{\boldsymbol{y}}\sim q}\mathbb{E}_{\tilde{\boldsymbol{x}}_j\in\tilde{\boldsymbol{\mathcal{X}}}|\tilde{\boldsymbol{y}}_s}\left[-\log p_{\boldsymbol{x}|\tilde{\boldsymbol{x}}_j}(\boldsymbol{x}|\tilde{\boldsymbol{x}}_j)+\log\frac{p_{\boldsymbol{x}}(\boldsymbol{x})}{p_{\tilde{\boldsymbol{x}}_j}(\tilde{\boldsymbol{x}}_j)}-\frac{1}{|\boldsymbol{\mathcal{X}}|}\int_{\boldsymbol{x}_i\in\boldsymbol{\mathcal{X}}}\log\frac{p_{\boldsymbol{x}_i}(\boldsymbol{x}_i)}{p_{\tilde{\boldsymbol{x}}_j}(\tilde{\boldsymbol{x}}_j)}d\boldsymbol{x}_i-\log|\boldsymbol{\mathcal{X}}|\right],
\end{aligned}
\tag{21}
$$

where (a) replaces $p_{\tilde{\boldsymbol{x}}_j|\boldsymbol{x}_i}(\tilde{\boldsymbol{x}}_j|\boldsymbol{x}_i)$ with $p_{\tilde{\boldsymbol{x}}_j|\boldsymbol{x}}(\tilde{\boldsymbol{x}}_j|\boldsymbol{x})$ using (20); (b) is derived by the Bayes' theorem in reverse; (c) introduces $|\boldsymbol{\mathcal{X}}|$ and its reciprocal, in which $|\boldsymbol{\mathcal{X}}|$ denotes the size of the ideal synset $\boldsymbol{\mathcal{X}}$, and can be used to express an arithmetic mean along with the followed integral over $\boldsymbol{\mathcal{X}}$; (d) and (e) can be scaled based on the Jensen's inequalities, respectively, since $-\log(\cdot)$ is a convex function.

Next, we examine the result of (21) separately.

1. **The Derivation of the First Term.** $\mathbb{E}_{\boldsymbol{x}\sim p(\boldsymbol{x})}\mathbb{E}_{\tilde{\boldsymbol{y}}\sim q}\mathbb{E}_{\tilde{\boldsymbol{x}}_j\in\tilde{\boldsymbol{\mathcal{X}}}|\tilde{\boldsymbol{y}}_s}\left[-\log p_{\boldsymbol{x}|\tilde{\boldsymbol{x}}_j}(\boldsymbol{x}|\tilde{\boldsymbol{x}}_j)\right]$ denotes an expected distortion which averaged on the source images and the corresponding samples of the reconstructed synsets. As a typical case, when the likelihood probability $p_{\boldsymbol{x}|\tilde{\boldsymbol{x}}_j}(\boldsymbol{x}|\tilde{\boldsymbol{x}}_j)$ follows a Gaussian distribution $\mathcal{N}(\boldsymbol{x}|\tilde{\boldsymbol{x}}_j,\sigma^2\boldsymbol{I}_d)$ (in which $d$ denotes the dimension of the original image $\boldsymbol{x}$), this term will be equivalent to

$$\mathbb{E}_{\boldsymbol{x}\sim p(\boldsymbol{x})}\mathbb{E}_{\tilde{\boldsymbol{y}}\sim q}\mathbb{E}_{\tilde{\boldsymbol{x}}_j\in\tilde{\boldsymbol{\mathcal{X}}}|\tilde{\boldsymbol{y}}_s}\left[-\log p_{\boldsymbol{x}|\tilde{\boldsymbol{x}}_j}(\boldsymbol{x}|\tilde{\boldsymbol{x}}_j)\right]=\frac{1}{2\sigma^2}\cdot\mathbb{E}_{\boldsymbol{x}\sim p(\boldsymbol{x})}\mathbb{E}_{\tilde{\boldsymbol{y}}\sim q}\mathbb{E}_{\tilde{\boldsymbol{x}}_j\in\tilde{\boldsymbol{\mathcal{X}}}|\tilde{\boldsymbol{y}}_s}||\boldsymbol{x}-\tilde{\boldsymbol{x}}||^2+\frac{d}{2}\log(2\pi\sigma^2),\tag{22}$$

in which the $\sigma^2$ is the variance term of the set Gaussian distribution, i.e., the power of the quantization noise. In this case, the term can be considered as a weighted Expected Mean Squared Error (E-MSE) loss (instead of the Mean Squared Error (MSE) loss) plus a constant.

In typical LIC methods (Ballé et al., 2017; 2018), the multiplier $\frac{1}{2\sigma^2}$ is often replaced with a hyperparameter $\lambda$ as the tradeoff factor to the MSE loss to the balance with the coding rate. However, in SIC, if E-MSE is used as the distortion loss term, it is incomplete to define the physical meaning of the hyperparameter as $\frac{1}{2\sigma^2}$ alone: The effect of scaling, due to the inequality in (21)(d), should be also considered. Therefore, we summarize the analysis results as

$$\lambda_d\cdot\mathbb{E}_{\boldsymbol{x}\sim p(\boldsymbol{x})}\mathbb{E}_{\tilde{\boldsymbol{y}}\sim q}\mathbb{E}_{\tilde{\boldsymbol{x}}_j\in\tilde{\boldsymbol{\mathcal{X}}}|\tilde{\boldsymbol{y}}_s}||\boldsymbol{x}-\tilde{\boldsymbol{x}}_j||^2+\text{const},\tag{23}$$

in which $\lambda_d=\alpha_d\left(|\tilde{\boldsymbol{\mathcal{X}}}|\right)\cdot\frac{1}{2\sigma^2}$. The multiplier $\alpha_d\left(|\tilde{\boldsymbol{\mathcal{X}}}|\right)$ is a scaling factor influenced by the size of the reconstructed synset $\tilde{\boldsymbol{\mathcal{X}}}$, and it can be implicitly incorporated into the value of the hyperparameter $\lambda_d$ thus it does not need to be

explicitly assigned. It should be noted that when the equality condition of Jensen inequality (21)(d) is satisfied or the reconstructed synset contains only one sample, the multiplier $\alpha_d\left(\left|\tilde{\boldsymbol{\mathcal{X}}}\right|\right) = 1$, and the analysis result will be degraded into $\lambda \cdot \mathbb{E}_{\boldsymbol{x}\sim p(\boldsymbol{x})}\mathbb{E}_{\tilde{\boldsymbol{y}}\sim q}\left\|\boldsymbol{x} - \tilde{\boldsymbol{x}}\right\|^2 + \text{const}$.

When other distortion measures (such as Expected MS-SSIM, abbreviated as E-MS-SSIM) are used instead of E-MSE, the situation is similar and will not be discussed further. Herein, we give a general analysis result as

$$\lambda_d \cdot \mathbb{E}_{\boldsymbol{x}\sim p(\boldsymbol{x})}\mathbb{E}_{\tilde{\boldsymbol{y}}\sim q}\mathbb{E}_{\tilde{\boldsymbol{x}}_j\in\tilde{\boldsymbol{\mathcal{X}}}|\tilde{\boldsymbol{y}}_s}\left[d\left(\boldsymbol{x}, \tilde{\boldsymbol{x}}_j\right)\right] + \text{const}, \tag{24}$$

in which $d\left(\cdot\right)$ denotes any distortion measure between the original image $\boldsymbol{x}$ and the reconstructed sample $\tilde{\boldsymbol{x}}_j$.

2. **The Derivation of the Second Term and the Third Term.** These two terms should be firstly considered together due to their linkage, which can be derived as

$$\begin{aligned}
&\mathbb{E}_{\boldsymbol{x}\sim p(\boldsymbol{x})}\mathbb{E}_{\tilde{\boldsymbol{y}}\sim q}\mathbb{E}_{\tilde{\boldsymbol{x}}_j\in\tilde{\boldsymbol{\mathcal{X}}}|\tilde{\boldsymbol{y}}_s}\left[\log\frac{p_{\boldsymbol{x}}\left(\boldsymbol{x}\right)}{p_{\tilde{\boldsymbol{x}}_j}\left(\tilde{\boldsymbol{x}}_j\right)} - \frac{1}{|\boldsymbol{\mathcal{X}}|}\int_{\boldsymbol{x}_i\in\boldsymbol{\mathcal{X}}}\log\frac{p_{\boldsymbol{x}_i}\left(\boldsymbol{x}_i\right)}{p_{\tilde{\boldsymbol{x}}_j}\left(\tilde{\boldsymbol{x}}_j\right)}d\boldsymbol{x}_i\right]\\
&= \mathbb{E}_{\boldsymbol{x}\sim p(\boldsymbol{x})}\mathbb{E}_{\tilde{\boldsymbol{y}}\sim q}\mathbb{E}_{\tilde{\boldsymbol{x}}_j\in\tilde{\boldsymbol{\mathcal{X}}}|\tilde{\boldsymbol{y}}_s}\left[\frac{1}{|\boldsymbol{\mathcal{X}}|}\int_{\boldsymbol{x}_i\in\boldsymbol{\mathcal{X}}}\left(\log\frac{p_{\boldsymbol{x}}\left(\boldsymbol{x}\right)}{p_{\tilde{\boldsymbol{x}}_j}\left(\tilde{\boldsymbol{x}}_j\right)} - \log\frac{p_{\boldsymbol{x}_i}\left(\boldsymbol{x}_i\right)}{p_{\tilde{\boldsymbol{x}}_j}\left(\tilde{\boldsymbol{x}}_j\right)}\right)d\boldsymbol{x}_i\right]\\
&= \mathbb{E}_{\boldsymbol{x}\sim p(\boldsymbol{x})}\mathbb{E}_{\tilde{\boldsymbol{y}}\sim q}\mathbb{E}_{\tilde{\boldsymbol{x}}_j\in\tilde{\boldsymbol{\mathcal{X}}}|\tilde{\boldsymbol{y}}_s}\left[\frac{1}{|\boldsymbol{\mathcal{X}}|}\int_{\boldsymbol{x}_i\in\boldsymbol{\mathcal{X}}}\log\frac{p_{\boldsymbol{x}}\left(\boldsymbol{x}\right)}{p_{\tilde{\boldsymbol{x}}_j}\left(\tilde{\boldsymbol{x}}_j\right)}\cdot\frac{p_{\tilde{\boldsymbol{x}}_j}\left(\tilde{\boldsymbol{x}}_j\right)}{p_{\boldsymbol{x}_i}\left(\boldsymbol{x}_i\right)}d\boldsymbol{x}_i\right]\\
&= \mathbb{E}_{\boldsymbol{x}\sim p(\boldsymbol{x})}\left[\frac{1}{|\boldsymbol{\mathcal{X}}|}\int_{\boldsymbol{x}_i\in\boldsymbol{\mathcal{X}}}\log\frac{p_{\boldsymbol{x}}\left(\boldsymbol{x}\right)}{p_{\boldsymbol{x}_i}\left(\boldsymbol{x}_i\right)}d\boldsymbol{x}_i\right]\\
&= \frac{1}{|\boldsymbol{\mathcal{X}}|}\int_{\boldsymbol{x}_i\in\boldsymbol{\mathcal{X}}}\mathbb{E}_{\boldsymbol{x}\sim p(\boldsymbol{x})}\left[\log\frac{p_{\boldsymbol{x}}\left(\boldsymbol{x}\right)}{p_{\boldsymbol{x}_i}\left(\boldsymbol{x}_i\right)}\right]d\boldsymbol{x}_i\\
&= \frac{1}{|\boldsymbol{\mathcal{X}}|}\int_{\boldsymbol{x}_i\in\boldsymbol{\mathcal{X}}}D_{\text{KL}}\left[p_{\boldsymbol{x}}||p_{\boldsymbol{x}_i}\right]d\boldsymbol{x}_i = f\left(\boldsymbol{x}, \boldsymbol{\mathcal{X}}\right),
\end{aligned} \tag{25}$$

i.e., the arithmetic mean of the KL divergence between the original sample $\boldsymbol{x}$ and the synonymous sample $\boldsymbol{x}_i$, which is a non-negative function $f\left(\boldsymbol{x}, \boldsymbol{\mathcal{X}}\right)$ of the original sample $\boldsymbol{x}$ and the ideal synset $\boldsymbol{\mathcal{X}}$. If these two factors at the source are determined, the result of the function will be constant. For a special case, when $\boldsymbol{\mathcal{X}}$ contains only one sample, i.e., the original image $\boldsymbol{x}$, this constant will be equal to 0 since the two distributions reduce to only one distribution $p_{\boldsymbol{x}}$.

Despite the foregoing facts, if it is treated as a constant, the existence of the ideal synset on the source will lose its meaning during the minimization process, and no samples with perceptual similarity to the source image will be available to provide a reference for the reconstructed samples. Therefore, we need to consider the second and third terms separately to determine the meaning of the constant value in the optimization process:

- For the second term $\mathbb{E}_{\boldsymbol{x}\sim p(\boldsymbol{x})}\mathbb{E}_{\tilde{\boldsymbol{y}}\sim q}\mathbb{E}_{\tilde{\boldsymbol{x}}_j\in\tilde{\boldsymbol{\mathcal{X}}}|\tilde{\boldsymbol{y}}_s}\left[\log\frac{p_{\boldsymbol{x}}\left(\boldsymbol{x}\right)}{p_{\tilde{\boldsymbol{x}}_j}\left(\tilde{\boldsymbol{x}}_j\right)}\right]$, it can be further derived as

$$\begin{aligned}
\mathbb{E}_{\boldsymbol{x}\sim p(\boldsymbol{x})}\mathbb{E}_{\tilde{\boldsymbol{y}}\sim q}\mathbb{E}_{\tilde{\boldsymbol{x}}_j\in\tilde{\boldsymbol{\mathcal{X}}}|\tilde{\boldsymbol{y}}_s}\left[\log\frac{p_{\boldsymbol{x}}\left(\boldsymbol{x}\right)}{p_{\tilde{\boldsymbol{x}}_j}\left(\tilde{\boldsymbol{x}}_j\right)}\right] &= \mathbb{E}_{\tilde{\boldsymbol{y}}\sim q}\mathbb{E}_{\tilde{\boldsymbol{x}}_j\in\tilde{\boldsymbol{\mathcal{X}}}|\tilde{\boldsymbol{y}}_s}\left[\mathbb{E}_{\boldsymbol{x}\sim p(\boldsymbol{x})}\log\frac{p_{\boldsymbol{x}}\left(\boldsymbol{x}\right)}{p_{\tilde{\boldsymbol{x}}_j}\left(\tilde{\boldsymbol{x}}_j\right)}\right]\\
&= \mathbb{E}_{\tilde{\boldsymbol{y}}\sim q}\mathbb{E}_{\tilde{\boldsymbol{x}}_j\in\tilde{\boldsymbol{\mathcal{X}}}|\tilde{\boldsymbol{y}}_s}D_{\text{KL}}\left[p_{\boldsymbol{x}}||p_{\tilde{\boldsymbol{x}}_j}\right],
\end{aligned} \tag{26}$$

  i.e., an Expected KL Divergence (E-KLD) between the distribution of the original image $p_{\boldsymbol{x}}$ and the distribution of the reconstructed sample $p_{\tilde{\boldsymbol{x}}_j}$ that averaged on the reconstructed synset $\tilde{\boldsymbol{\mathcal{X}}}$.

- As for the third term, i.e., $\mathbb{E}_{\boldsymbol{x}\sim p(\boldsymbol{x})}\mathbb{E}_{\tilde{\boldsymbol{y}}\sim q}\mathbb{E}_{\tilde{\boldsymbol{x}}_j\in\tilde{\boldsymbol{\mathcal{X}}}|\tilde{\boldsymbol{y}}_s}\left[-\frac{1}{|\boldsymbol{\mathcal{X}}|}\int_{\boldsymbol{x}_i\in\boldsymbol{\mathcal{X}}}\log\frac{p_{\boldsymbol{x}_i}\left(\boldsymbol{x}_i\right)}{p_{\tilde{\boldsymbol{x}}_j}\left(\tilde{\boldsymbol{x}}_j\right)}d\boldsymbol{x}_i\right]$, although it has a similar form to KL divergence, it cannot be called KL divergence because it uses the arithmetic mean instead of the mathematical expectation, thus lacking the non-negative properties of KL divergence. It should be noted that the outer mathematic expectations $\mathbb{E}_{\boldsymbol{x}\sim p(\boldsymbol{x})}\mathbb{E}_{\tilde{\boldsymbol{y}}\sim q}\mathbb{E}_{\tilde{\boldsymbol{x}}_j\in\tilde{\boldsymbol{\mathcal{X}}}|\tilde{\boldsymbol{y}}_s}$ is actually calculated the expectation value according to the conditional probability $p_{\tilde{\boldsymbol{x}}_j|\boldsymbol{x}}\left(\tilde{\boldsymbol{x}}_j|\boldsymbol{x}\right)$ instead of $p_{\tilde{\boldsymbol{x}}_j}\left(\tilde{\boldsymbol{x}}_j\right)$, thus the result cannot be regarded as KL divergence from this perspective neither. In spite of this, we can intuitively find the conditions under which this term equals 0, which can be expressed as

$$\mathbb{E}_{\boldsymbol{x}\sim p(\boldsymbol{x})}\mathbb{E}_{\tilde{\boldsymbol{y}}\sim q}\mathbb{E}_{\tilde{\boldsymbol{x}}_j\in\tilde{\boldsymbol{\mathcal{X}}}|\tilde{\boldsymbol{y}}_s}\left[-\frac{1}{|\boldsymbol{\mathcal{X}}|}\int_{\boldsymbol{x}_i\in\boldsymbol{\mathcal{X}}}\log\frac{p_{\boldsymbol{x}_i}\left(\boldsymbol{x}_i\right)}{p_{\tilde{\boldsymbol{x}}_j}\left(\tilde{\boldsymbol{x}}_j\right)}d\boldsymbol{x}_i\right] = 0 \quad\Longleftrightarrow\quad p_{\boldsymbol{x}_i}\left(\boldsymbol{x}_i\right) = p_{\tilde{\boldsymbol{x}}_j}\left(\tilde{\boldsymbol{x}}_j\right) \quad \forall i, j \tag{27}$$

i.e., for arbitrary sample pair $(\boldsymbol{x}_i, \tilde{\boldsymbol{x}}_j)$, the probabilities of both samples are equal. To facilitate subsequent analysis, we label this term as $-\delta_p$.

Based on the above analysis, we obtain the following equation relationship:

$$\mathbb{E}_{\tilde{\boldsymbol{y}}\sim q}\mathbb{E}_{\tilde{\boldsymbol{x}}_j\in\tilde{\boldsymbol{\mathcal{X}}}|\tilde{\boldsymbol{y}}_s}D_{\mathrm{KL}}\left[p_{\boldsymbol{x}}||p_{\tilde{\boldsymbol{x}}_j}\right] - \delta_p = \frac{1}{|\boldsymbol{\mathcal{X}}|}\int_{\boldsymbol{x}_i\in\boldsymbol{\mathcal{X}}}D_{\mathrm{KL}}\left[p_{\boldsymbol{x}}||p_{\boldsymbol{x}_i}\right]d\boldsymbol{x}_i = f\left(\boldsymbol{x}, \boldsymbol{\mathcal{X}}\right). \tag{28}$$

From this formula, we can see that although $f\left(\boldsymbol{x}, \boldsymbol{\mathcal{X}}\right)$ is a constant when $\boldsymbol{x}$ and $\boldsymbol{\mathcal{X}}$ are determined, we can approximate it by minimizing the E-KLD term $\mathbb{E}_{\tilde{\boldsymbol{x}}_j\in\tilde{\boldsymbol{\mathcal{X}}}|\tilde{\boldsymbol{y}}_s}D_{\mathrm{KL}}\left[p_{\boldsymbol{x}}||p_{\tilde{\boldsymbol{x}}_j}\right]$. For a special case, by forcing $\delta_p = 0$ with equal probability sampling for $\tilde{\boldsymbol{x}}_j$, the E-KLD term $\mathbb{E}_{\tilde{\boldsymbol{x}}_j\in\tilde{\boldsymbol{\mathcal{X}}}|\tilde{\boldsymbol{y}}_s}D_{\mathrm{KL}}\left[p_{\boldsymbol{x}}||p_{\tilde{\boldsymbol{x}}_j}\right]$ is equal to the arithmetic mean term $\frac{1}{|\boldsymbol{\mathcal{X}}|}\int_{\boldsymbol{x}_i\in\boldsymbol{\mathcal{X}}}D_{\mathrm{KL}}\left[p_{\boldsymbol{x}}||p_{\boldsymbol{x}_i}\right]d\boldsymbol{x}_i$.

Similar to the first term, considering the effect of scaling in (3.2)(d), we summarize the analysis results as

$$\lambda_p \cdot \mathbb{E}_{\tilde{\boldsymbol{y}}\sim q}\mathbb{E}_{\tilde{\boldsymbol{x}}_j\in\tilde{\boldsymbol{\mathcal{X}}}|\tilde{\boldsymbol{y}}_s}D_{\mathrm{KL}}\left[p_{\boldsymbol{x}}||p_{\tilde{\boldsymbol{x}}_j}\right], \tag{29}$$

in which $\lambda_p = 1_{|\boldsymbol{\mathcal{X}}|\neq 1}\left(|\boldsymbol{\mathcal{X}}|\right)\cdot\alpha_p\left(|\tilde{\boldsymbol{\mathcal{X}}}|\right)$ is also a scaling factor influenced by the size of the ideal synset $\boldsymbol{\mathcal{X}}$ and the reconstructed synset $\tilde{\boldsymbol{\mathcal{X}}}$, and $1_{|\boldsymbol{\mathcal{X}}|\neq 1}$ is a indicated function of the size of the ideal synset $|\boldsymbol{\mathcal{X}}|$. Similar to the first term, we discuss the following two special cases:

- When the equality condition of Jensen inequality (21)(e) is satisfied, or the ideal synset is considered with multiple samples while the reconstructed synset contains only one, the multiplier $1_{|\boldsymbol{\mathcal{X}}|\neq 1}\left(|\boldsymbol{\mathcal{X}}|\right) = 1$ and $\alpha_p\left(|\tilde{\boldsymbol{\mathcal{X}}}|\right) = 1$, and the analysis result will be degraded into $\mathbb{E}_{\tilde{\boldsymbol{y}}\sim q}D_{\mathrm{KL}}\left[p_{\boldsymbol{x}}||p_{\tilde{\boldsymbol{x}}}\right]$.
- When the ideal synset contains only one sample, i.e., the original image $\boldsymbol{x}$, the multiplier $1_{|\boldsymbol{\mathcal{X}}|\neq 1}\left(|\boldsymbol{\mathcal{X}}|\right) = 0$, which makes the analysis result equal to 0.

3. **The Derivation of the Fourth Term.** With the determination of the ideal synset $\boldsymbol{\mathcal{X}}$ for the original image $\boldsymbol{x}$, the term $-\log|\boldsymbol{\mathcal{X}}|$ is a constant that cannot be optimized.

To summarize, by consolidating the above analysis results, we complete the proof of Lemma 3.2, that is,

$$\begin{aligned}
&\min \mathbb{E}_{\boldsymbol{x}\sim p(\boldsymbol{x})}\mathbb{E}_{\tilde{\boldsymbol{y}}\sim q}\left[-\log p_{\boldsymbol{\mathcal{X}}|\tilde{\boldsymbol{y}}_s}\left(\boldsymbol{\mathcal{X}}|\tilde{\boldsymbol{y}}_s\right)\right] \\
&= \min\left\{\lambda_d\cdot\mathbb{E}_{\boldsymbol{x}\sim p(\boldsymbol{x})}\mathbb{E}_{\tilde{\boldsymbol{y}}\sim q}\mathbb{E}_{\tilde{\boldsymbol{x}}_j\in\tilde{\boldsymbol{\mathcal{X}}}|\tilde{\boldsymbol{y}}_s}\left[d\left(\boldsymbol{x},\tilde{\boldsymbol{x}}_i\right)\right] + \text{const}\right\} + \left\{\lambda_p\cdot\mathbb{E}_{\tilde{\boldsymbol{y}}\sim q}\mathbb{E}_{\tilde{\boldsymbol{x}}_j\in\tilde{\boldsymbol{\mathcal{X}}}|\tilde{\boldsymbol{y}}_s}D_{\mathrm{KL}}\left[p_{\boldsymbol{x}}||p_{\tilde{\boldsymbol{x}}_j}\right]\right\} + \text{const} \quad (30)\\
&\Leftrightarrow \min\mathbb{E}_{\tilde{\boldsymbol{y}}\sim q}\mathbb{E}_{\tilde{\boldsymbol{x}}_i\in\tilde{\boldsymbol{\mathcal{X}}}|\tilde{\boldsymbol{y}}_s}\left\{\lambda_d\cdot\mathbb{E}_{\boldsymbol{x}\sim p(\boldsymbol{x})}\left[d\left(\boldsymbol{x},\tilde{\boldsymbol{x}}_i\right)\right] + \lambda_p\cdot D_{\mathrm{KL}}\left[p_{\boldsymbol{x}}||p_{\tilde{\boldsymbol{x}}_i}\right]\right\}.
\end{aligned}$$

$\square$

## A.2. The Proof of Theorem 3.3

**Theorem 3.3.** *For an image source $\boldsymbol{x}\sim p\left(\boldsymbol{x}\right)$ together with its bounded expected distortion $\mathbb{E}_{\boldsymbol{x}\sim p(\boldsymbol{x})}\mathbb{E}_{\hat{\boldsymbol{x}}_i\in\hat{\boldsymbol{\mathcal{X}}}|\hat{\boldsymbol{y}}_s}\left[d\left(\boldsymbol{x},\hat{\boldsymbol{x}}_i\right)\right]$ and expected KL divergence $\mathbb{E}_{\hat{\boldsymbol{x}}_i\in\hat{\boldsymbol{\mathcal{X}}}|\hat{\boldsymbol{y}}_s}D_{KL}\left[p_{\boldsymbol{x}}||p_{\hat{\boldsymbol{x}}_i}\right]$, the minimum achievable rate of perceptual image compression is*

$$\begin{aligned}
R\left(\boldsymbol{\mathcal{X}}\right) = \min_{p(\hat{\boldsymbol{\mathcal{X}}}|\boldsymbol{x})}\ &I\left(\boldsymbol{X};\hat{\tilde{\boldsymbol{X}}}\right) \\
\text{s.t.}\quad &\mathbb{E}_{\boldsymbol{x}\sim p(\boldsymbol{x})}\mathbb{E}_{\hat{\boldsymbol{x}}_i\in\hat{\boldsymbol{\mathcal{X}}}|\hat{\boldsymbol{y}}_s}\left[d\left(\boldsymbol{x},\hat{\boldsymbol{x}}_i\right)\right] \leq D, \\
&\mathbb{E}_{\hat{\boldsymbol{x}}_i\in\hat{\boldsymbol{\mathcal{X}}}|\hat{\boldsymbol{y}}_s}D_{KL}\left[p_{\boldsymbol{x}}||p_{\hat{\boldsymbol{x}}_i}\right] \leq P,
\end{aligned} \tag{31}$$

*where $I\left(\boldsymbol{X};\hat{\tilde{\boldsymbol{X}}}\right) = H_s\left(\hat{\tilde{\boldsymbol{X}}}\right) - H_s\left(\hat{\tilde{\boldsymbol{X}}}|\boldsymbol{X}\right)$ with semantic variable $\hat{\tilde{\boldsymbol{X}}}$ corresponds to the reconstructed synset $\hat{\boldsymbol{\mathcal{X}}}$.*

*Proof.* The key point to prove this problem is to consider an ideal scenario, in which there are multiple image samples $\boldsymbol{x}_i$ at the source with similar perceptual similarities to the original image $\boldsymbol{x}$. In this scenario, each sample can be assumed to be potentially generated by the ideal perceptual image decoder. To this end, it is necessary to assume the existence of an ideal

synset $\boldsymbol{\mathcal{X}}$ at the source which can encompass these samples (including the original image $\boldsymbol{x}$). These samples must share the same synonymous representations which can be represented as $\tilde{\boldsymbol{y}}_s$ in the latent space, while the unique detailed features of each sample should be represented as $\tilde{\boldsymbol{y}}_\epsilon$ in the latent space.

Based on this assumption, there should be an ideal image codec, in which the encoder ensures that any sample within the ideal synset $\boldsymbol{\mathcal{X}}$ obtains the same synonymous representation $\tilde{\boldsymbol{y}}_s$ after encoding, while the decoder, upon receiving only the synonymous representation $\tilde{\boldsymbol{y}}_s$, can generate different samples $\boldsymbol{x}_i$ within the ideal synset $\boldsymbol{\mathcal{X}}$ through sampling $\hat{\boldsymbol{y}}_{\epsilon,j}$. The optimization process leading to this ideal encoder-decoder pair can be modeled as a variational auto-encoder model, which can be achieved by minimizing the partial semantic KL divergence based on the idea of the proposed synonymous variational inference (SVI), that is,

$$
\begin{aligned}
\mathbb{E}_{\boldsymbol{x}\sim p(\boldsymbol{x})} D_{\mathrm{KL},s}\left[q||p_{\tilde{\boldsymbol{y}}_s|\boldsymbol{\mathcal{X}}}\right] &= \mathbb{E}_{\boldsymbol{x}\sim p(\boldsymbol{x})}\mathbb{E}_{\tilde{\boldsymbol{y}}|\boldsymbol{x}\sim q}\left[\log\frac{q\left(\tilde{\boldsymbol{y}}|\boldsymbol{x}\right)}{p_{\tilde{\boldsymbol{y}}_s|\boldsymbol{\mathcal{X}}}\left(\tilde{\boldsymbol{y}}_s|\boldsymbol{\mathcal{X}}\right)}\right] \\
&= \mathbb{E}_{\boldsymbol{x}\sim p(\boldsymbol{x})}\mathbb{E}_{\tilde{\boldsymbol{y}}|\boldsymbol{x}\sim q}\left[\log\frac{q\left(\tilde{\boldsymbol{y}}|\boldsymbol{x}\right)}{p_{\boldsymbol{\mathcal{X}},\tilde{\boldsymbol{y}}_s}\left(\boldsymbol{\mathcal{X}},\tilde{\boldsymbol{y}}_s\right)/p_{\boldsymbol{\mathcal{X}}}\left(\boldsymbol{\mathcal{X}}\right)}\right] \\
&= \mathbb{E}_{\boldsymbol{x}\sim p(\boldsymbol{x})}\mathbb{E}_{\tilde{\boldsymbol{y}}|\boldsymbol{x}\sim q}\left[\log\frac{q\left(\tilde{\boldsymbol{y}}|\boldsymbol{x}\right)}{p_{\boldsymbol{\mathcal{X}}|\tilde{\boldsymbol{y}}_s}\left(\boldsymbol{\mathcal{X}}|\tilde{\boldsymbol{y}}_s\right)\cdot p_{\tilde{\boldsymbol{y}}_s}\left(\tilde{\boldsymbol{y}}_s\right)/p_{\boldsymbol{\mathcal{X}}}\left(\boldsymbol{\mathcal{X}}\right)}\right] \\
&= \mathbb{E}_{\boldsymbol{x}\sim p(\boldsymbol{x})}\mathbb{E}_{\tilde{\boldsymbol{y}}|\boldsymbol{x}\sim q}\left[\log q\left(\tilde{\boldsymbol{y}}|\boldsymbol{x}\right) - \log p_{\boldsymbol{\mathcal{X}}|\tilde{\boldsymbol{y}}_s}\left(\boldsymbol{\mathcal{X}}|\tilde{\boldsymbol{y}}_s\right) - \log p_{\tilde{\boldsymbol{y}}_s}\left(\tilde{\boldsymbol{y}}_s\right) + \log p_{\boldsymbol{\mathcal{X}}}\left(\boldsymbol{\mathcal{X}}\right)\right]
\end{aligned}
\tag{32}
$$

Next, We examine the result of (32) term by term.

1. For the first term $\log q\left(\tilde{\boldsymbol{y}}|\boldsymbol{x}\right)$, since the noisy latent representation $\tilde{\boldsymbol{y}}$ can be separated into two parts, i.e., a synonymous representation $\tilde{\boldsymbol{y}}_s$ and a detailed representation $\tilde{\boldsymbol{y}}_\epsilon$, it can be expanded to

$$
\log q\left(\tilde{\boldsymbol{y}}|\boldsymbol{x};\boldsymbol{\phi}_g\right) = \log q\left(\tilde{\boldsymbol{y}}_s,\tilde{\boldsymbol{y}}_\epsilon|\boldsymbol{x};\boldsymbol{\phi}_g\right) = \log q\left(\tilde{\boldsymbol{y}}_s|\boldsymbol{x};\boldsymbol{\phi}_g\right) + \log q\left(\tilde{\boldsymbol{y}}_\epsilon|\boldsymbol{x},\tilde{\boldsymbol{y}}_s;\boldsymbol{\phi}_g\right).
\tag{33}
$$

Since both $\tilde{\boldsymbol{y}}_s$ and $\tilde{\boldsymbol{y}}_\epsilon$ are determined based on a parametric inference model $g_a\left(\boldsymbol{x};\boldsymbol{\phi}_g\right)$ and uniform density on the unit interval centered on $\boldsymbol{y}_s$ and $\boldsymbol{y}_\epsilon$, this term equals a constant 0.

2. For the second term $-\log p_{\boldsymbol{\mathcal{X}}|\tilde{\boldsymbol{y}}_s}\left(\boldsymbol{\mathcal{X}}|\tilde{\boldsymbol{y}}_s\right)$, with the outside expectations $\mathbb{E}_{\boldsymbol{x}\sim p(\boldsymbol{x})}\mathbb{E}_{\tilde{\boldsymbol{y}}|\boldsymbol{x}\sim q}$, the minimization of this term can be equivalent to minimizing an weighted expected distortion $\mathbb{E}_{\boldsymbol{x}\sim p(\boldsymbol{x})}\mathbb{E}_{\tilde{\boldsymbol{y}}\sim q}\mathbb{E}_{\tilde{\boldsymbol{x}}_i\in\tilde{\boldsymbol{\mathcal{X}}}|\tilde{\boldsymbol{y}}_s}\left[d\left(\boldsymbol{x},\tilde{\boldsymbol{x}}_i\right)\right]$ plus an weighted E-KLD term $\mathbb{E}_{\tilde{\boldsymbol{y}}\sim q}\mathbb{E}_{\tilde{\boldsymbol{x}}_i\in\tilde{\boldsymbol{\mathcal{X}}}|\tilde{\boldsymbol{y}}_s}D_{\mathrm{KL}}\left[p_{\boldsymbol{x}}||p_{\tilde{\boldsymbol{x}}_i}\right]$, by Lemma 3.2.

3. For the third term $-\log p_{\tilde{\boldsymbol{y}}_s}\left(\tilde{\boldsymbol{y}}_s\right)$, it is the coding rate of the synonymous representations. With the outside expectations, it is also equivalent to the semantic entropy of semantic variable $\tilde{\tilde{Y}}$ corresponding to a latent synset $\tilde{\boldsymbol{\mathcal{Y}}}$. This can be derived by

$$
\begin{aligned}
\mathbb{E}_{\boldsymbol{x}\sim p(\boldsymbol{x})}\mathbb{E}_{\tilde{\boldsymbol{y}}|\boldsymbol{x}\sim q}\left[-\log p_{\tilde{\boldsymbol{y}}_s}\left(\tilde{\boldsymbol{y}}_s\right)\right] &= \mathbb{E}_{\boldsymbol{x}\sim p(\boldsymbol{x})}\mathbb{E}_{\tilde{\boldsymbol{y}}|\boldsymbol{x}\sim q}\left[-\log\int_{\tilde{\boldsymbol{y}}_\epsilon}p_{\tilde{\boldsymbol{y}}_s}\left(\tilde{\boldsymbol{y}}_s,\tilde{\boldsymbol{y}}_\epsilon\right)d\tilde{\boldsymbol{y}}_\epsilon\right] \\
&= \mathbb{E}_{\boldsymbol{x}\sim p(\boldsymbol{x})}\mathbb{E}_{\tilde{\boldsymbol{y}}|\boldsymbol{x}\sim q}\left[-\log\int_{\tilde{\boldsymbol{y}}\in\tilde{\boldsymbol{\mathcal{Y}}}}p_{\tilde{\boldsymbol{y}}}\left(\tilde{\boldsymbol{y}}\right)d\tilde{\boldsymbol{y}}\right] \\
&\overset{(a)}{=} H_s\left(\tilde{\tilde{Y}}\right),
\end{aligned}
\tag{34}
$$

in which $(a)$ is achieved based on the definition of semantic entropy in Niu and Zhang's paper (2024), with the help of the weak law of large numbers. Additionally, considering the determined codec, the semantic entropy $H_s\left(\tilde{\tilde{Y}}\right)$ is equivalent to single-side semantic mutual information, i.e.,

$$
H_s\left(\tilde{\tilde{Y}}\right) \overset{(a)}{=} H_s\left(\tilde{\tilde{X}}\right) \overset{(b)}{=} H_s\left(\tilde{\tilde{X}}\right) - H_s\left(\tilde{\tilde{X}}|X\right) \overset{(c)}{=} I\left(X;\tilde{\tilde{X}}\right),
\tag{35}
$$

where $(a)$ is by giving a determined decoder to map the latent synset $\tilde{\boldsymbol{\mathcal{Y}}}$ to the reconstructed synset $\tilde{\boldsymbol{\mathcal{X}}}$; $(b)$ is achieved by a determined encoder, which makes $\log q\left(\tilde{\boldsymbol{\mathcal{X}}}|\boldsymbol{x}\right) = \log q\left(\tilde{\boldsymbol{y}}_s|\boldsymbol{x}\right) = 0$ with a uniform density on the unit interval centered on $\boldsymbol{y}_s$; $(c)$ is by the definition of this single-side semantic mutual information.

4. For the fourth term $-\log p_{\mathcal{X}}(\mathcal{X})$, with the determination of the ideal synset $\mathcal{X}$ for the original image $x$, it is a constant that cannot be optimized.

To summarize, the minimization of (32) is equivalent to the following optimization directions

$$
\begin{aligned}
\mathcal{L}_{\mathcal{X}} = {}& \lambda_d \cdot \mathbb{E}_{x \sim p(x)} \mathbb{E}_{\tilde{\boldsymbol{y}} \sim q} \mathbb{E}_{\tilde{\boldsymbol{x}}_i \in \tilde{\mathcal{X}} | \tilde{\boldsymbol{y}}_s} \left[ d\left(\boldsymbol{x}, \tilde{\boldsymbol{x}}_i\right) \right] + \lambda_p \cdot \mathbb{E}_{\tilde{\boldsymbol{y}} \sim q} \mathbb{E}_{\tilde{\boldsymbol{x}}_i \in \tilde{\mathcal{X}} | \tilde{\boldsymbol{y}}_s} D_{\mathrm{KL}} \left[ p_{\boldsymbol{x}} || p_{\tilde{\boldsymbol{x}}_i} \right] \\
& + \mathbb{E}_{x \sim p(x)} \mathbb{E}_{\tilde{\boldsymbol{y}} \sim q} \left[ -\log p_{\tilde{\boldsymbol{y}}_s} \left(\tilde{\boldsymbol{y}}_s\right) \right].
\end{aligned}
\tag{36}
$$

At the convergence point, when the minimum value of the loss function is achieved, the model effectively minimizes the single-side semantic mutual information with the quantized form of $\hat{\tilde{X}}$ under the constraints of bounded quantized expected distortion and E-KLD, as follows in (31). The weights $\lambda_d$ and $\lambda_p$ can be considered as Lagrange multipliers to the rate term. So we conclude this theorem. $\qquad\square$

From the proof of Lemma 3.2 and Theorem 3.3 above, we can see that the differences between the proposed SVI and conventional variational inference in guiding image compression tasks mainly include:

- ***The Analysis Method for the Likelihood Term:*** Conventional variational inference treats the likelihood term $-\log(\boldsymbol{x}|\tilde{\boldsymbol{y}})$ in usual LIC methods as a weighted distortion (Ballé et al., 2017; Blau & Michaeli, 2018), primarily because the synonymous relationship emphasized in this paper is not incorporated in these works.

  In our proposed SVI, since the synonymous relationship is considered, the likelihood term primarily focuses on the mapping relationship between the latent synset and the ideal synset, which is formed by $-\log p_{\mathcal{X}|\tilde{\boldsymbol{y}}_s}(\mathcal{X}|\tilde{\boldsymbol{y}}_s)$. Lemma 3.2 states that the minimization of the expected synonymous likelihood term is equivalent to the minimization of a tradeoff function with weighted expected distortion and weighted E-KLD term.

  Additionally, the analytical process of Lemma 3.2 emphasizes that **the fundamental reason for the expected KL divergence term's existence is due to the consideration of the ideal synset $\mathcal{X}$ centered by the original image $x$.** Once the ideal synset is unconsidered (equal to the ideal synset only contains the original image, i.e., $\mathcal{X} = \{\boldsymbol{x}\}$), the expected KL divergence term will disappear in the analysis result, which means the degradation to solely the weighted distortion term. Therefore, we can give the following statement:

  **To the best knowledge of our authors, our method is the first work that can theoretically explain the fundamental reason for the divergence measure's existence in perceptual image compression, which stems from considering the ideal synset as the reconstruction reference.**

- ***The Considerations on Coding Rates:*** Conventional variational inference considers performing entropy coding for all latent representations, whose coding rate can be represented as $-\log p(\tilde{\boldsymbol{y}})$.

  In our proposed SVI, only a partial of the latent representation is required to be encoded, whose coding rate is expressed as $-\log p(\tilde{\boldsymbol{y}}_s)$. We emphasize that the other partial, i.e., the detailed representation, is not required to be coded: it can also be sampled from the latent synset $\tilde{\mathcal{Y}}$ by the decoder, which is not necessary to keep consistency with it at the encoder end. This stems from the change in optimization direction from the original sample-oriented to the ideal synset-oriented, which allows the decoder to generate any samples that can exist in the ideal synset.

  Since the detailed representation can be obtained by sampling at the receiving end and is allowed to contribute effective information for image reconstruction, encoding only the synonymous representation part theoretically improves coding efficiency compared to traditional methods. From the perspective of mutual information, this can be expressed as $I\left(\boldsymbol{X}; \hat{\tilde{\boldsymbol{X}}}\right) \leq I\left(\boldsymbol{X}; \hat{\boldsymbol{X}}\right)$, in which the condition for the inequality to hold as equality is that the reconstruction of the synonym set is restricted to producing only one sample, meaning the decoder is not allowed to sample $\hat{\boldsymbol{y}}_{\epsilon,j}$, nor is it permitted to use $\hat{\boldsymbol{y}}_{\epsilon,j}$ as input to the generator.

### A.3. Discussions on the Relationships with Existing Image Compression Theories

From Lemma 3.2, Theorem 3.3, and their respective proof processes, it is evident that the optimization objective derived in this paper through synonymous variational inference is compatible with the optimization objectives of existing image compression theories:

- **Compatibility with Existing Rate-Distortion-Perception Tradeoff:** According to the triple tradeoff shown as (31), when the reconstructed synset is not considered (equal to the reconstructed synset contains only one sample, represented as $\hat{\mathcal{X}} = \{\hat{x}\}$), the optimization objective will be degraded into the existing rate-distortion-perception tradeoff. This relationship can be expressed by

$$R(\mathcal{X}) = \min_{p(\hat{\mathcal{X}}|\boldsymbol{x})} I\left(\boldsymbol{X}; \hat{\mathring{\boldsymbol{X}}}\right) \qquad\qquad R(D,P) = \min_{p(\hat{\boldsymbol{x}}|\boldsymbol{x})} I\left(\boldsymbol{X}; \hat{\boldsymbol{X}}\right)$$
$$\text{s.t.} \quad \mathbb{E}_{\boldsymbol{x}\sim p(\boldsymbol{x})}\mathbb{E}_{\hat{\boldsymbol{x}}_i \in \hat{\mathcal{X}}|\hat{\boldsymbol{y}}_s}\left[d\left(\boldsymbol{x},\hat{\boldsymbol{x}}_i\right)\right] \leq D, \quad \xRightarrow{\hat{\mathcal{X}}=\{\hat{x}\}} \quad \text{s.t.} \quad \mathbb{E}_{\boldsymbol{x}\sim p(\boldsymbol{x})}\left[d\left(\boldsymbol{x},\hat{\boldsymbol{x}}\right)\right] \leq D, \tag{37}$$
$$\mathbb{E}_{\hat{\boldsymbol{x}}_i \in \hat{\mathcal{X}}|\hat{\boldsymbol{y}}_s} D_{\text{KL}}\left[p_{\boldsymbol{x}}||p_{\hat{\boldsymbol{x}}_i}\right] \leq P, \qquad\qquad D_{\text{KL}}\left[p_{\boldsymbol{x}}||p_{\hat{\boldsymbol{x}}}\right] \leq P,$$

in which the KL divergence $D_{\text{KL}}\left[p_{\boldsymbol{x}}||p_{\hat{\boldsymbol{x}}}\right]$ is a typical measure of the divergence between distributions, i.e., $d_p\left(p_{\boldsymbol{x}}, p_{\hat{\boldsymbol{x}}}\right)$ in (3), as stated in (Blau & Michaeli, 2019). In view of this, the existing rate-distortion-perception tradeoff (4) is a special case of (11) when there is only one sample $\hat{x}$ in the reconstructed synset $\hat{\mathcal{X}}$.

- **Compatibility with Traditional Rate-Distortion Tradeoff:** Based on the analytical process of Lemma 3.2, when the ideal synset is not considered (equal to the ideal synset contains only the original image, represented as $\mathcal{X} = \{\boldsymbol{x}\}$), the expected synonymous likelihood term will be degraded into the usual likelihood term, i.e.,

$$\mathbb{E}_{\boldsymbol{x}\sim p(\boldsymbol{x})}\mathbb{E}_{\tilde{\boldsymbol{y}}\sim q}\left[-\log p_{\mathcal{X}|\tilde{\boldsymbol{y}}_s}\left(\mathcal{X}|\tilde{\boldsymbol{y}}_s\right)\right] \xRightarrow{\mathcal{X}=\{\boldsymbol{x}\}} \mathbb{E}_{\boldsymbol{x}\sim p(\boldsymbol{x})}\left[-\log p_{\boldsymbol{x}|\tilde{\boldsymbol{y}}}\left(\boldsymbol{x}|\tilde{\boldsymbol{y}}\right)\right], \tag{38}$$

in which the minimization of the usual likelihood term $\mathbb{E}_{\boldsymbol{x}\sim p(\boldsymbol{x})}\mathbb{E}_{\tilde{\boldsymbol{y}}\sim q}\left[-\log p_{\boldsymbol{x}|\tilde{\boldsymbol{y}}}\left(\boldsymbol{x}|\tilde{\boldsymbol{y}}\right)\right]$ is equivalent to the minimization of a weighted distortion $\lambda \mathbb{E}_{\boldsymbol{x}\sim p(\boldsymbol{x})}\mathbb{E}_{\tilde{\boldsymbol{y}}\sim q}\left[d\left(\boldsymbol{x},\tilde{\boldsymbol{x}}\right)\right]$, makes the existence of the KL divergence term unnecessary. Additionally, $\mathcal{X} = \{\boldsymbol{x}\}$ will also implicitly makes the existence of $\hat{\boldsymbol{y}}_{\epsilon,j}$ unnecessary, which results in $\hat{\mathcal{X}} = \{\hat{\boldsymbol{x}}\}$. Therefore, the relationship with the traditional rate-distortion tradeoff can be represented by

$$R(\mathcal{X}) = \min_{p(\hat{\mathcal{X}}|\boldsymbol{x})} I\left(\boldsymbol{X}; \hat{\mathring{\boldsymbol{X}}}\right) \qquad\qquad R(D) = \min_{p(\hat{\boldsymbol{x}}|\boldsymbol{x})} I\left(\boldsymbol{X}; \hat{\boldsymbol{X}}\right)$$
$$\text{s.t.} \quad \mathbb{E}_{\boldsymbol{x}\sim p(\boldsymbol{x})}\mathbb{E}_{\hat{\boldsymbol{x}}_i \in \hat{\mathcal{X}}|\hat{\boldsymbol{y}}_s}\left[d\left(\boldsymbol{x},\hat{\boldsymbol{x}}_i\right)\right] \leq D, \quad \overset{\mathcal{X}=\{\boldsymbol{x}\}}{\underset{(\hat{\mathcal{X}}=\{\hat{x}\})}{\Longrightarrow}} \quad \text{s.t.} \quad \mathbb{E}_{\boldsymbol{x}\sim p(\boldsymbol{x})}\left[d\left(\boldsymbol{x},\hat{\boldsymbol{x}}\right)\right] \leq D. \tag{39}$$
$$\mathbb{E}_{\hat{\boldsymbol{x}}_i \in \hat{\mathcal{X}}|\hat{\boldsymbol{y}}_s} D_{\text{KL}}\left[p_{\boldsymbol{x}}||p_{\hat{\boldsymbol{x}}_i}\right] \leq P,$$

In view of this, we state that the traditional rate-distortion tradeoff is also a special case of (11), where the condition is that the ideal synset $\mathcal{X}$ always contains only a single sample, i.e., the original $\boldsymbol{x}$.

Therefore, we demonstrate that the foundational theories underlying existing image compression methods can be viewed as special cases of our analysis viewpoint. In other words, our theoretical analysis provides consistent and universal guidance for designing image compression approaches, regardless of whether compression is considered for perceptual quality.

It should be noted that, although the E-KLD term in the above analysis represents the optimal distribution distance selection based on minimizing the partial semantic KL divergence, accurately calculating the E-KLD between two image sets may be unrealistic due to the complexity of the image source. Therefore, we can refer to existing empirical practices in perceptual image compression, such as using metrics that tend to be subjective, like LPIPS or DISTS, instead of the KL divergence calculation, or using adversarial losses in the GAN training process, such as Wasserstein loss, as a substitute for KL divergence.

## B. Relevant Thoughts on Semantic Information Theory

In this appendix section, we will briefly provide relevant thoughts on semantic information theory based on our analytical results. We emphasize that the semantic information theory referenced here specifically pertains to the synonymity-based semantic information theory, primarily derived from Niu and Zhang's paper (Niu & Zhang, 2024).

### B.1. Relationships with Existing Conclusions in Semantic Information Theory

- **Semantic Entropy:** Given a semantic variable $\mathring{U}$, whose possible values are the synset $\mathcal{U}_{i_s} = \{u_i | i \in \mathcal{N}_{i_s}\}$, , $i_s = 1, 2, ..., N$, and $\mathcal{N}_{i_s}$ is the set of ordinal numbers of all syntactic symbols $u_i \in \mathcal{U}_{i_s}$, the semantic entropy of the semantic

variable is expressed as:

$$H_s\left(\mathring{U}\right) = -\sum_{i_s=1}^{N} p\left(\mathcal{U}_{i_s}\right) \log p\left(\mathcal{U}_{i_s}\right) = -\sum_{i_s=1}^{N} \sum_{i\in\mathcal{N}_{i_s}} p\left(u_i\right) \log\left(\sum_{i\in\mathcal{N}_{i_s}} p\left(u_i\right)\right),$$  (40)

in which the probability a single semantic symbol corresponds to the synset $\mathcal{U}_{i_s}$ is the sum of the probability of each syntactic value $u_i, i\in\mathcal{N}_{i_s}$, i.e., $p\left(\mathcal{U}_{i_s}\right) = \sum_{i\in\mathcal{N}_{i_s}} p\left(u_i\right)$.

Additionally, the semantic source coding theorem in (Niu & Zhang, 2024) states that, for a semantic source $\mathring{U}$ with its corresponding syntactic source $U$ and its determined synonymous mapping between these two types of sources (which results in the determined synsets $\mathcal{U}_{i_s} = \{u_i|i\in\mathcal{N}_{i_s}\}$, $i_s = 1, 2, ..., N$), the achievable coding rate for semantic lossless rate is $R \geq H_s\left(\mathring{U}\right)$ without the necessity to focus on symbol-level accuracy.

In our work, we point out that the ultimate goal of synonymous variational inference is to construct an ideal synset $\mathcal{X}$ corresponding to the original image $x$ by finding a synonymous mapping rule and encoding the shared latent features of the ideal synset $\hat{y}_s$ as the coding sequences. In the ideal scenario where model training has converged, the reconstructed synset $\hat{\mathcal{X}}$ at the SIC decoder can perfectly overlap with the ideal synset $\mathcal{X}$. Under such conditions, the average coding rate of the synonymous representation $\hat{y}_s$ in the latent space can approach the semantic entropy of the semantic variable corresponding to the ideal synset, i.e.,

$$\mathbb{E}_{\boldsymbol{x}\sim p(\boldsymbol{x})}\left[-\log p\left(\hat{\boldsymbol{y}}_s\right)\right] \overset{(a)}{=} H_s\left(\hat{\mathring{\boldsymbol{Y}}}\right) \overset{(b)}{=} H_s\left(\hat{\mathring{\boldsymbol{X}}}\right) \overset{(c)}{=} I\left(\boldsymbol{X}; \hat{\mathring{\boldsymbol{X}}}\right) \overset{(d)}{=} H_s\left(\mathring{\boldsymbol{X}}\right),$$  (41)

in which the established conditions of (a) $\sim$ (c) is the same as the conditions in (34) and (35), and (d) is achieved by the ideal SIC codec. At this point, the found synonymous mapping rule is determined by the bounded expected distortion and the bounded E-KLD, i.e., $\mathbb{E}_{\boldsymbol{x}\sim p(\boldsymbol{x})}\mathbb{E}_{\hat{\boldsymbol{x}}_i\in\hat{\mathcal{X}}|\hat{\boldsymbol{y}}_s}\left[d\left(\boldsymbol{x}, \hat{\boldsymbol{x}}_i\right)\right] \leq D$, $\mathbb{E}_{\hat{\boldsymbol{x}}_i\in\hat{\mathcal{X}}|\hat{\boldsymbol{y}}_s} D_{\mathrm{KL}}\left[p_{\boldsymbol{x}}||p_{\hat{\boldsymbol{x}}_i}\right] \leq P$.

- **Down Semantic Mutual Information:** The down semantic mutual information is a measure defined as

$$I_s\left(\mathring{U}; \mathring{V}\right) = H_s\left(\mathring{U}\right) + H_s\left(\mathring{V}\right) - H\left(U, V\right),$$  (42)

which is proved to be the minimum coding rate in semantic lossy source coding when the "semantic distortion" (denoted as $d_s\left(\mathring{X}, \hat{\mathring{X}}\right)$ in concept) satisfying $d_s\left(\mathring{X}, \hat{\mathring{X}}\right) \leq D_s$, using an extended joint asymptotic equipartition property (AEP) analysis called *Semantically Joint AEP*.

In our work, since the ideal synset is not explicitly constructed in practice, directly determining the overlap degree between the reconstructed synset and the ideal synset is challenging. Consequently, the model obtained after training convergence may still function as a semantic lossy coding model. We propose that the coding rate of the SIC model should also serve as an upper bound for the lower semantic mutual information, expressed as:

$$\begin{aligned}\mathbb{E}_{\boldsymbol{x}\sim p(\boldsymbol{x})}\left[-\log p\left(\hat{\boldsymbol{y}}_s\right)\right] &\overset{(a)}{=} H_s\left(\hat{\mathring{\boldsymbol{Y}}}\right) \overset{(b)}{=} H_s\left(\hat{\mathring{\boldsymbol{X}}}\right) \overset{(c)}{=} H_s\left(\hat{\mathring{\boldsymbol{X}}}\right) - H_s\left(\hat{\mathring{\boldsymbol{X}}}|\boldsymbol{X}\right) \\ &\overset{(d)}{=} H\left(\boldsymbol{X}\right) + H_s\left(\hat{\mathring{\boldsymbol{X}}}\right) - H_s\left(\boldsymbol{X}, \hat{\mathring{\boldsymbol{X}}}\right) \\ &\overset{(e)}{\geq} H_s\left(\mathring{\boldsymbol{X}}\right) + H_s\left(\hat{\mathring{\boldsymbol{X}}}\right) - H\left(\boldsymbol{X}, \hat{\boldsymbol{X}}\right) \\ &= I_s\left(\mathring{\boldsymbol{X}}; \hat{\mathring{\boldsymbol{X}}}\right),\end{aligned}$$  (43)

in which the established conditions of (a) $\sim$ (c) is the same as the conditions in (34) and (35), (d) is achieved by the equation $H_s\left(\hat{\mathring{\boldsymbol{X}}}|\boldsymbol{X}\right) = H_s\left(\boldsymbol{X}, \hat{\mathring{\boldsymbol{X}}}\right) - H\left(\boldsymbol{X}\right)$, and (e) can be proved by a scaling process according to $H\left(\boldsymbol{X}\right) \geq H_s\left(\mathring{\boldsymbol{X}}\right)$ and $H_s\left(\boldsymbol{X}, \hat{\mathring{\boldsymbol{X}}}\right) \leq H\left(\boldsymbol{X}, \hat{\boldsymbol{X}}\right)$, which follows the fundamental properties of semantic variables stated in (Niu & Zhang, 2024).

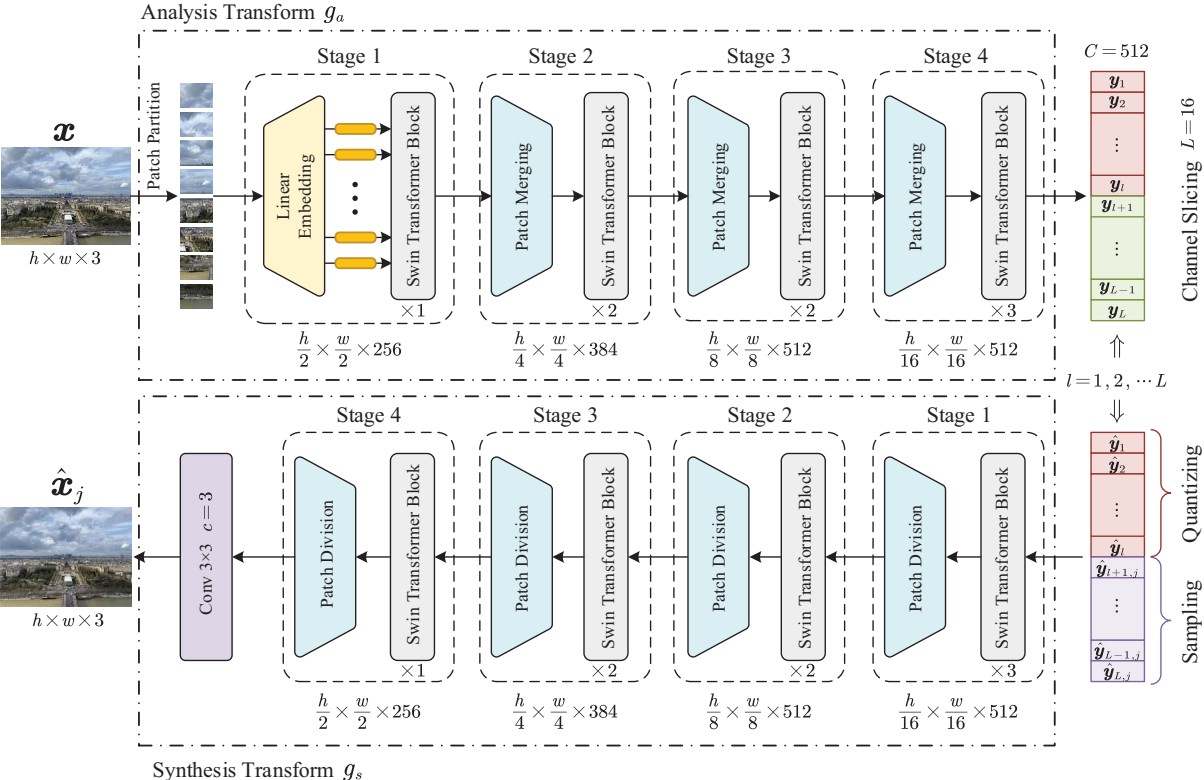

*Figure 7.* The implementation details of the auto-encoder framework designed for the progressive SIC model.

## B.2. Extended Thinking: About Synonymous Idempotence Constraints

As discussed earlier, since the ideal synset is not explicitly constructed, directly determining the overlap between the reconstructed synset and the ideal synset is infeasible. However, since the ideal synset directly corresponds to the synonymous representation $\hat{\boldsymbol{y}}_s$ in the latent space, the distance between the two synsets in the data space can be evaluated by first re-encoding the samples from the reconstructed synset, then obtaining the new synonymous representation $\hat{\boldsymbol{y}}'_s$, and finally calculating the distance between $\hat{\boldsymbol{y}}_s$ and $\hat{\boldsymbol{y}}'_s$. This idea can be directly integrated into the training process. By incorporating constraints into the loss function, the SIC codec can be explicitly guided to optimize toward maximizing the overlap between the reconstructed synset and the ideal synset during the optimization process, i.e.,

$$\mathcal{L}_c^{\text{synset}} = \left\| \hat{\boldsymbol{y}}'_s - \hat{\boldsymbol{y}}_s \right\|^2, \tag{44}$$

in which the adopted distance measure is based on the MSE function in our model optimization.

Additionally, if sufficient diversity among the samples in the reconstructed synset should be ensured, the following constraints can be incorporated into the loss function:

$$\mathcal{L}_c^{\text{detail}} = \left\| \hat{\boldsymbol{y}}'_{\epsilon,j} - \hat{\boldsymbol{y}}_{\epsilon,j} \right\|^2. \tag{45}$$

It refers to re-feeding the reconstructed image into the encoder, computing the difference between the new detailed representation $\hat{\boldsymbol{y}}'_{\epsilon,j}$ and the original detail representation $\hat{\boldsymbol{y}}_{\epsilon,j}$, and incorporating it into the optimization process of the SIC codec.

When both of the above constraints are equal to 0, the idempotence property of the reconstructed image samples is effectively satisfied (in which idempotence "refers to the stability of image code to re-compression", as stated in Xu's paper (2024)). Here, we term the above constraint as the synonymous idempotence constraint and incorporate it into the loss function for the neural SIC model implemented in this paper. Please refer to Appendix C.1 for specific training details.

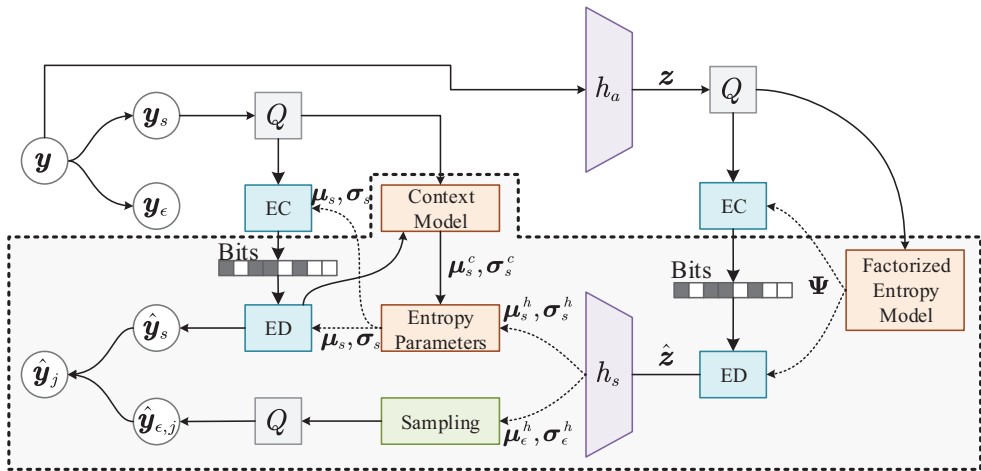

*Figure 8.* The joint autoregressive and hierarchical prior architecture of the progressive SIC model.

## C. Experimental Illustrations: Implementations and Supplementary Results

In this section, we provide implementation details of the progressive SIC model and supplementary results for Section 5.

### C.1. Implementation details

The auto-encoder architecture, including an analysis transform as the encoder and a synthesis transform as the decoder, is implemented based on the Swin Transformer (Liu et al., 2021). The implementation details are shown in Figure 7. Both the analysis transform and the synthesis transform perform a 4-stage nonlinear processing, in which each layer includes a dimension adjustment module (Linear Embedding, Patch Merging, and Patch Division) and a Swin Transformer module.

For the analysis transform, the image $x$ with the resolution of $h \times w \times 3$ is first partitioned into patches with the resolution of $2 \times 2$. Then, the patches are fed into the Linear Embedding module at stage 1 for dimension expansion, and the features are further extracted using a followed Swin Transformer Block. The following processing stages all use the Patch Merging block and several Swin Transformer blocks to extract deeper features. Finally, the analysis transform outputs a feature map with a dimension of $\frac{h}{16} \times \frac{w}{16} \times 512$, in which the channel dimension $C = 512$.

To support multiple synonymous level partitioning, we perform equal slicing along the channel dimension of the latent representation. We set the number of the synonymous levels $L = 16$, thus the latent representations are partitioned along the channel dimension into 16 groups. Each synonymous group contains a sub-feature map with the size of $\frac{h}{16} \times \frac{w}{16} \times 32$. When the $l$-th synonymous level is selected, the synonymous representations $y_s$ contains the first $l$ groups of the sub-feature map with the full size of $\frac{h}{16} \times \frac{w}{16} \times 32l$, thus makes the remaining levels serve as detailed representations $y_\epsilon$.

For the synthesis transform, the quantized synonymous representations $\hat{y}_s$ and sampled detailed representations $\hat{y}_\epsilon$ are as input. Then, four upsampled Swin Transformer stages and a final convolutional layer are applied to the input to integrate the global information of the image, in which each stage increases the input resolution through the corresponding number of Swin Transformer modules in the analysis transform and a Patch Division module. Finally, a convolutional layer outputs the reconstructed image $\hat{y}_j$.

For rate estimation, the progressive SIC model adopts a joint autoregressive and hierarchical prior architecture based on (Minnen et al., 2018) as shown in Figure 8. All these modules in this architecture are implemented based on convolutional neural networks represented by Figure 9, in which $Q$ represents the quantization model using the round function.

In this architecture, the hyperprior $h_a$ and $h_s$ are performed for both the distribution estimation for the synonymous representation $\hat{y}_s$ and the sampling for the detailed representation. The hyperprior performs the "forward adaptation" to estimate $\mu_s^h, \sigma_s^h$ and $\mu_\epsilon^h, \sigma_\epsilon^h$ based on the side information $\hat{z}$, while the context model performs the "backward adaptation" to estimate $\mu_s^c, \sigma_s^c$ based on the already-coded synonymous representations, in which the expressions "forward adaptation"

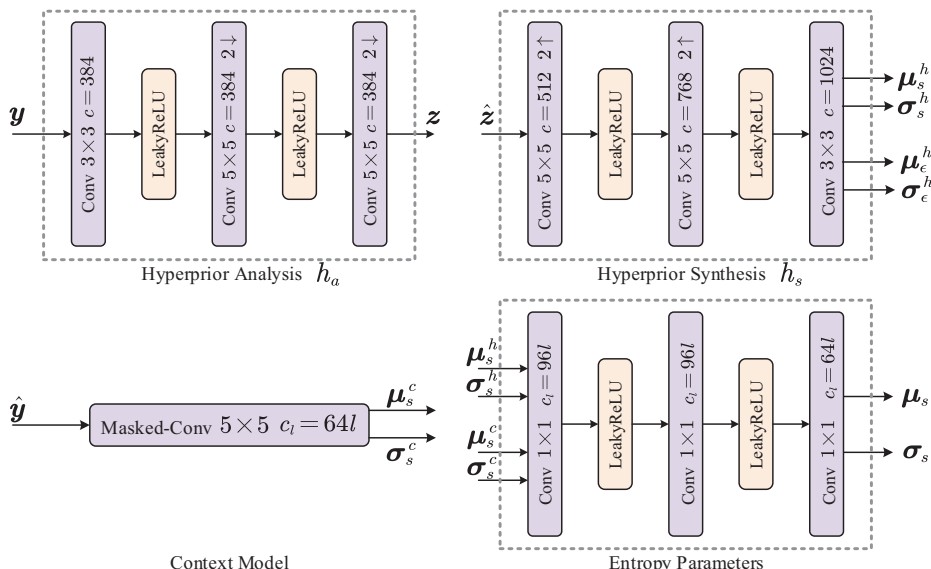

*Figure 9.* The implementation details of the modules in the joint autoregressive and hierarchical prior architecture.

and "backward adaptation" are stated in (Ballé et al., 2020). For the synonymous representations $\hat{\boldsymbol{y}}_s$, an Entropy Parameters modules are employed to integrate the input $\boldsymbol{\mu}_s^h, \boldsymbol{\sigma}_s^h$ and $\boldsymbol{\mu}_s^c, \boldsymbol{\sigma}_s^c$ to an output accurate estimation $\boldsymbol{\mu}_s, \boldsymbol{\sigma}_s$. And for the detailed representation $\hat{\boldsymbol{y}}_\epsilon$, a uniform sampling based on the following equation is utilized:

$$\hat{\boldsymbol{y}}_{\epsilon,j} = Q\left(\boldsymbol{\mu}_\epsilon^h + \boldsymbol{\mathcal{U}}\left(-2, 2\right)\right), \tag{46}$$

in which the uniform distribution $\boldsymbol{\mathcal{U}}\left(-2, 2\right)$ is set empirically. We realize that this sampling method cannot fit the ideal conditional distribution **in vector form**: **The SVI theoretical analysis provides an ideal detail sampling principle that follows a conditional distribution** $p_{\hat{\boldsymbol{y}}_{\epsilon,j}|\tilde{\boldsymbol{y}}_s}$ **in vector form (stated in the Equation (18)), which guides the prediction of the samples vector** $\hat{\boldsymbol{y}}_{\epsilon,j}$ **conditioned on synonymous representations** $\tilde{\boldsymbol{y}}_s$. However, **the current unideal sampling mechanism cannot ensure reasonable contextual structure in the details of the reconstructed images**, which may affect the distribution consistency focused by measures like FID. We argue that sampling the detailed representation to match the ideal conditional distribution in vector form is challenging, especially when multiple distinct samples (i.e., $M > 1$). Although we are still exploring effective solutions to this problem, adopting a simple yet suboptimal sampling method is currently a necessary compromise. This will be a key breakthrough direction for our future research.

To support the multiple synonymous level partitioning mechanism, we modify the masked convolutional layer of the spatial context autoregressive model in (Minnen et al., 2018) to a spatial-channel context autoregressive model like (Li et al., 2020), in which the 3D mask for the masked Context Model is presented by Figure 10. The core mechanism is to estimate the current feature's probability distribution by conditioning on encoded spatial and channel features. This spatial-channel context autoregressive module is implemented based on a $5 \times 5$ convolutional layer, in which the spatial context autoregressive process within the single group of sub-latent feature map $c_k$ is achieved by the left matrix shown in Figure 10, and the channel context autoregressive process from the coded groups of sub-latent feature map to the current group is achieved by the full 1 matrix shown as the right matrix in Figure 10.

Additionally, for the Entropy Parameters module, all the estimations of $\boldsymbol{\mu}_s$ and $\boldsymbol{\sigma}_s$ across diverse synonymous levels share the same module. To achieve this, each convolutional layer in the Entropy Parameters module assigns a group parameter of $l$, enabling the layer to process $l$ independent groups in parallel. These groups perform the estimation processes separately for each sub-feature map, and their outputs are concatenated to form $\boldsymbol{\mu}_s$ and $\boldsymbol{\sigma}_s$.

Before model training, hyperparameter values for the loss function (12) and (13) must be specified before model training. Table 1 presents these hyperparameters of our progressive SIC model, which are configured empirically.

Since multiple sub-feature maps are partitioned in the channel dimension to build different synonymous levels $l =$

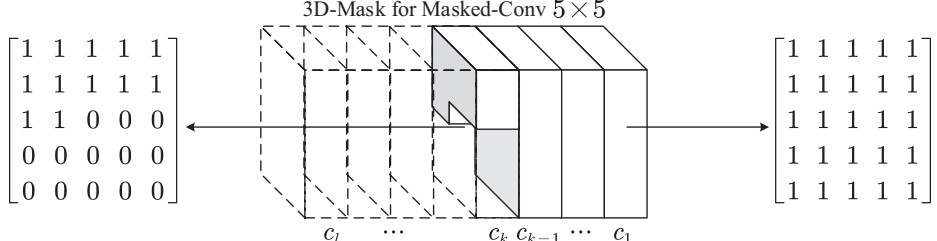

Figure 10. The 3D mask for the masked convolutional layer in the context model.

Table 1. Hyperparameters Configurations for progressive SIC model training.

| $l$ | 1 | 2 | 3 | 4 | 5 | 6 | 7 | 8 |
|---|---|---|---|---|---|---|---|---|
| $\alpha$ | | | | 0.5 | | | | |
| $\lambda_r^{(l)}$ | 128 | 256 | 384 | 512 | 640 | 768 | 896 | 1024 |
| $\lambda_d^{(l)}$ | $2^{39/8}$ | $2^{38/8}$ | $2^{37/8}$ | $2^{36/8}$ | $2^{35/8}$ | $2^{34/8}$ | $2^{33/8}$ | $2^{32/8}$ |
| $\lambda_p^{(l)}$ | $2^{45/8}$ | $2^{42/8}$ | $2^{39/8}$ | $2^{36/8}$ | $2^{33/8}$ | $2^{30/8}$ | $2^{27/8}$ | $2^{24/8}$ |
| $l$ | 9 | 10 | 11 | 12 | 13 | 14 | 15 | 16 |
| $\alpha$ | | | | 0.5 | | | | |
| $\lambda_r$ | 1152 | 1280 | 1408 | 1536 | 1664 | 1792 | 1920 | 2048 |
| $\lambda_d^{(l)}$ | $2^{31/8}$ | $2^{30/8}$ | $2^{29/8}$ | $2^{28/8}$ | $2^{27/8}$ | $2^{26/8}$ | $2^{25/8}$ | $2^{24/8}$ |
| $\lambda_p^{(l)}$ | $2^{21/8}$ | $2^{18/8}$ | $2^{15/8}$ | $2^{12/8}$ | $2^{9/8}$ | $2^{6/8}$ | $2^{3/8}$ | $2^{0/8}$ |

$1, 2, \cdots, L$, each synonymous level $l$ needs to learn different levels of information during model training. However, due to the limitations of computing resources during training, it is not possible to cover all synonymous levels in each forward process, and we can only use the loss functions of different synonymous levels alternately for training. This alternating training between layers can lead to some layers losing the ability to extract effective information early in training, causing the coding rate of the corresponding sub-feature map to approach 0 in the subsequent training process and in the final model. To avoid this, an effective trick is to introduce the following constraints into the loss function during the warming-up phase, ensuring that each sub-feature map learns valid information:

$$\mathcal{L}_c^{\text{warming}} = a \cdot \log p_{\tilde{\boldsymbol{y}}_l}(\tilde{\boldsymbol{y}}_l) + b \cdot \mathbf{std}\left(-\log p_{\tilde{\boldsymbol{y}}_1}(\tilde{\boldsymbol{y}}_1), \cdots, -\log p_{\tilde{\boldsymbol{y}}_L}(\tilde{\boldsymbol{y}}_L)\right), \tag{47}$$

in which the former term is the minus coding rate estimation of the current sub-feature map corresponds to the synonymous level $l$; the latter term, i.e., $\mathbf{std}\left(\cdot\right)$ is a standard deviation function, which calculates the standard deviation of the coding rates of each sub-feature map; $a$ and $b$ are the corresponding tradeoff factors. This constraint increases the coding rate estimation of the sub-feature map $l$ and limits the standard deviation of all the sub-feature maps' coding rate estimation, allowing each sub-feature map to learn a certain amount of effective information during the warm-up stage without excessive learning. We empirically set $a = 4, b = 64$ in this constraint.

Combining the synonymous constraints discussed at the end of Appendix B.2, the constraints in the training loss function in the warming-up process can be summarized as

$$\mathcal{L}_c^{(l)} = \mathcal{L}_c^{\text{synset}} + \mathcal{L}_c^{\text{detail}} + \mathcal{L}_c^{\text{warming}}, \tag{48}$$

After the warming-up phase, the constraints will be modified to

$$\mathcal{L}_c^{(l)} = \mathcal{L}_c^{\text{synset}} + \mathcal{L}_c^{\text{detail}}. \tag{49}$$

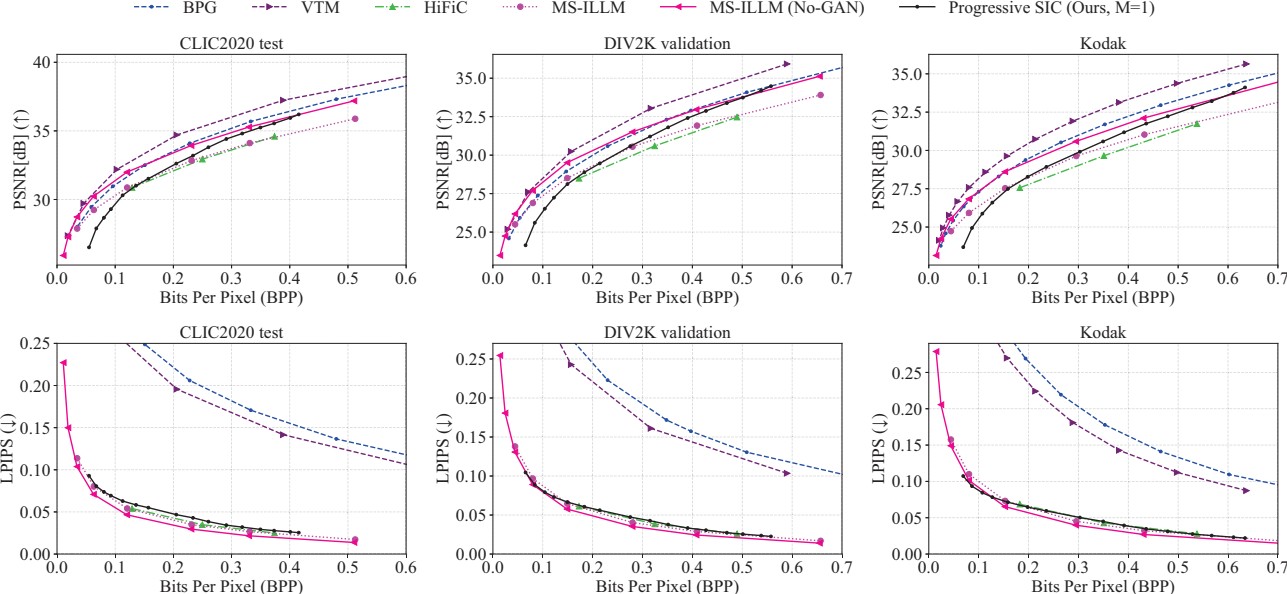

*Figure 11.* Comparisons of methods using PSNR and LPIPS on different datasets. Each point on the HiFiC and MS-ILLM performance curves is from a single model, while our entire performance curves are achieved by a single progressive SIC model.

During training, we treat the first 12500 iterations as the warming-up phase, and the subsequent iterations are for the formal training process.

In performance validation, for each input image at each synonymous level, we randomly select a single sample $\hat{x}_j$ from the reconstructed synset $\hat{\mathcal{X}}$ as the resulting image. Then we compare this result with the baseline methods by calculating the average coding rate, PSNR (distortion quality), and LPIPS and DISTS (perceptual quality) across the dataset.

## C.2. Supplementary Results

Figure 11 provides the supplementary PSNR and LPIPS quality results for Figure 4, which are quality assessment measures corresponding to the distortion and perceptual terms in the training loss function. As shown in the figure, for the distortion evaluation measure PSNR, as the coding rate increases, PSNR progressively approaches the performance of No-GAN MS-ILLM and even that of BPG. For the perceptual quality evaluation measure LPIPS, our solution reaches near the LPIPS quality of the perceptual solution at full rate. Even for the Kodak dataset, at low rates, our method outperforms the No-GAN MS-ILLM scheme. In this case, even without applying DISTS to the loss function for optimization, our method still demonstrates performance gains under most DISTS rates, shown as the results in Figure 4. This indicates that our method aligns better with the resampling tolerance emphasized by DISTS.

Consequently, our method achieves comparable rate-distortion-perception performance using a single progressive SIC model, which demonstrates our advantages in variable-rate support.

Besides, the visualization results are presented from Figure 12 to Figure 17, which can be divided in two groups:

1. The first group illustrates the process of improving the reconstructed image as the synonymous level $l$ increases, which includes Figure 12, Figure 13, and Figure 14, corresponding to the test image from CLIC2020 test, DIV2K validation, and the Kodak dataset, respectively.

   We captured the effects of reconstructed images at specific synonymous levels to clearly demonstrate how switching synonymous levels impacts the quality of the reconstructed images: At low synonymous levels, the coding rate for the synonymous representation is relatively low. As a result, the reconstructed image is sampled from a larger synset, capturing only the global semantic content of the image, with limited pixel-level detail accuracy. As the synonymous level increases, the coding rate rises, which reduces the size of the reconstructed synset. This allows for more accurate

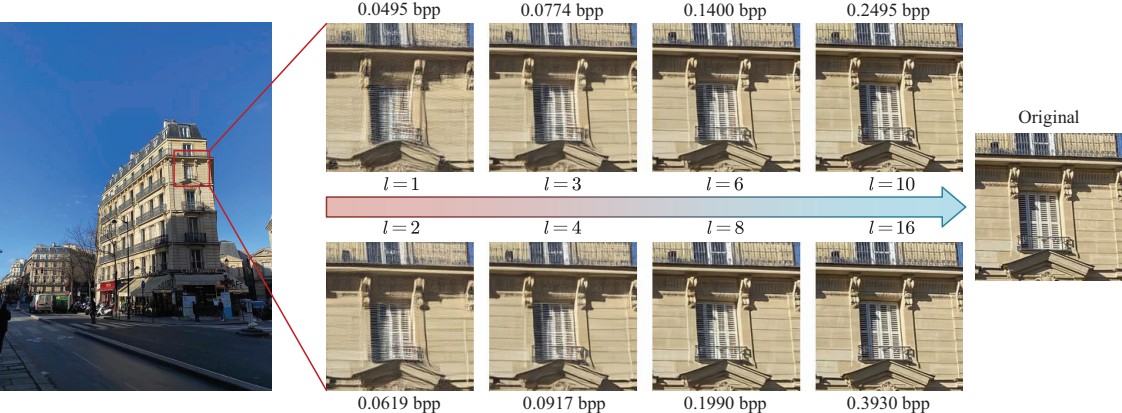

Figure 12. Visualization results of reconstructed images at different synonymous levels using progressive SIC ($M = 1$). Image from the CLIC2020 test set.

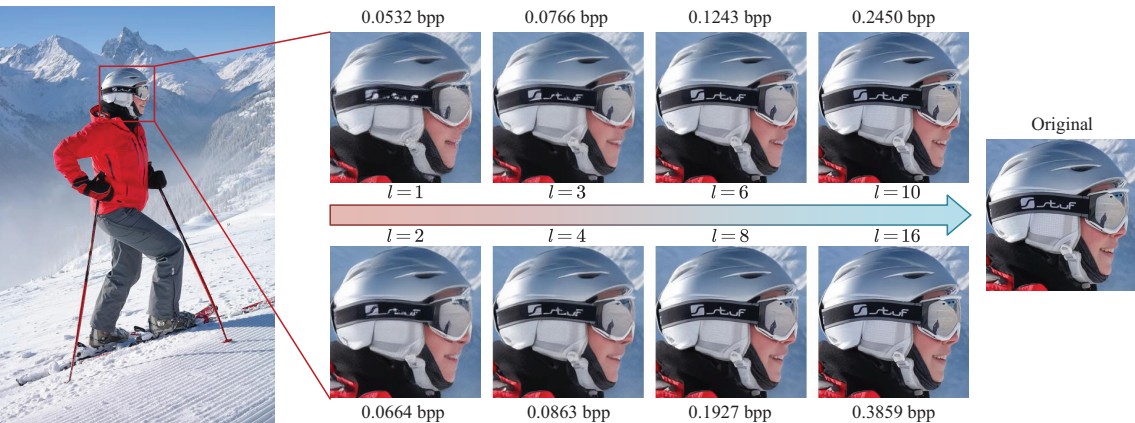

Figure 13. Visualization results of reconstructed images at different synonymous levels using progressive SIC ($M = 1$). Image from the DIV2K validation set.

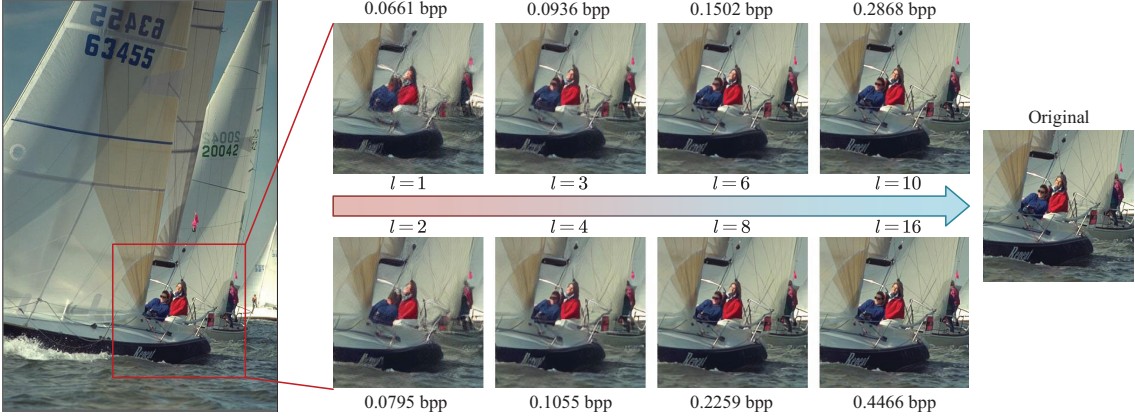

Figure 14. Visualization results of reconstructed images at different synonymous levels using progressive SIC ($M = 1$). Image from the Kodak validation set.

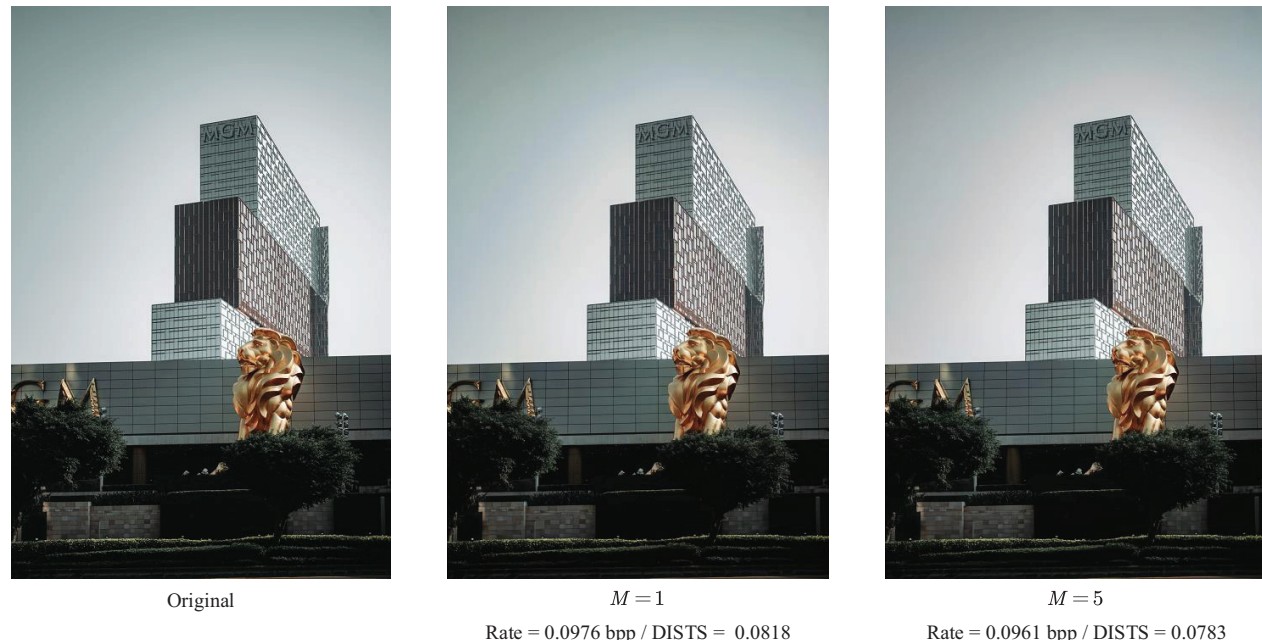

| Original | $M=1$ | $M=5$ |
|---|---|---|
| | Rate = 0.0976 bpp / DISTS = 0.0818 | Rate = 0.0961 bpp / DISTS = 0.0783 |

*Figure 15.* Visualization comparison of reconstructed images at synonymous level $l = 5$ using progressive SIC with $M = 1$ and $M = 5$. Image from the CLIC2020 test set.

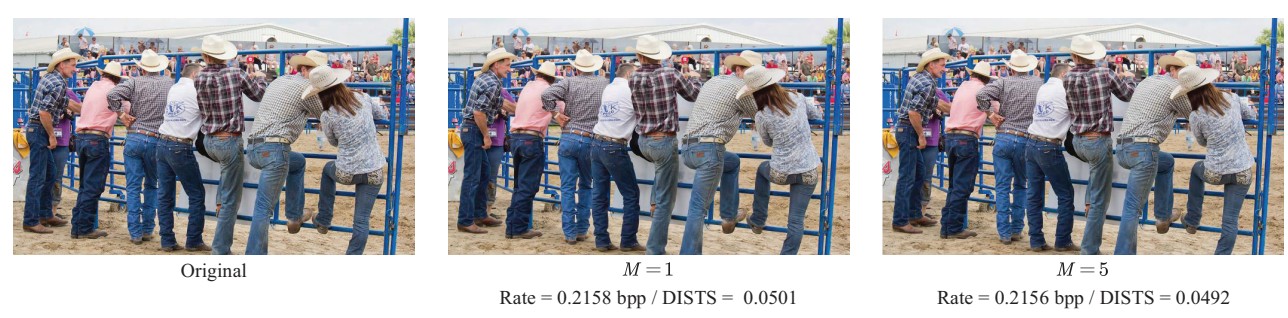

| Original | $M=1$ | $M=5$ |
|---|---|---|
| | Rate = 0.2158 bpp / DISTS = 0.0501 | Rate = 0.2156 bpp / DISTS = 0.0492 |

*Figure 16.* Visualization comparison of reconstructed images at synonymous level $l = 6$ using progressive SIC with $M = 1$ and $M = 5$. Image from the DIV2K validation set.

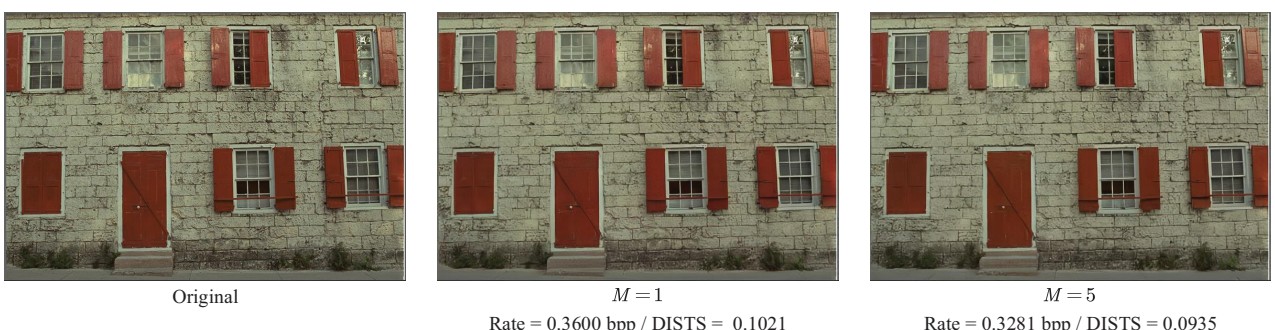

| Original | $M=1$ | $M=5$ |
|---|---|---|
| | Rate = 0.3600 bpp / DISTS = 0.1021 | Rate = 0.3281 bpp / DISTS = 0.0935 |

*Figure 17.* Visualization comparison of reconstructed images at synonymous level $l = 7$ using progressive SIC with $M = 1$ and $M = 5$. Image from the Kodat dataset.

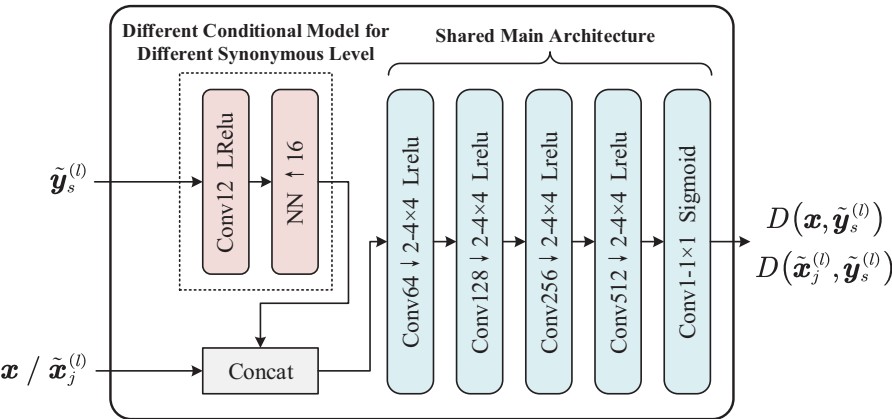

*Figure 18.* Employed Discriminator Architecture in our finetuning attempts.

semantic information and progressively enhances the details in the reconstructed image.

These visualization results demonstrate that the progressive SIC model we implemented can effectively leverage the switching of synonymous levels to adjust to different coding rates, enabling the accuracy of the reconstructed image to improve as the coding rate increases, while ensuring a smooth enhancement of perceptual quality.

2. The second group presents a visualized quality comparison of the reconstructed images corresponding to specific synonymous levels, with the number of detailed representations $\hat{\boldsymbol{y}}_{\epsilon,j}$ set to $M = 1$ and $M = 5$ during training. It includes Figure 15, Figure 16, and Figure 17, corresponding to the test image from CLIC2020 test, DIV2K validation, and the Kodak dataset, respectively. The perceptual quality is evaluated using the DISTS measure.

These visualization results demonstrate certain advantages of increasing the sampling number of $\hat{\boldsymbol{y}}_{\epsilon,j}$ in perceptual qualities at certain synonymous levels, since more sampling results in effective learning of the shared characteristics within the reconstructed synset $\hat{\boldsymbol{\mathcal{X}}}$.

We state that the results presented in this article are some of our preliminary results. Currently, our investigations on the perceptual loss (i.e., the divergence term) utilization, synonymous level partition mechanisms, sampling numbers, and hyperparameter settings on progressive SIC schemes are still insufficient. These factors may contribute to the issues observed in Figure 4 and Figure 5. Future works are needed to explore these aspects in more detail.

## D. Limitations: Implementation Details and Supplementary Results

In this section, we present the implementation details and results of the supplementary experiments in Section 6, focusing on the primary concern of using GAN-based adversarial loss to replace the divergence term in the loss function of Equation (11).

### D.1. Implementation Details

As described in Section 6, we use GAN-based adversarial loss to replace the divergence term in the loss function and improve the performance of our implemented SIC model. Hence, the auto-encoder architecture remains consistent with those in Appendix C.1. This section mainly details the discriminator design.

The utilized conditional discriminator is a convolutional neural network with two input branches—the original image $\boldsymbol{x}$ / reconstructed image $\tilde{\boldsymbol{x}}_j^{(l)}$ as a main branch and the synonymous representation $\tilde{\boldsymbol{y}}_s^{(l)}$ as a condition branch—and outputs the probability that the input image is judged as real. Besides, the discriminator model consists of two parts, i.e., a group of conditional models and a main architecture. The conditional branch is first upsampled by 16 times, concatenated with the main branch in the channel dimension, and then fed into the main architecture to estimate the output probability. We fine-tune the synthesis transform $g_s$ (i.e., the generator) with a single discriminator model, using a corresponding conditional model for each synonymous level $l$ and sharing the main architecture across all synonymous levels.

### D.2. Supplementary Results

Figure 19 shows the performance of the fine-tuned model (labeled as "with GAN") using PSNR, LPIPS, DISTS, and FID, compared with the original model (labeled as "no-GAN") and other comparison schemes across different datasets. It should be noted that calculating FID on the Kodak dataset is unavailable as this dataset has only 24 images to yield the useful metrics, which is also stated in (Muckley et al., 2023). These results show that introducing adversarial loss improves perceived quality, indicating its optimization direction is closer to the optimal divergence than metrics like LPIPS.

However, unlike in classical perceptual compression methods (Muckley et al., 2023), we observe that while DISTS—more tolerent to resampling—improves significantly, FID—which measures distribution distance—shows limited gains. This suggests our empirical detail sampling leads to a suboptimal reconstructed image distribution, likely because it does not follow the ideal probability distribution of the detail representations in vector form, which verifies our statements in Appendix C.1. Therefore, solving this problem is also one of the key research directions of our future work.

Figure 20 compares reconstructed images before and after fine-tuning with adversarial loss at synonymous level $l = 1$, using samples from different datasets. Although DISTS and FID show limited numerical improvement at this level, the perceptual improvement can be visibly enhanced, which confirms that adversarial loss improves generation quality across synonymous levels (i.e., across different rate ranges).

Although we acknowledge a performance gap between our method and state-of-the-art approaches, we also need to clarify once again the advantages of our approach. That is, our implementation scheme—unlike schemes that require separate models for each rate—offers much easier deployment by achieving acceptable perceptual qualities using one single model when we set different synonymous levels. We have identified its current limitations and proposed potential solutions; addressing these could enable SIC to approach the theoretically optimal scheme instructed by SVI and thus potentially surpass existing methods in future works.

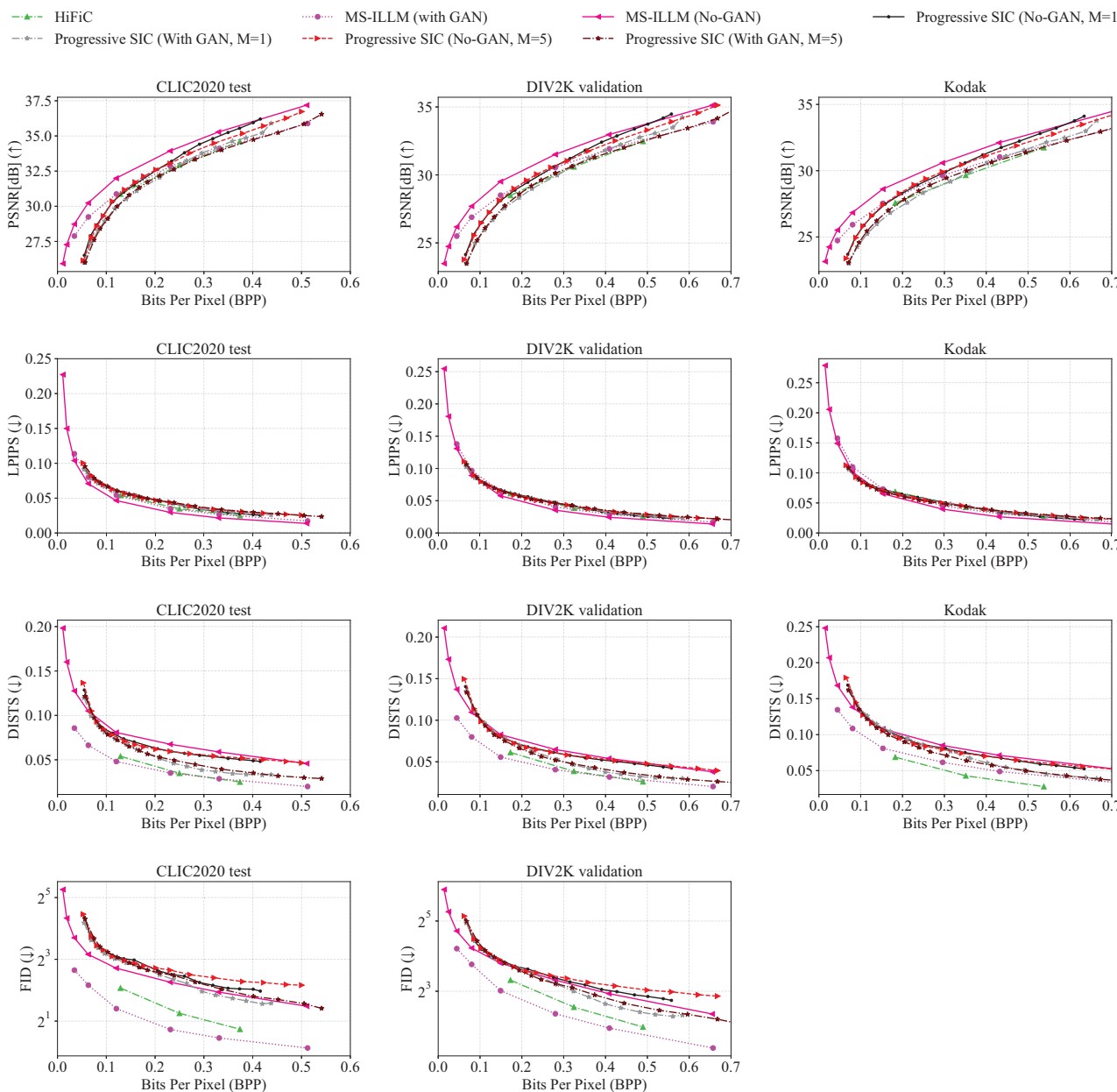

*Figure 19.* Comparisons of methods using PSNR, LPIPS, DISTS, and FID on different datasets (Supplemented fine-tuned model performance). Each point on the HiFiC and MS-ILLM performance curves is from a single model, while our entire performance curves are achieved by a single progressive SIC model.

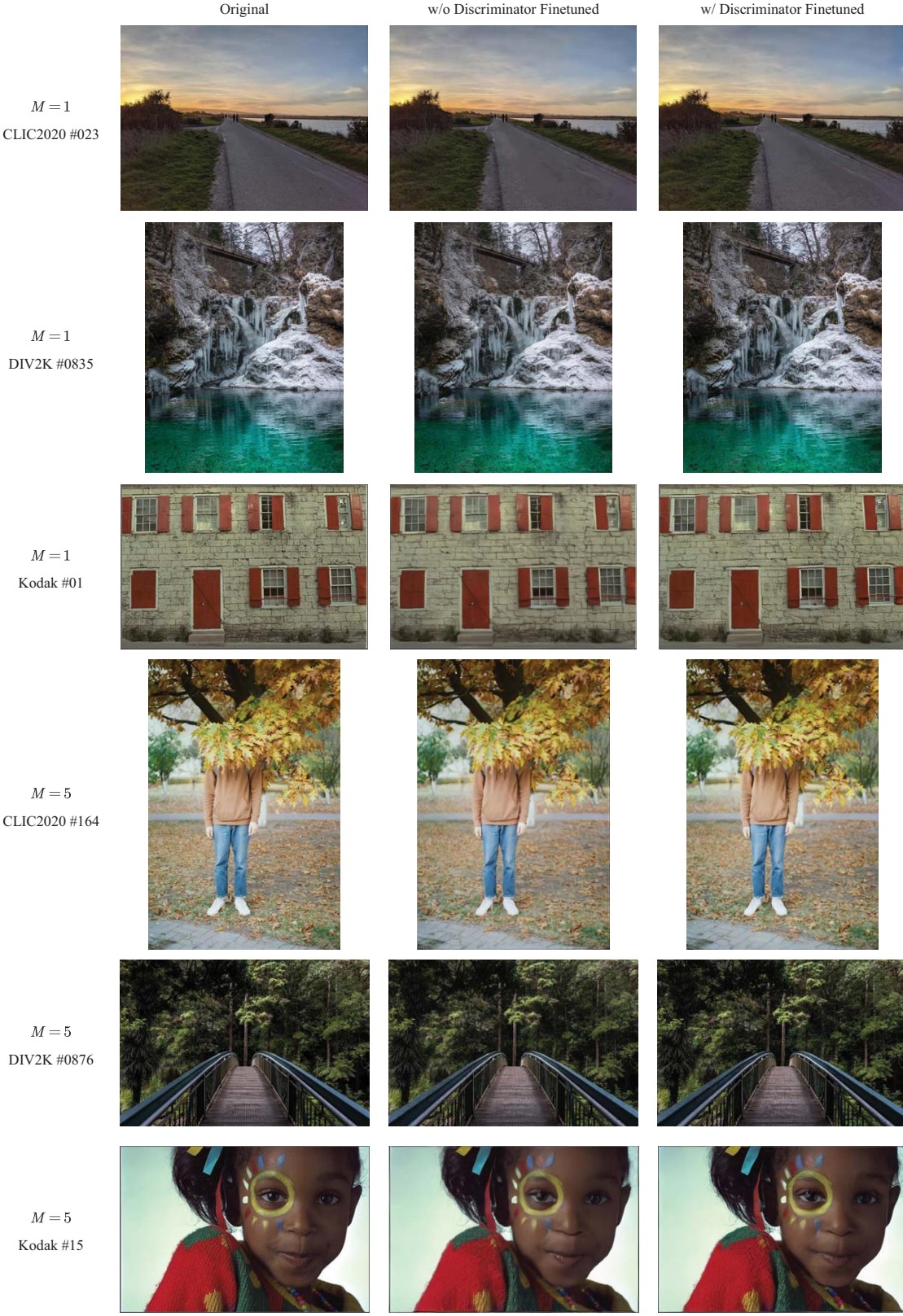

*Figure 20.* Visualization results of reconstruction images: No-GAN vs. with GAN (Synonymous level $l = 1$). Images from the CLIC2020 test set, the DIV2K validation set, and the Kodak dataset.

