# OpenReview forum: "Synonymous Variational Inference for Perceptual Image Compression"
_ICML.cc/2025/Conference — ICML 2025 poster_

### Official Review · Reviewer_KhzF · 2025-02-21

**Overall Recommendation:** 2

**Summary:**

This paper proposes a novel framework for perceptual image compression based on synonymous variational inference (SVI). Specifically, the paper introduces a method to analyze the optimization direction of perceptual image compression using semantic information theory. A new image compression scheme called Synonymous Image Compression (SIC) is devised to encode latent synonymous representations and reconstruct images through sampling. A progressive SIC codec is developed to leverage multiple synonymous levels for variable-rate compression. Besides, to the best knowledge of the authors, this method is the first work that can theoretically explain the fundamental reason for the divergence measure’s existence in existing perceptual image compression schemes. However, it lacks comprehensive comparisons with essential state-of-the-art methods and does not convincingly validate the effectiveness of the proposed models through ablation studies.

**Claims And Evidence:**

Claim: Experimental results demonstrate comparable rate-distortion-perception performance using a single neural progressive SIC image codec, thus verifying our method’s effectiveness.
Evidence: The paper shows some improvements in DISTS and LPIPS metrics, but lacks comprehensive comparisons with state-of-the-art methods like HiFiC and MS-ILLM (with GANs).
Evaluation: This claim is not fully convincing due to the limited comparison. More extensive experiments with diverse datasets and state-of-the-art methods are needed.

**Essential References Not Discussed:**

None

**Experimental Designs Or Analyses:**

Yes, I have reviewed the experimental designs and analyses presented in the paper. The authors aim to validate the effectiveness of their proposed Synonymous Variational Inference (SVI) method and the corresponding Synonymous Image Compression (SIC) scheme.
1.The choice of datasets and metrics is appropriate for evaluating the performance of the proposed method. These datasets are widely used in the field, ensuring that the results are comparable to other studies.
2. While the paper provides comparisons with several methods, the results for some state-of-the-art perceptual image compression methods (e.g., HiFiC and MS-ILLM with GANs) are not comprehensive. This limits the strength of the claims regarding the superiority of the proposed method.

**Methods And Evaluation Criteria:**

No, the proposed methods and evaluation criteria do not fully align with the problem or application at hand. While the paper introduces a novel approach to perceptual image compression from the perspective of Synonymous Variational Inference (SVI) and theoretically explores the fundamental reason for the divergence measure’s existence in existing schemes, the experimental results fail to demonstrate sufficient competitiveness in practical applications. The evaluation criteria, though relevant to perceptual quality, do not comprehensively validate the effectiveness of the proposed method compared to state-of-the-art techniques. Therefore, the proposed methods and criteria are insufficient to address the practical needs of perceptual image compression.

**Other Comments Or Suggestions:**

1.The paper provides comprehensive experimental results, but it would be beneficial to include more detailed ablation studies. For example, analyzing the impact of different components of the proposed method (e.g., the effect of learnable weights, the choice of architecture) could provide deeper insights into the method's effectiveness.
2.The description of the SIC scheme is thorough, but it could benefit from additional visualizations showing more results about the progression of image quality at different synonymous levels. This would help readers better understand the practical implications of the method.
3.The paper provides several performance metrics (PSNR, LPIPS, DISTS), but it would be useful to include additional qualitative results.

**Other Strengths And Weaknesses:**

Strengths:
1.Originality and Novelty: The paper introduces a novel perspective on perceptual image compression by leveraging semantic information theory and synonymous variational inference (SVI). To the best knowledge of the authors, this method is the first work that can theoretically explain the fundamental reason for the divergence measure’s existence in existing perceptual image compression schemes.
2.Clarity and Presentation:
The paper is well-structured and clearly presents the theoretical concepts, methods, and experimental results. The use of visualizations and detailed explanations helps in understanding the complex ideas and their practical implications.

Weaknesses:
1.Performance: The paper shows some improvements in DISTS and LPIPS metrics, but lacks comprehensive comparisons with state-of-the-art methods like HiFiC and MS-ILLM (with GANs).

**Questions For Authors:**

1.Could the authors provide a more detailed comparison with state-of-the-art perceptual image compression methods, particularly those using GANs or other advanced techniques like HiFiC and MS-ILLM? How does the proposed SVI method differ from these methods in terms of performance and computational efficiency?
2.Could the authors provide additional ablation studies to validate the theoretical claims made in the paper? For example, how does the performance change when different components of the SVI method (e.g., partial semantic KL divergence, synonymous level partitioning) are removed or modified?

**Relation To Broader Scientific Literature:**

The key contributions of the paper are closely related to several areas of the broader scientific literature, particularly in the fields of image compression, semantic information theory, and variational inference. It introduces novel ideas such as synonymous sets and partial semantic KL divergence, which offer new perspectives on handling perceptual similarity in image compression.

**Theoretical Claims:**

Claim: The proposed Synonymous Variational Inference (SVI) method is the first to theoretically explain the fundamental reason for the divergence measure’s existence in existing perceptual image compression schemes.
Proof: The authors provide a theoretical analysis showing that the divergence measure arises naturally from the consideration of synonymous sets and semantic information theory. They introduce the concept of partial semantic KL divergence and demonstrate its relevance in the context of perceptual image compression.

---

> ### Author Rebuttal · Authors · 2025-04-01
>
> Dear Reviewer KhzF,
>
> Thank you for recognizing our work, especially **the originality and novelty of our SVI theory analysis for perceptual image compression** and our **well-structured and clearly presented**.
>
> Your concern centers on our unsatisfactory experimental results, which is also a concerned by many reviewers. However, we want to claim that **our focus is on the SVI analysis theory, which offers a new theoretical framework for perceptual image compression**, while **the implemented SIC model serves only as a preliminary validation**. Compared to the completeness of our theory, our method is indeed relatively rough. This is due to several issues in the theoretical analysis that need further attention in method design, while these issues in turn offer significant potential for our future research.
>
> Below are our responses to your concerns.
>
> **[Q1] The paper shows some improvements in DISTS and LPIPS metrics, but lacks comprehensive comparisons with state-of-the-art methods like HiFiC and MS-ILLM (with GANs).**
>
> [A1] As several reviewers have raised concerns about this issue, we conducted additional experiments, introduced a discriminator into the model structure, and fine-tuned our model using discriminative loss. We have included some of the results and comparison with HiFiC and MS-ILLM in [Figure 5] of the [[Anonymous Link]](https://anonymous.4open.science/r/supplementaryResults_SVI-F92C) and provided visualizations in [Figure 6]. For detailed configuration, reasoning, and discussion, please check the rebuttal to **Reviewer 7ghf [A2]**.
>
> Although our method still has some gaps in perceptual quality compared with HiFiC and MS-ILLM, it offers a clear advantage in deployment cost. Specifically, we can use a single model to achieve all bit rates, with a model size of 465MB. In contrast, an MS-ILLM model can only reconstruct the received information for one bitrate, with a single model size of 693MB. To support all bit rates (e.g., MS-ILLM with GAN, 6 points), you would need 6×693MB = 4158MB = 4.06GB, which is significantly larger than our model. Therefore, in this respect, our model has a clear advantage.
>
> **[Q2] It would be beneficial to include more detailed ablation studies.**
>
> [A2] We added an ablation study on the detail representation part in the provided anonymous link, comparing random sampling with forcing it to 0. Please refer to [Figure 2~3] of the above anonymous link.
>
> We believe this ablation is important as it highlights the difference between our approach and existing methods (HiFiC and MS-ILLM). Specifically, it shows that random details provide effective information for image reconstruction, making distortion and perceptual quality not entirely dependent on the information provided by the coding sequence.
>
> Additionally, although an ablation study on different components of SVI (i.e., partial semantic KL divergence, synonymous level partitioning) is essential, we prefer to discuss this issue based on our theoretical results:
>
> - If partial semantic KL divergence is not utilized, the existence of the ideal synset will not be considered (i.e., only consider the original image), which makes the loss function only a rate-distortion tradeoff without perceptual term. This is the core reason why we propose using this divergence for analysis: **It provides a sufficient theoretical foundation for using distribution distance as a perceptual loss.** This can be referred to Appendix A.3 (especially Lines 949~967) in our manuscript.
>
> - If synonymous level partitioning is not utilized, the model should be a single-point model, which only accepts a specific coding rate, similar to how HiFiC and MS-ILLM operate. In this case, the performance of each point in our method has the potential to be improved, as there is no competition between different synonymous levels, as illustrated by the tradeoff loss in Equation (12). However, the disadvantage of this approach is that it increases the cost of deployment, as a separate model needs to be deployed for each bitrate.
>
> **[Q3] It would be useful to include additional qualitative results.**
>
> [A3] In [figure 5] of our provided anonymous link, we added performance curves of our model (no-GAN and with GAN) for **FID** and compared the results with HiFiC and MS-ILLM (with GAN). Although the performances under FID are still unsatisfactory, we analyze the underlying causes and consider it an essential direction for future research. Please check the rebuttal to **Reviewer 7ghf [A2]** for this discussion.
>
> **[Q4] More results about the progression of image quality at different synonymous levels are required.**
>
> [A4] Thanks for your suggestion. We added a progression of images at different synonymous levels in [Figure 7~8] in the anonymous link, in which [Figure 7] shows gradual changes of the images by our $M=1$ model, and [Figure 8] by our $M=5$ model.
>
> In summary, we sincerely thank you for your suggestions. These suggestions will help us improve our paper.

---

### Official Review · Reviewer_7ghf · 2025-03-09

**Overall Recommendation:** 4

**Summary:**

This paper presents a novel perspective to analyze the perceptual image compression problem, which is based on the notion of synonymy in semantic information theory, which suggests that images with perceptual similarity constitute a synonymous set. Based on this, the authors propose a synonymous variational inference method (SVI) and theoretically demonstrate that the optimization direction of perceptual image compression follows a three-pronged trade-off that encompasses bit rate, distortion, and perception, which is in line with existing related research. In addition, the paper designs a new image compression scheme, synonymous image compression (SIC), and verifies its effectiveness experimentally.

## update after rebuttal
The author answered my questions, so I upped my score. I hope the author will make the code public in the future to promote the community.

**Claims And Evidence:**

The analysis proves theoretically that the optimization direction of perceptual image compression follows a triple tradeoff, i.e., a synonymous rate-distortion-perception tradeoff, and that this tradeoff can cover existing rate-distortion-perception schemes. Theorem 3.3 indeed gives a theoretical formulation of this triple tradeoff (Eq. 9), and Appendix A.2 provides a detailed proof procedure. Appendix A.3 also discusses compatibility with existing schemes. However, “covering” existing programs may mean that existing programs are special cases of the theory, which is mathematically supported. However, this does not necessarily mean that the design concepts of all existing programs can be fully explained or guided by the theory.

**Essential References Not Discussed:**

Research on “Perceptual Equivalence” or “Just Noticeable Difference (JND)” in image compression: In this paper, perceptual similarity is considered as a criterion for tautological relationships. In the field of image processing and computer vision, there has been researched on when two images are perceptually “identical” or “indistinguishable”, such as image compression methods based on the JND model. These models attempt to identify perceptually unimportant information and remove it, thereby improving compression rates while maintaining perceptual quality. Although JND models do not directly correspond to the concept of “synonym set”, they are concerned with the sensitivity of the human perceptual system to image differences, which can complement the understanding of the theoretical basis of the “perceptual similarity” criterion in this paper.

**Experimental Designs Or Analyses:**

1) Use of LPIPS instead of KL scatter: although the authors explain the use of LPIPS instead of KL scatter due to computational challenges, LPIPS is, after all, a depth-feature based perceptual metric that is not mathematically equivalent to the KL scatter in the theoretical derivation. This may affect the extent to which the experimental results directly validate the theory.
2) Simplicity of the synonym level division: a uniform division of the channel dimensions is used in the paper to define the different synonym levels. The effectiveness and optimality of this approach may require further exploration and experimental validation. More complex or data-driven based segmentation strategies may be able to better capture different levels of semantic information.
3) Setting of hyperparameters: the paper mentions that the hyperparameters in the loss function are configured empirically. The selection of these hyperparameters is crucial to model performance, but the lack of systematic hyperparameter search and analysis may affect the robustness and reproducibility of the results.
4) Impact of different numbers of samples (M): Fig. 5 illustrates a comparison of Progressive SIC performance using different numbers of samples M (M=1 and M=5) in the reconstructed synonym set X̂. The results show that different M values perform inconsistently at different code rates, sometimes M=1 is better and sometimes M=5 is better. The authors recognize that this may be related to the mechanism of synonym level division, number of samples, and insufficient setting of hyperparameters, which need to be further explored in future work.
5) Interpretation of the visualization results: although Figs. 11-16 show the visual effects of the reconstructed images, a more in-depth analysis of these visualization results, such as the changing patterns of image details and perceived quality under different synonym levels, and the effects of different M values on the diversity and quality of the generated images may be more helpful in understanding the performance of SIC.

**Methods And Evaluation Criteria:**

1) The SVI framework proposed in the paper provides a new theoretical foundation for perceptual image compression from the perspective of semantic information theory.
2) The SIC compression scheme is a concrete realization based on this theory, and its design concept is consistent with the core idea of SVI.
3) The design of progressive SIC takes into account the needs of practical applications and allows the generation of images with different perceptual qualities at different rates.
4) The selection of commonly used benchmark datasets and an evaluation system that includes both traditional metrics (PSNR) and perceptual metrics (LPIPS, DISTS) enables a comprehensive evaluation of the performance of the proposed method and facilitates the comparison with existing techniques. The authors particularly emphasize the DISTS metrics, which are related to their proposed approach based on tautology.

**Other Comments Or Suggestions:**

There were some problems with the format of the references, for example, some conference papers had “Proceedings” while others did not. In addition, there is no consistency in the case of the names of the conferences covered in the references.

**Other Strengths And Weaknesses:**

Pros:
1) The originality is highlighted by the innovative application of the concept of “tautology” in semantic information theory to the field of perceptual image compression. This not only provides a new theoretical perspective on the problem, but also makes the first attempt to explain, at a theoretical level, the fundamental reason for the existence of distributional distance metrics (e.g., KL dispersion or its substitutes) in perceptual image compression, which stems from the consideration of an ideal set of synonyms.
2) The paper proposes a novel variational inference method, the Synonymy Variational Inference (SVI), and derives a synonymy rate-distortion-perception ternary trade-off based on this theory. This theoretically extends and unifies the existing rate-distortion and rate-distortion-perception theories by showing that the latter can be regarded as a special case of the former under specific conditions. This theoretical unification has important academic value.

Weaknesses:
1) The experimental results, although competitive, do not show a significant advantage in terms of perceptual quality over some state-of-the-art GAN-based methods such as HiFiC and MS-ILLM. The authors also acknowledge that this may be related to the direct use of LPIPS instead of KL dispersion in the loss function and propose to explore the discriminative mechanism in the future to further improve the perceptual quality. This suggests that there is still room for improvement in the practical performance of the currently proposed SIC model.

**Questions For Authors:**

The experimental results show that the advantage of the proposed method over some advanced GAN-based perceptual image compression methods (e.g., HiFiC and MS-ILLM) in terms of perceptual quality (DISTS metrics) is not significant. Considering that the SVI framework aims to provide a unified perspective on perceptual image compression, how do the authors view this performance gap? How are future plans to integrate discriminative mechanisms or adversarial training within the SVI framework to narrow this gap while still adhering to the semantic information theory based on tautology? A clear articulation of the author's understanding of this gap and promising future research directions would help to enhance the long-term potential and impact of this work.

**Relation To Broader Scientific Literature:**

1) Links between synonymous variational inference (SVI) and semantic information theory.
2) The SVI connection to rate-distortion-perception (R-D-P) theory.
3) The difference and connection between Synonymous Image Compression (SIC) and Learning Image Compression (LIC).

**Theoretical Claims:**

1) Proof of Lemma 3.2: This lemma states the equivalence between minimizing the expected negative logarithmic tautological likelihood term and minimizing the weighted expected distortion term and the weighted expected KL scatter term. A detailed proof of the lemma is given in Appendix A.1.
2) Proof of Theorem 3.3: The theorem presents a formula for the minimum achievable rate of perceptual image compression, i.e., the tautology rate-distortion-perceptual tradeoff. The detailed proof of the theorem is given in Appendix A.2.

---

> ### Author Rebuttal · Authors · 2025-04-01
>
> Dear Reviewer 7ghf,
>
> We sincerely appreciate your high regard for our work, especially your recognition of our work including:
>
> - **New theoretical perspective on perceptual image compression**
>
> - **Mathematically supported unified theory**, which can **extends and unifies the existing RD and RDP theories**
>
> - **Holding important academic value**.
>
> Your main concern is the unsatisfactory experimental results, which many reviewers have also raised. Below are our responses to the relevant questions.
>
> **[Q1] How do the authors view the performance gap between the SIC methods and the advanced RDP methods in terms of DISTS?**
>
> [A1] We believe the insignificant performance of our method on DISTS is due to multiple factors:
>
> - **The choice of perceptual loss.** Your understanding of using LPIPS instead of KL divergence is accurate. With LPIPS replacing KL divergence as the perceptual loss, our scheme outperforms MS-ILLM (no-GAN) on DISTS mainly due to DISTS' resampling tolerance, which aligns with our method's random detail sampling. However, the advantage is limited. Previous work (MS-ILLM) shows that using adversarial loss significantly improves DISTS quality. Therefore, we believe that fine-tuning our model with a discriminator and adversarial loss can help bridge this gap to some extent.
>
> - **Confronts between multiple synonymous levels**. To ensure each synonymous level in the progressive SIC model is effective, we use Equation (12) (line 319) for alternating training. This creates battles between different optimization directions within a single SIC model, inevitably impacting overall performance.
>
> - **Suboptimal choice of sampling mechanism**. In the detail representation part, we empirically use uniform sampling, which may cause significant differences between the distribution of the reconstructed and real images.
>
> Therefore, future research should combine theoretical analysis and method design to address these issues.
>
> **[Q2] Future plans to integrate discriminative mechanisms or adversarial training to narrow this gap while still adhering to the semantic information theory?**
>
> [A2] In fact, this is also a concern of many reviewers. To this end, we tried to build a discriminator model based on HiFiC’s discriminator structure and fine-tuned our $M=1$ and $M=5$ models with non-saturating loss for $2×10^5$ steps. We plot the discriminator structure in [Figure 4] and the fine-tuned performance curves in [Figure 5] (PSNR, LPIPS, DISTS, and KID) of the [[Anonymous Link]](https://anonymous.4open.science/r/supplementaryResults_SVI-F92C/). The supplementary results show that:
>
> - **The perceptual quality (DISTS, FID) has improved**, with more noticeable gains at higher bitrates, and DISTS gradually approaches the performance of MS-ILLM (with GAN). Besides, while DISTS and FID values change little at low rates, the visual quality improves significantly, as shown in [Figure 6] ([Table 3] corresponds to the bitrates in [Figure 6]).
>
> - **The distortion has degraded,** which can verify the distortion-perception tradeoff as mentioned in (Blau & Michaeli, 2018).
>
> - **The gap compared to HiFiC and MS-ILLM is still obvious,** especially on FID. This may be due to insufficient fine-tuning with multiple synonymous layers battling against each other. Besides, this also means that the issues stated in [A1] need to be solved in the future, including designing better perceptual optimization directions, broadening SVI theory to address multi-synonymous level confronts, and improving detail prediction and sampling methods to fit the true distribution.
>
> **[Q3] Concerns about the correlation between synset and "JND"?**
>
> [A3] This comment is crucial for completing our explanation of the synset. JND's ideas are somewhat similar to ours. However, while JND provides the minimum stimulus difference that causes a perceptible change, which may help us to determine a certain synset threshold to keep human-perceptual consistency, our concept of synset takes it a step further:
>
> In certain cases, only ultra-low bitrates are sent to the decoder, where the reconstructed image can differ significantly from the original image—beyond the JND threshold—and still be tolerable to people in that context.
>
> A typical example is the use of diffusion models for ultra-low bitrate image compression, where the reconstructed images exhibit significant differences from the originals, yet maintain good perceptual quality that is acceptable to humans.
>
> **[Q4] "Covering" is mathematically supported, but not mean that all existing programs can be fully explained or guided by the theory.**
>
> [A4] You are correct, but this also means there is room for further development of our theory. We believe the SVI theory presented in this manuscript is just the beginning, and future research will involve theoretical explanations of various existing methods.
>
> In summary, thanks for your valuable comments and suggestions. We will update our manuscript accordingly.

---

> > ### Comment · Reviewer_7ghf · 2025-04-06
> >
> > Thanks for the reply. The author answered my questions. I hope the author will make the code public in the future to promote the community.

---

> > > ### Author Response · Authors · 2025-04-06
> > >
> > > We sincerely thank you for your recognition and encouragement of our work! We will make our code public in the future to promote the community.

---

### Official Review · Reviewer_VZKv · 2025-03-11

**Overall Recommendation:** 3

**Summary:**

This paper introduces a novel progressive training approach for image compression, which focuses on both improving image quality and maintaining semantic consistency during compression. By using synonymous latent representations, the model progressively decodes and recovers image details, ensuring high-quality reconstruction, especially at low bitrates. The loss function combines rate-distortion loss with a partial semantic KL-divergence loss to optimize semantic integrity. While comparable to direct LPIPS optimization in terms of perceptual quality, this method offers advantages in preserving semantic consistency and flexibility in quality recovery.

## update after rebuttal
The authors have already resolved my issues. As a result, I raise the final score.

**Claims And Evidence:**

The claims made in the submission are supported by clear and convincing evidence.

**Essential References Not Discussed:**

None.

**Experimental Designs Or Analyses:**

Please refer to weakness in ‘Other Strengths And Weaknesses’.

**Methods And Evaluation Criteria:**

Yes.

**Other Comments Or Suggestions:**

Refer to weakness in ‘Other Strengths And Weaknesses’.

**Other Strengths And Weaknesses:**

Strength：
1. Unlike conventional perceptual image compression methods that primarily focus on pixel-wise reconstruction, this work leverages synonymous latent representations to ensure semantic consistency between the compressed and original images. This helps retain core image features while allowing for flexibility in fine details.
2. By focusing on semantic information, the proposed approach is valuable for multi-modal compression tasks, such as image-text joint encoding, where maintaining cross-modal consistency is crucial.

Weaknesses
1. In Section 3.2, the paper derives the Partial Semantic KL Divergence, emphasizing its role in maintaining semantic consistency. However, the final training loss(equation 12) almost identical to existing Rate-Distortion-Perception (RDP) frameworks.
2. The experimental results show that the proposed method performs almost identically to direct LPIPS-optimized methods in perceptual metrics (LPIPS, DISTS). This raises concerns about whether the method provides any meaningful improvement.
3. The paper claims that synonymous latent representations improve semantic consistency, ensuring that images at different bitrates retain the same semantic information. However, there is no quantitative validation of this claim in the experiments. The experiments only report perceptual metrics (LPIPS, DISTS), which do not directly assess semantic consistency. Can the authors provide quantitative evidence that their method maintains better semantic consistency across bitrates?

**Questions For Authors:**

Refer to weakness in ‘Other Strengths And Weaknesses’.

**Relation To Broader Scientific Literature:**

This paper builds on prior work in perceptual image compression, progressive coding, and semantic-aware latent representations. It extends perceptual compression techniques such as LPIPS-optimized methods by introducing synonymous latent representations to enhance semantic consistency across compression levels. Additionally, it relates to progressive coding approaches but differentiates itself by focusing on gradual semantic detail recovery rather than purely hierarchical bitstream refinement. However, its final training loss remains similar to existing rate-distortion-perception (RDP) methods, raising questions about its practical advantages over LPIPS-optimized compression.

**Theoretical Claims:**

I have carefully read the proofs for theory in the article, and they are all reasonable.

---

> ### Author Rebuttal · Authors · 2025-04-01
>
> Dear Reviewer VZKv,
>
> We sincerely appreciate your recognition of our SVI analysis, especially your understanding of **semantic consistency** and the **crucial potential in multi-modal compression tasks of our SVI theory**.
>
> Your concerns are essential for refining our paper and guiding future research. Below are our responses.
>
> **[Q1]  The paper derives the Partial Semantic KL Divergence, emphasizing its role in maintaining semantic consistency. However, the final training loss (equation 12) is almost identical to existing Rate-Distortion-Perception (RDP) frameworks.**
>
> [A1] You may misunderstand the role of the Partial Semantic KL Divergence: It optimizes image samples in the reconstructed synset within the image space conditioned on **ensuring explicit semantic consistency in the latent space**. This will **finally result in implicit semantic consistency with the original image, aligning with the ideal synset**, as shown in Figure 1 of the manuscript.
>
> As for the final loss function, which is almost identical to the existing RDP framework, there are two misunderstandings:
>
> - **The object of coding rates**: Only contains the **common features** of all samples in the reconstructed synset, which is **partial** latent features between the original image and any reconstructed image. This can be intuitively illustrated as the Venn Diagrams [Figure 1] in our [[Anonymous Link]](https://anonymous.4open.science/r/supplementaryResults_SVI-F92C/). So the rate term is actually different from the existing RDP methods.
>
> - **Almost identical means compatibility**: In Appendix A.3 (especially Lines 935~967), we show that SVI analysis is compatible with both the RDP and RD frameworks, as both are special cases of SVI under specific conditions. This leads to our training loss being almost identical to that of existing RDP schemes. However, this is not a weakness but **an advantage**—**our theory effectively explains the theoretical foundations of the existing RDP framework**. This is also mentioned by **Reviewer 7ghf** as our **strengths**.
>
> **[Q2] The experimental results raise concerns about whether the method provides any meaningful improvement.**
>
> [A2] We apologize that our implemented model design is relatively rough compared to the completeness of our theoretical analysis. The model is implemented to verify the effectiveness and potential of our SVI theory. To surpass existing methods, further exploring the model design is necessary, such as **better detail prediction and sampling module** to fit the true distribution, and **better choice on perceptual optimize direction**.
>
> To verify the potential of further perceptual optimization, we implement a discriminator to finetune our model. The results are available at the [[Anonymous Link]](https://anonymous.4open.science/r/supplementaryResults_SVI-F92C) [Figure 5]. Please check the rebuttal to **reviewer 7ghf [A2]** for the relevant experimental configuration and results analysis.
>
> **[Q3] Can the authors provide quantitative evidence that their method maintains better semantic consistency across bitrates?**
>
> [A3] Your suggestion is crucial. Verifying semantic consistency is essential as the ultimate goal of our solution optimization. To verify semantic consistency between samples in the reconstructed synset and original images across bitrates, we conducted the following supplementary tests on both $M=1$ and $M=5$ models:
>
> 1. Set the synonymous level $l$ to obtain the reconstructed samples $\tilde{\boldsymbol{x}}_i^{(l)}$.
>
> 2. Sent back $\tilde{\boldsymbol{x}}_i^{(l)}$ to $g_a$ and get a new synonymous representation ${\hat{\boldsymbol{y}}'}_s^{(l)}$.
>
> 3. Compare ${\hat{\boldsymbol{y}}'}_s^{(l)}$ with the original $\hat{\boldsymbol{y}}_s^{(l)}$ from the original image $\boldsymbol{x}$ and calculate the numerical difference ratio (Labeled as "DiffRatio").
>
> The results on Kodak are plotted in the synonymous link **[Table 1]** and **[Table 2]**. These results show good semantic consistency of our models, as the DiffRatios across every bitrate are very low (means consistency higher than 98%). However, this also means the ultimate optimization goal of SVI is not finally achieved because of the rough design of our implemented model. One of the most intuitive and objective reasons is that errors arise from the **nonlinearity** and **irreversibility** of the adopted neural network structure. This suggests that in our future work, we can explore using reversible neural network structures to address these issues.
>
> Besides, another observation is the intersection of DiffRatios at different rates for $M=1$ and $M=5$, which mirrors the intersection trend of the performance curve in Figure 5 in our manuscript. This suggests that, in future research, we can try to address the issues in Figure 5 by resolving the problems in DiffRatio.
>
> In summary, we sincerely thank you for your suggestions and will update the relevant content in the revised manuscript.

---

> > ### Comment · Reviewer_VZKv · 2025-04-08
> >
> > Thank you for your response, especially regarding A1, which has clarified my confusion on this aspect.
> >
> > As for A2, based on the results, incorporating GAN does not show a significant improvement, which does not adequately address my concern. The outcomes fail to validate the effectiveness of the theoretical method described in the paper. Simply stating that better model design leads to better results is insufficient—I believe this part of the work needs further refinement.
> > Regarding A3, first, I do not fully understand how the "numerical difference ratio (DiRatio)" is calculated. Is it obtained by directly subtracting the latent variables? Moreover, your table does not include comparisons with other methods. Without such comparisons, how can you prove that SVI leads to better semantic consistency? I find this experiment unconvincing.
> >
> > In summary, I believe the theory presented in this article is interpretable, but the overall design and experiments are not sufficiently thorough.

---

> > > ### Author Response · Authors · 2025-04-08
> > >
> > > Dear Reviewer VZKv,
> > >
> > > First of all, we would like to thank you again for **your recognition of the interpretability of our theory**. We also regret that our previous response did not fully address your concerns.
> > >
> > > Here, we provide further explanation and clarification on the two issues you raised.
> > >
> > > **[Q2'] Regarding the issue of unsatisfactory performance improvement and insufficient statement on better model design.**
> > >
> > > **[A2']** Firstly, we want to clarify the performance improvement with GAN finetuned: The improvement of DISTS is relatively obvious, while the enhancement in FID remains limited. This is a noteworthy phenomenon since **it is absent in the non-sampling schemes**, as confirmed by the experimental results in the MS-ILLM paper (Muckley et al., 2023).
> > >
> > > Unlike DISTS' resampling tolerance, FID focuses on the consistency of the distribution between the original and reconstructed image groups. This suggests that our implementation is still insufficient in optimizing the distribution of reconstructed images, **especially in our detailed sampling mechanism**.
> > >
> > > Actually, **our SVI theoretical analysis provides an ideal detail sampling principle** that **follows a conditional distribution $p_{\hat{\boldsymbol{y}}_{\epsilon,j}|\tilde{\boldsymbol{y}}_s}$ in vector form** (stated in the equations in Lines 631~639), which guides the prediction of the samples vector $\hat{\boldsymbol{y}}_{\epsilon,j}$ conditioned on synonymous representations $\tilde{\boldsymbol{y}}_s$.
> > >
> > > However, in our method design, we empirically adopted a **uniform distribution** for detail sampling (Lines 1136) performed on **each element**. We realize that this sampling method cannot fit the needed conditional distribution **in vector form**, which **cannot ensure reasonable contextual structure in the details of the reconstructed images, thereby affecting the distribution consistency focused by FID**. Although we are still exploring effective solutions to this problem, adopting a simple yet suboptimal sampling method is currently a necessary compromise. This will be a key breakthrough direction for our future research.
> > >
> > > **[Q3'] Regarding the issue of "DiffRatio" and the absence of comparisons.**
> > >
> > > **[A3']** I apologize for not clearly explaining how "DiffRatio" is calculated in my previous reply, which may have led to confusion in your understanding. Here, we define DiffRatio as
> > >
> > > $\frac{1}{n}\sum_{k=1}^n\mathbf{1}\left( \hat{y}\prime_{s,k}^{(l)}\ne \hat{y}_{s,k}^{(l)} \right), $
> > >
> > > in which $n$ denotes the number of elements in $\hat{\boldsymbol{y}}_s^{(l)}$, and $\mathbf{1}(\cdot)$ is the indicator function which outputs 1 when the inequality holds and 0 otherwise.
> > >
> > > In SVI theory analysis, the ideal reconstructed synset should cover the ideal synset, ensuring that the reconstructed images share the same synonymous representation as the original image at the corresponding synonymous level $l$. Therefore, **a DiffRatio value closer to 0 indicates that the reconstructed image is nearer to the ideal synset, reflecting better semantic consistency**.
> > >
> > > Since the optimization directions of the existing RD and RDP methods are special cases of SVI under specific conditions (in Appendix A.3., Lines 930~978), these methods can use the quantized latent feature map $\hat{\boldsymbol{y}}$ as a synonymous representation (as no detailed representation exists) and compute DiffRatio with
> > >
> > > $\frac{1}{n}\sum_{k=1}^n\mathbf{1}\left( \hat{y}\prime_k \ne \hat{y}_k \right).$
> > >
> > > We have added the DiffRatio calculation results for the RD method (Mean-Scale Hyperprior) and MS-ILLM (no-GAN & with GAN) in the [anonymous link](https://anonymous.4open.science/r/supplementaryResults_SVI-F92C) (see [Tables S1–S3 and Figure S1]).
> > >
> > > These results indicate that the RD method without perceptual optimization has a DiffRatio exceeding 10%, while the MS-ILLM method with perceptual optimization achieves a DiffRatio in the range of approximately 1%~3%, regardless of whether GAN fine-tuning is applied. This proves that the introduction of perceptual optimization improves the semantic consistency that it can establish. However, our SVI-based methods hold the DiffRatio less than 1% across most bitrate ranges and even close to 0 at high bitrate with $M=1$, which demonstrates that **our approach achieves superior semantic consistency**.
> > >
> > > **[Statement 1]** Here, we clarify that **our main contribution is to establish a unified mathematical theory for perceptual image compression, supporting existing RDP theory, and suggesting potential future research directions**. Our current rough implementation serves as a preliminary validation, demonstrating that a single model can adapt to multiple rates while approaching the performance of existing RDP methods, which aligns with our intended verification. **Achieving further performance breakthroughs will require future research efforts.**
> > >
> > > We hope the above responses help you clarify any confusion and better understand the contribution of our work.

---

### Official Review · Reviewer_uGiv · 2025-03-13

**Overall Recommendation:** 4

**Summary:**

This paper is about perceptual image compression, where previous works measure the perceptual quality by calculate certain divergence distance between the source distribution and the reconstructed distribution.

This paper is inspired by a recent advancements in semantic information theory (Niu & Zhang, 2024), where manipulating a set
of samples with the same meaning (referred as to a synonymous set, abbreviated as “Synset”) is considered as the principle of semantic information processing.

Important terminology to understand the paper:
semantic information (i.e., the meaning) and syntactic information (i.e., data samples), where one meaning can be expressed in diverse syntactic forms.

Then this paper proposes the synonymous variational inference, which is based on a modified KL divergence. This partial semantic KL divergence is defined with regard to a syntactic distribution $q$ and a semantic distribution $p_s$. For VAE based learned image compression, q is defined as parametric latent density conditional on the source image $x$, which is viewed as a sample in the ideal and unknown synset X. By minimizing this KL divergence, the output of the semantic decoder can be considered as a sample of the ideal synset, thus producing the perceptual optimized reconstruction.

**Claims And Evidence:**

Yes.

**Essential References Not Discussed:**

[Q3] There exist some previous works for semantic perceptual definition and optimization from different perspectives.

[R1] What makes an image realistic? ICML 2024.  This paper also provides an analysis of the defination of perceptual distance, which I think should be discussed as related works.

[R2] The Rate-Distortion-Perception Trade-Off with Algorithmic Realism, this paper propose a perceptual metric for individual or batch images, which is quite related to this work.

Another previous attempt to extend the RDP tradeoff for semantic compression is '[R3] Conditional perceptual quality preserving image compression', which porposes to measure the divergence conditional on semantic information and should also be discussed as related works.

In my understanding, this paper formulates semantic oriented compression problems by introducing  Synset to previous variational inference framework, which is a meaningful extension. I think the nature of synset is quite similar to the conditional perception proposed in [R3], which referes to samples (synset) or sample distribution (conditional posterior in R3) with same semantics. The correlation with those three previous works should be clarified in the manuscript.

**Experimental Designs Or Analyses:**

[Q2] The overall design is good. However, the tradeoff between distortion and perception is not illustrated as previous rate-distoration-perception works.  Another problem is the performance. The implementation did not show the advantage of this new synonymous variational formulation compared with previous perceptual image compression methods.

**Methods And Evaluation Criteria:**

Dataset includes set CLIC2020, DIV2K and Kodak, which is sufficient. [Q1] While I think FID should be evaluated as previous perceptual image compression works.

**Other Comments Or Suggestions:**

None

**Other Strengths And Weaknesses:**

This paper is based on a very new advancements in semantic information theory (Niu & Zhang, 2024), I think this paper will provide important and new insight for learned image compression community.


In equation 5, symbols like $U$ and $N_{i_s}$ are not explained. Though I can infer the meaning of those symbols, it's better to write it more clear for general readers.

**Questions For Authors:**

Please consider my concerns in previous sections [Q1][Q2][Q3]. To sum up, I like the formulation by semantic set and variational inference, which bring new insight to this field. However, the implementation did not show the advantage of this new formulation.

**Relation To Broader Scientific Literature:**

This paper might be related to perceptual image quality assessment. Which is also discussed in What makes an image realistic? ICML 2024.

**Theoretical Claims:**

I check the theorems, they are extended versions of previous derivation for rate-distortion-perception and they are correct to me.

---

> ### Author Rebuttal · Authors · 2025-04-01
>
> Dear Reviewer uGiv,
>
> Thank you for recognizing our contributions, especially the viewpoint that **our SVI theory will provide important and new insight for learned image compression community**.
>
> We believe that your questions and suggestions are crucial for improving our work. Below are our responses.
>
> **[Q1] FID should be evaluated as previous perceptual image compression works.**
>
> [A1] Your suggestion is valid. Since FID is commonly used to measure distribution similarity between image groups, it is crucial for evaluating the performance of perceptual image compression. However, previous work (MS-ILLM) shows that using adversarial loss significantly improves FID quality. Since our method roughly uses LPIPS rather than adversarial loss to replace KL divergence for optimization, its FID performance is worse than HiFiC and MS-ILLM, which is why we did not include FID comparisons in the manuscript.
>
> Since several reviewers noted our method's performance weaknesses, especially compared to GAN-based perceptual image compression, we conducted additional experiments, adding a discriminator to our method and fine-tuning the model with a non-saturating loss in the loss function. The results including FID are available at the [[Anonymous Link]](https://anonymous.4open.science/r/supplementaryResults_SVI-F92C). Please check the rebuttal to **reviewer 7ghf [A2]** for the relevant experimental setup and results analysis.
>
> **[Q2] The tradeoff between distortion and perception is not illustrated as previous rate-distortion-perception works. Besides, the implementation did not show the advantage of the method compared with previous perceptual image compression methods.**
>
> [A2] For your first sub-question:
>
> - From the perspective of our SVI theory analysis, the discussion of the tradeoff between distortion and perception is optional. If we view **Lemma 3.2** (Lines 245~258) inversely, the lemma shows that **different distortion-perception tradeoffs actually correspond to different criteria for determining ideal synonymous sets $\boldsymbol{\mathcal{X}}$**.
>
> - From the perspective of the experimental results, indeed, our previous manuscript version did not discuss the distortion-perception tradeoff problem. In our provided anonymous link, we provide the performance curves with adversarial loss optimization. As perceptual metrics like DISTS and FID improve, the distortion of the reconstructed image decreases. This confirms that our method aligns with previous work on the distortion-perception tradeoff.
>
> For your second sub-question:
>
> - Our theoretical analysis of SVI demonstrates that, under ideal conditions and with ideal model structure designs, the SIC method has the potential to outperform existing perceptual image compression methods. This is also mentioned in the rebuttal to **Reviewer 1wA9 [A1]**.
>
> - As shown in [figure 5] of the provided anonymous link, introducing the discriminator for training effectively improves perceptual quality. We note that since the discriminator is balanced across different quality levels, the fine-tuning has not fully converged, which means that there is still potential for further improvement. Nevertheless, we still consider possible issues with the current design, including **the absence of better detail prediction and sampling method** and **the better choice of perceptual optimize direction**. These issues direct the key directions for our future work.
>
> **[Q3] Some previous works for semantic perceptual definition and optimization from different perspectives should be clarified.**
>
> [A3] Thanks for providing the relevant papers. They have given deeper insights into the area of perceptual image compression and our own works. Below are some insights on the correlation with these previous works.
>
> - The paper [R1] attemps to build a universal critic of **realism**. The authors build a universal realistic critic based on *Kolmogorov complexity*. This critic allows the measure to evaluate the realism without original image reference. In contrast, our analysis originates from semantic information theory and establishes a perceptual-oriented optimization direction with original image reference.
>
> - The paper [R2] follows [R1] and proposes a new model for rate-distortion-perception tradeoff, which defines a log-likelihood ratio critic as the realistic critic without comparing to the original one, while our theory provides an expected log-likelihood ratio critic that requires that comparing.
>
> - The paper [R3] is indeed related to our works, and your understanding is completely correct. They explored the optimization direction of conditional posteriors with common features but did not consider that these conditions can serve as synonymous representations for synonymous compression. Besides, they did not approach the optimization problem from the perspective of a perceptual-similar set.
>
> We sincerely appreciate the questions you raised. These questions will help us improve our paper.

---

> > ### Comment · Reviewer_uGiv · 2025-04-02
> >
> > Thanks for the reply, after reading other reviews, I think this is a good paper. Thus I raise my score.
> >
> > Please include those new results and discussions in the final version to make the paper more complete.

---

> > > ### Author Response · Authors · 2025-04-02
> > >
> > > Thank you for your affirmation of our paper and your valuable suggestions! Your suggestions have indeed improved our work! We will incorporate these new results and discussions into the revised manuscript to strengthen it further.

---

### Official Review · Reviewer_1wA9 · 2025-03-14

**Overall Recommendation:** 4

**Summary:**

This paper proposes synonymous variational inference and introduces synonymous image compression. It is based on the observation that a given image to be encoded has a set of synonymous images that share the same semantic meaning. Instead of optimizing the variational distribution at the pixel level, we optimize it at the semantic level. Building on this insight, we propose a simple modification to the existing training scheme to achieve synonymous variational inference.

**Claims And Evidence:**

Yes. The claims are supported by experiments and proof.

**Essential References Not Discussed:**

N/A

**Experimental Designs Or Analyses:**

The experiments are sound to me. Also, the author provides an ablation study on M.

One thing I am not sure of is that the proposed algorithm is simply modifying the latent code with a deterministic one and a stochastic one. What if we do not use the stochastic code, but keep all other components, like loss / network to be the same? Will this reduce the performance?
Or to say, my question is: is the performance gain due to the SVI proposed in this paper, or the loss / network / other tricks?

Disclaimer: I am familiar with the Rate-Distortion-Perception trade-off in general, but I only know some old baselines in this area, like HiFiC, and I am not familiar the recent development in this area. Therefore, I cannot evaluate the significance of the performance gain. Also, I my questions regarding the performance/experiments might be biased. If that is the case, feel free to correct me.

**Methods And Evaluation Criteria:**

Yes. The paper reports DISTS and BPP, and also report LPIPS and PSNR in appendix.

**Other Comments Or Suggestions:**

N/A

**Other Strengths And Weaknesses:**

This paper is well-structured and well-written. It is easy to follow and is a pleasure to read.

**Questions For Authors:**

Please see above

**Relation To Broader Scientific Literature:**

The SVI proposed in this paper is new to me. I think this result is insightful yet simple enough to employ. I believe this is a good contribution to this area.

However, I cannot confidently evaluate the significance of the performance of this approach.

**Theoretical Claims:**

I checked the outline of the proof in the main text.

---

> ### Author Rebuttal · Authors · 2025-04-01
>
> Dear Reviewer 1wA9,
>
> Thank you for recognizing our work, especially our proposed analysis theory, i.e., **synonymous variational inference (SVI)**, as **a good contribution to the area of perceptual image compression**.
>
> We think that your questions are valuable for improving our work. Below are our responses.
>
> **[Q1] Is the performance gain due to the SVI proposed in this paper, or the loss / network / other tricks?**
>
> [A1] We address your question from two perspectives: **theory and method**.
>
> - **Theory**: According to the proof and compatibility analysis in Appendix A of our manuscript, SVI can offer theoretical potential performance advantages for two reasons:
>
>   - **Stochastic Effectiveness**. As Equation (15) (especially Lines 636~639) presents in our manuscript, each sampled random detail corresponds to a sample $\tilde{\boldsymbol{x}}_j$ in the reconstructed synset. When the ideal and reconstructed synsets fully overlap, that sample should match an image $\boldsymbol{x}_j$ in the ideal synset, which is perceptually similar to the original one. Thus, the reconstruction quality can be supported by both coded synonymous representations and random details, rather than relying solely on the code sequence as in HiFiC and MS-ILLM.
>
>   - **Encode only common features**. As Equation (31) (Lines 862~868) and (32) (Line 874) present, ideally, the coded rates of the synonymous representation reflects the rates of common features of the reconstructed synset $\tilde{\boldsymbol{\mathcal{X}}}$, i.e, the minimized single-side semantic mutual information $I(\boldsymbol{X};\tilde{\mathring{\boldsymbol{X}}})$, rather than the minimized mutual information $I(\boldsymbol{X};\tilde{\boldsymbol{X}})$ between the original image and specific reconstructed sample, in which the latter one is greater (line 925-926). This can be intuitively illustrated as the Venn Diagrams [Figure 1] in our [[Anonymous Link]](https://anonymous.4open.science/r/supplementaryResults_SVI-F92C/), and means that our compression limits will be lower than the existing methods with the same distortion and perceptual quality constraints ideally.
>
> - **Method**: The advantages of our proposed scheme including:
>
>   - **Multi-rate adaptability**. This relies on **Stochastic Effectiveness**, since the adaptability across various rates is achieved by determining partial of the random details to the accurate representation of the source image, with the guidance of our derived loss function.
>
>   - **Performance Improvements on DISTS**. This relies on both the two theoretical advantages, since the difference on the details is more acceptable to DISTS and human perception than other measures like LPIPS; besides, the compression efficiency can be further improved by encoding only synonymous representations.
>
> However, we admit that our proposed SIC method struggles to fully achieve SVI's theoretical advantage due to **the absence of better detail prediction and sampling method** as well as **better perceptual optimize direction like adversarial losses**. These are key directions for future research.
>
> To verify its potential for subsequent optimization, we plot some additional experimental results to the [[Anonymous Link]](https://anonymous.4open.science/r/supplementaryResults_SVI-F92C), in which we finetune our model with a CNN-based discriminator using non-saturating loss. Please check the rebuttal to **Reviewer 7ghf [A2]** for the relevant experimental configuration and results analysis.
>
> **[Q2] What if we do not use the stochastic code, but keep all other components, like loss / network to be the same? Will this reduce the performance?**
>
> [A2] Your suggestion is crucial since it touches on a core question we are concerned with: whether random details positively support both distortion and perceptual quality. Therefore, we added an ablation test result in our provided anonymous link [Figure 2~3], where we force the random details $\hat{\boldsymbol{y}}_{\epsilon,j}$ to 0, preventing them from providing detail information. We obtained test results at each rate under $M=1$ and $M=5$ and compared them with the performance of random sampling. All other conditions remain unchanged.
>
> The experimental results show that without random sampling, both models reduce the performance significantly in distortion and perceptual quality. This indicates that random details contribute effective information to the reconstructed image's distortion and perceptual quality, i.e., verifying the **Stochastic Effectiveness** mentioned above. Besides, it also suggests that perceptual image compression performance can be supported not only by coding sequences as existing RDP methods do. Based on this phenomenon, we conclude that when designed properly, the SVI-based method has the potential to surpass the performance of the existing RDP methods.
>
> Again, we appreciate your affirmation of our work and the questions you feedback. We will update our manuscript accordingly.

---

### Decision · Program_Chairs · 2025-05-01

**Decision:**

Accept (poster)

**Comment:**

The paper proposes a theoretical framework for perceptual image compression grounded in semantic information theory, introducing the concept of synonymous variational inference (SVI). It further presents an implementation called SIC, and evaluates its performance across standard datasets.
Several reviewers pointed out that while the empirical performance compared to existing methods like GAN are limited, theoretical formulation is clear and well-motivated and extends existing rate distortion perception frameworks in a principled manner. The authors provided additional experiments and clarifications in response to several concerns, including analysis of semantic consistency and comparisons under various metrics.
Although some practical aspects of the implementation might be improved, the theoretical contribution is clear and seem to be useful for the community. Based on these, I recommend acceptance.